## Registered report

psychology

preregistration, registered reporting, trustworthiness, questionable research practice

**Author for correspondence:**
Sarahanne M. Field
e-mail: s.m.field@rug.nl

# The effect of preregistration on trust in empirical research findings: results of a registered report

Sarahanne M. Field[1,†], E.-J. Wagenmakers[2],
Henk A. L. Kiers[1], Rink Hoekstra[1], Anja F. Ernst[1]
and Don van Ravenzwaaij[1]

[1]Department of Psychometrics and Statistics, Rijksuniversiteit Groningen, Groningen, The Netherlands
[2]Department of Psychology, University of Amsterdam, Amsterdam, The Netherlands

(iD) SMF, 0000-0001-7874-1261

The crisis of confidence has undermined the trust that researchers place in the findings of their peers. In order to increase trust in research, initiatives such as preregistration have been suggested, which aim to prevent various questionable research practices. As it stands, however, no empirical evidence exists that preregistration does increase perceptions of trust. The picture may be complicated by a researcher's familiarity with the author of the study, regardless of the preregistration status of the research. This registered report presents an empirical assessment of the extent to which preregistration increases the trust of 209 active academics in the reported outcomes, and how familiarity with another researcher influences that trust. Contrary to our expectations, we report ambiguous Bayes factors and conclude that we do not have strong evidence towards answering our research questions. Our findings are presented along with evidence that our manipulations were ineffective for many participants, leading to the exclusion of 68% of complete datasets, and an underpowered design as a consequence. We discuss other limitations and confounds which may explain why the findings of the study deviate from a previously conducted pilot study. We reflect on the benefits of using the registered report submission format in light of our results. The OSF page for this registered report and its pilot can be found here: http://dx.doi.org/10.17605/OSF.IO/B3K75.

The crisis of confidence in psychology has given rise to a wave of doubt within the scientific community: we now question the quality and trustworthiness of several findings upon which

†Department of Psychometrics and Statistics, University of Groningen, Groningen, The Netherlands

researchers have built entire careers, and the trust of our peers is more difficult to earn than it once was [1–3]. In response to the crisis, in what can be called a 'methodological revolution' [4,5], several new initiatives have been launched which focus on improving the quality of published research findings, and on restoring the trust of the scientific community in that higher-quality research. Preregistration and registered reporting (henceforth PR and RR, respectively) are two such initiatives.

PR is the process in which a researcher articulates her plans for a research project—including study rationale, hypotheses, design and analysis and sampling plans—before the data are collected and analysed. A growing number of researchers choose to upload preregistration documents, which lay out these plans, onto sites such as the open science framework (OSF), AsPredicted.org, or onto their personal websites. Others choose to send their plans to colleagues on a departmental mailing list [6,7]. The plans can be made public, be released only to reviewers, or be kept private, depending on the preference of the author. RR extends the preregistration process, involving the peer-review of the preregistration document through a journal, just as in the review process of a complete research report. Once the preregistration plan has been accepted, the study has been accepted in principle, and the researcher can begin conducting the study. Crucially, a journal's acceptance of the RR comes with outcome-independent publication [8].

A 'critical part of urgent wider reform' [9], both PR and RR are explicitly geared towards lending credibility to research findings. They benefit the researcher, the quality of their output, and the integrity of the entire field's literature. They may lead to increased study reproducibility [10]. The popularity of preregistration and RR appears to be gaining momentum in the scientific community. For many, it is now part of the regular research process [4]. This move is popular with individual researchers and academic publishers alike. In 2013, 80 researchers authored an open letter to *The Guardian*, petitioning others to take up preregistration practice [9]; to date, more than 140 major journals accept RRs. These include *Royal Society Open Science*; *Cortex*; *Perspectives on Psychological Science*; *Attention, Perception, and Psychophysics*; *Nature Human Behavior*; and *Comprehensive Results in Social Psychology* (see https://cos.io/rr/ for an up-to-date list of participating journals).

The increasing adoption of PR and RR is accompanied by a growing body of literature that argues for its benefits (see [11,12]). One major benefit is that the author can be transparent about the research process. This lends credibility to research because it allows for the claims contained in that research to be more thoroughly verified (as opposed to unregistered studies) [13]. Another benefit is that they provide researchers with a means of distinguishing between exploratory and confirmatory research. Authors often unwittingly present exploratory research as confirmatory. This is problematic because it heavily influences the validity of the statistical testing procedure, and changes the interpretation of 'statistical significance'. Hypothesizing after the results are known (HARKing; [14]) is a questionable research practice (QRP), in which researchers blur this crucial distinction. As PR and RR protocols require the researcher to register their hypotheses before they collect data, the researcher is inoculated against the effects of their own hindsight and confirmation biases [15].

P-hacking, when researchers influence the data collection or analysis process until a statistically significant result appears, and cherry-picking, in which one chooses to use or report only those dependent variables which were statistically significant, are two more well-known examples of QRP [16]. They undermine the reliability of research findings, in that they produce results that misrepresent the true nature of a study's findings, typically making them look more compelling than they actually are. P-hacking gets counteracted by the explicit description of the sampling plan and the procedure for outlier removal provided by the author. Any deviation from this in the final manuscript has to be justified. Cherry-picking is prevented by specification of the hypotheses and expected results before the data are seen.

The file drawer phenomenon [17] refers to another common QRP, in which researchers do not write up their null findings for publication. This QRP is similar to publication bias, a QRP perpetrated by academic journals, whereby the results of articles influence whether or not an outlet publishes them. The file drawer problem and publication bias are jointly responsible for the construction of a literature that does not faithfully represent all the research findings that have been produced. RR targets these QRPs specifically: when null results are found, authors will not automatically relegate them to the file drawer, as participating RR journals typically guarantee outcome-independent publication. When they accept RRs, journals commit to outcome-independent publishing, and are usually bound to that agreement, providing the author has adequately adhered to the originally reviewed and accepted study methodology and sampling plan. There are often other stipulations made by the journal, such as making the data public.

Simmons *et al*. [3] state that QRPs are usually committed in good faith. They typically arise from ambiguity as to which methodological and statistical procedures to follow, in conjunction with a desire to report interesting and eye-catching findings, which is fuelled by the incentive structures in

place for publication. Unfortunately, QRPs are not uncommon: an estimated 94% of researchers admit to having committed at least one QRP (possibly a conservative estimate, see [16]). This high rate is not the preserve of psychological research: a similar report on QRP has recently emerged for ecology and evolution biology research based on data from 807 participants [18].

Further benefits to adopting PR and RR are directly relevant to the researcher's career and academic reputation [19]. It is thought that researchers will produce higher-quality, more reproducible research. This will ultimately benefit them as scientists, for different reasons. It is possible that researchers' academic work will be trusted more by other academics when PR or RR have been part of the research process. By adopting PR and RR, proponents argue that researchers will increase their chances of getting articles accepted by journals, regardless of whether or not the results obtained favour their hypotheses.[1] Finally, they may be more confident in trusting the work of colleagues in their own fields if they know that others' work has been preregistered, or is a registered report. In the specific case of RR, authors will benefit from extra review and input on their methodology before they conduct the study. This allows them to save their time and resources for the highest-quality studies.

The merits of PR and RR are advocated by many researchers worldwide: by high-profile meta-science researchers such as Chambers [20], Wagenmakers [19] and Nosek [4], as well as by the general population of academics. Despite this, it is unclear whether the scientific community at large trusts the quality of the results born of the PR or RR process more than studies which have not been subject to these processes. Indeed, we are unaware of any published direct empirical assessment. It is vital to establish whether researchers do trust PR and RR protocols more than those unregistered, because without trust, PR and RR will not truly become part of standard practice in psychology, and the quality of our studies may not continue to increase.[2]

While rebuilding the quality of our evidence base is vital, so too is the trust researchers place in other researchers [21]. One's judgement of information credibility is directly related to one's perception of the credibility of its source [22], and it is often the case that researchers must evaluate the quality of a research article from an author (or author's body of work) with whom they are familiar. As such, these elements within the research process—familiarity and trust—go hand in hand, and should be evaluated in conjunction with one another. But what is the mechanism that determines the extent to which we trust our academic colleagues, and their research output? This mechanism may be understood through the recognition or familiarity heuristic; a cognitive heuristic that can act to assist us in making judgements about the credibility of information based on the familiarity of its source [23]. Another possible explanatory mechanism may be ingroup bias—we may favour the research output of ingroup others, leading us to trust those findings relatively more than those of outgroup members. In the context of the proposed study, familiarity with a publication's authors is expected to lead to increased trust in the findings of that study. We cannot predict the precise nature of this relationship, however.

# 1. Research questions and hypotheses

The research questions under investigation were: (1) *does preregistering a research protocol increase the trust a fellow researcher places in the resulting research findings?*, (2) *does familiarity with the author of a research study increase the trust a fellow researcher places in the resulting research?* and (3) *does familiarity combine with preregistration status such that findings which feature both a familiar author and preregistration are maximally trustworthy?*

We expected that PR increases the trust participants have in research results over no preregistration at all, and expected that RR evokes the most trust in participants overall. We had less clear expectations about familiarity: we expected that familiarity increases trust to some extent, relative to no familiarity overall. We also expected familiarity would garner higher trust in conjunction with preregistration, relative to no familiarity.

This article contains the results of our registered report. The methodology we describe was peer reviewed, and subsequently granted 'in principal acceptance'. The experimental materials, participant information statement and data associated with the current study and a previously conducted pilot, described briefly below, are available on the project's OSF page: http://dx.doi.org/10.17605/OSF.IO/B3K75.

---

[1]Whether this benefit is actually reaped by those who use PR as well as for those who submit RRs has yet to be established—it is possible that acceptance of null results remains challenging for publications even when they have been subject to preregistration.

[2]It is important to note that determining whether or not registered findings are actually more credible, due to higher quality and a decreased risk of QRP, is not our aim. We are focusing on how researchers *perceive* registered studies. We assume that researchers expect registered studies to be of higher quality than their non-registered counterparts, but we emphasize that the actual credibility of a study is ultimately more important than how it is perceived.

## 2. Pilot study

Prior to submitting the plan for the full study described in this article, we conducted a pilot study. It featured simpler stimuli, and was conducted on a smaller sample, compared with the full study. The pilot led us to further develop our methods and provided compelling support for our hypotheses, therefore reinforcing our expectations. The pilot strongly supported the first of our hypotheses: our analysis yielded extreme evidence in favour of preregistration status having a true effect on trust, given the data. Familiarity did not compellingly influence trust for unregistered studies; however, it did increase trust for the preregistration and registered report conditions. Finally, the pilot results showed that the model including the two main effects and their interaction was also strongly supported. Though the experimental design for the pilot study is similar to those described in the full study, they are not the same and comparisons should be cautious. The analysis strategies are identical. A detailed description of the pilot appears in the Phase 1 registered report proposal document. It is available on the OSF page, along with the materials used, the raw and processed data, the analysis code and other associated documentation: http://dx.doi.org/10.17605/OSF.IO/B3K75.

## 3. Current study

We designed this study to attempt to empirically determine the role of preregistration status in influencing perceptions of trust in research findings, and to determine if familiarity with the author of the research reinforces those trust perceptions.

## 4. Method

### 4.1. Participants and recruitment

This study aimed to obtain information from all actively publishing researchers in the broad field of psychology. As a proxy, we used the population of authors from psychology articles on the Web of Science (WoS) database for a period of several years.[3]

The recruitment procedure we followed is complex due to recruitment for the pilot study, and our later decision to collect extra data for the full study. For clarity, we have put the recruitment, filtering and exclusion steps into a schematic (figure 1) which accompanies the following summary.

The first part of the sample was originally obtained as part of recruitment for the pilot study. A total of 9996 email addresses were initially extracted from the WoS, from article records between January 2013 and May 2017, and the resulting sample was randomly split into two groups. In addition, we extracted 3267 addresses, updating the $N$ to 8265. These two steps resulted in the sample of researchers contacted in the first sweep of data collection. Via the Qualtrics browser-based survey software suite, we sent a total of 8265 emails for the first sweep. Of these, 1084 emails were bounced due to expired or incorrect email addresses or inbox spam filters. We were left, therefore, with $N = 7181$.

As we describe later, we mined the WoS for more records in 2009 and 2010 to help reach our minimum overall $N$ of 480. This yielded 2449 emails, of which 611 bounced. The total number of mined email addresses over both batches is 10 714. The number of email addresses which received an initial contact email from us is the sum of the two batches of email addresses extracted, meaning that, in total 9019 people were successfully contacted (i.e. 8.4% of the contact emails sent out were not delivered).

We observed a response rate of just over 6%. The response rate for the first sweep was approximately 6.7%, while the rate for the second was around 4%. This means that the number of people who completed the study *before* exclusions is 654. Further exclusions are made due to participants failing the manipulation checks. We discuss these in the results section. The participants were not remunerated for their participation.

### 4.2. Design

To investigate the influence of preregistration status and familiarity on trust, we employed a $2 \times 3$ ANOVA design. Our two independent variables were thus preregistration status (with three

---

[3]The original aim was to use only corresponding authors, however, in approximately 9 per cent of cases, more than one author's email address was given. In these cases, all author email addresses were used.

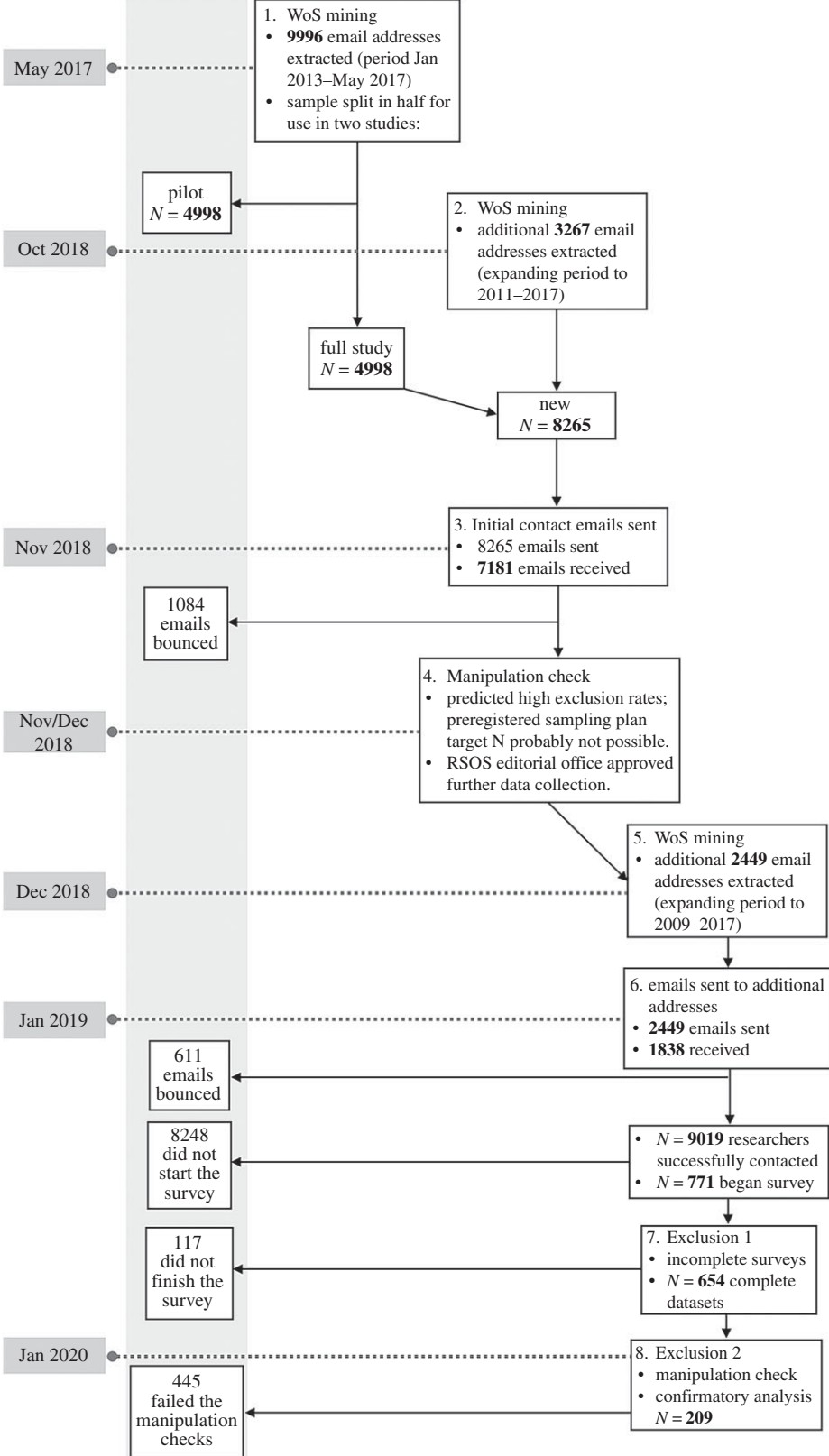

**Figure 1.** Visual representation of the recruitment procedure as a series of steps. WoS refers to the Web of Science.

levels—none, PR and RR) and familiarity (with two levels—familiar and unfamiliar). We asked participants: 'How much do you trust the results of this study?', and measured trust through the rating participants entered on a 1–9 Likert-scale.

### 4.2.1. Materials

Six fictional research scenarios were constructed for this study according to the study's six design cells, in which preregistration status and familiarity were manipulated. Six conditions resulted from the factorial combination of three preregistration statuses (none, PR and RR) and two familiarity types (unfamiliar, familiar).

### 4.2.2. Construct operationalization

#### 4.2.2.1. Independent variables: preregistration and familiarity

In each hypothetical study scenario, preregistration status and familiarity were manipulated in the text. Preregistration status reflected whether or not the fictional study protocol had been preregistered in some way before data collection. In the scenario texts, preregistration status was described in terms of the process involved. With the descriptions used, we hoped to broadly capture the main differences between different levels of preregistration: none at all, preregistration and registered report. To avoid the use of potentially loaded terminology, we described preregistration and registered reporting in terms of their processes.

The familiarity manipulation was intended to simulate a commonly encountered situation in the academic setting, whereby collaboration often coincides with one researcher being familiar with another to the extent that they would possibly look upon their work as being more credible and trustworthy in comparison with the work of a complete stranger. This study features familiarity as an exploratory variable, included as a main effect in the analysis we describe later in this manuscript.

#### 4.2.2.2. Dependent variable: trust

A measurement of trust was taken for each participant once they had read their fictional scenario. The word 'trust' was chosen, as it best encompassed the dimensions we wanted to test. We intended trust to capture the judgement people made based on their perceptions of the credibility of the fictional study, but is a broad enough word such that people can interpret it in different ways. We considered that it could be interpreted as 'valid', 'reliable', 'reproducible' or to indicate the robustness of the effect presented in the fictional scenario. We found the possibility of different interpretations of the word 'trust' desirable, as people use different heuristics and definitions to establish a trust judgement when it comes to evaluating the research output of others: we wanted to capture as many of these interpretations as possible.

All scenario texts and a trial of the study as it appeared to participants can be found on the project's OSF page: http://dx.doi.org/10.17605/OSF.IO/B3K75.

All participants received the same instructional text:

> We aim to measure your direct or 'primary' response to the scenario on the next page. To that end, please attempt to answer the question on the following page without thinking at length about it. You may use any of the information given on the next page to answer the question.

In addition to this initial text, a participant in the **none/unfamiliar** condition received the following author profile text:

> A researcher with whom you have never collaborated (and with whom you are unlikely to collaborate in the future) has recently conducted a study. The study was subsequently published in a peer-reviewed journal. The paper makes no mention of any previously documented sampling plan or study design.

In the **none/familiar** condition, the author profile text becomes (changes in bold):

> A researcher **with whom you have collaborated previously (and with whom you would collaborate again)** has recently conducted a study. The study was subsequently published in a peer-reviewed journal. The paper makes no mention of any previously documented sampling plan or study design.

All receive either the unfamiliar or familiar condition text as described above, depending on which condition they are assigned to. The difference between none, PR and RR condition texts is as follows. A participant in the **PR/unfamiliar** condition received:

> A researcher with whom you have never collaborated (and with whom you are unlikely to collaborate in the future) has recently conducted a study. The study was subsequently published in a peer-reviewed journal. Prior to conducting this study, a detailed sampling plan, analysis strategy, and the study hypotheses were posted publicly on the author's Open Science Framework (OSF) page.

For those in the **RR/familiar** condition, this text was presented instead:

> A researcher with whom you have collaborated previously (and with whom you would collaborate again) has recently conducted a study. Prior to conducting the study, its protocol was peer-reviewed and conditionally accepted by the editorial committee at the publishing journal. The study was subsequently published.

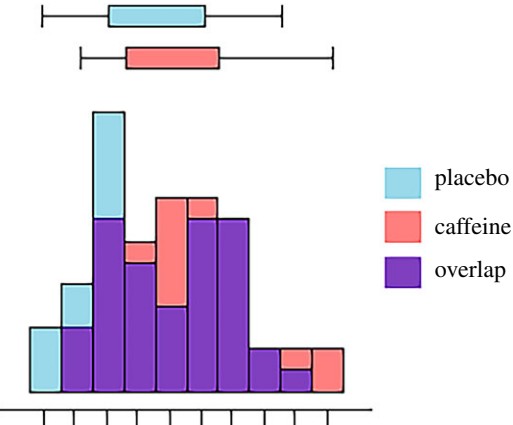

**Figure 2.** The boxplots and histograms show the mental rotation error rates for the two experimental conditions after caffeine administration.

Once the instruction and author profile has been read, participants were presented with the following scenario text and data plot:

Introduction
We aimed to investigate the relationship between caffeine and visual representation in the brain. We hypothesized that participants given a high dosage of caffeine would make more errors in the mental rotation task, compared with participants administered a placebo.

Methods
*Participants.* The experimental sample comprised 100 participants—52 females (mean age = 26.5) and 48 males (mean age = 25) that had been recruited from the general public in London, UK. Normal or corrected-to-normal vision and full use of both hands was a requirement for participation. Participants with a known allergy to caffeine were excluded, as were participants who reported drinking an excessive amount of coffee daily (greater than 4 cups per day).

*Design.* To ensure gender balance in the experiment, males and females were separately randomly assigned to the 'caffeine' and 'placebo' conditions: both conditions contained 26 females and 24 males. The caffeine group were given a 'high-dose' (400 mg) caffeine tablet with a 8-ounce glass of water 45 min before the mental rotation task, while the placebo group were given a placebo pill with an 8-ounce glass of water 45 minutes before the task.

The experiment was double-blind, and the pills administered were visually indistinguishable. The dependent variable was error rate on a mental rotation task, quantified as total percentage incorrect in 40 trials.

Materials
In each of the 40 trials, two objects were presented alongside one another on a computer monitor (display dimensions: 336.3 mm × 597.9 mm) in one of 8 rotation angles ($0°$, $45°$, $90°$, etc.). The trial presentation order was randomized. Each trial stimulus appeared as black 3D figures on a white background as in figure 1. Participants indicated whether or not the right-hand object was the same as the left-hand object ('same') or different ('different'). To respond to each object pair, participants pressed either the 'S' key for 'same', or the 'D' key for 'different'. The task was not speeded.

Results
A one-tailed independent samples *t*-test was conducted to compare mental rotation error rate between the two groups. As predicted, the caffeine group had a significantly higher error rate than the placebo group: $t(98) = 2$, $p = 0.024$, $d = 0.40$. The data are shown in figure 2.

Once the participant read the scenario, they clicked to the next page, on which they found the key experimental question:

How much do you trust the results of this study? (Please note that the word 'trust' in this question can be interpreted in many ways: 'reliable', 'replicable', 'a true effect', 'valid' or 'high quality'. Any such interpretations of trust are acceptable for the purposes of answering the question.)

A sliding scale accompanied this question, which allowed possible responses ranging from 1 *Not at all* to 9 *Completely*. Responses recorded on the sliding scale numerically captured participants' ratings of how trustworthy they thought the results presented to be.

## 4.3. Manipulation checks and follow-up questions

After participants answered the primary experimental question about trust, we posed two manipulation questions—one for each of the independent variables.

The questions were:

(1) *In the fictional scenario you just read, was the researcher responsible for the study someone with whom you were familiar?*

(2) *In the fictional scenario you just read, was there either: (a) mention of any previously documented sampling plan or study design? or (b) mention of the study protocol being peer-reviewed and conditionally accepted by the publishing journal?*

A participant in any of the three unfamiliar conditions was expected to answer 'no' to such a question, while the other three conditions is expected to answer in the affirmative. We also asked participants whether the study had been preregistered in some way, or whether it was the subject of a registered report, or whether neither of these conditions are true. A participant in either of the none conditions was expected to answer 'no', while participants in the other conditions was expected to indicate that there was either PR or RR present.

The data of people who indicated that they did not note the manipulations in the scenario they were given, or indicate something unexpected (e.g. familiarity with the study author if they are in the unfamiliar condition) as revealed by the manipulation check, were excluded from the analysis. We describe the exclusion procedure in detail shortly.

At the recommendation of a reviewer in the first phase of the registered report, we included some follow-up questions. These were intended to help occlude the manipulation check questions, and by extension, the purpose of the study, ensuring that people did not discuss the study before data collection was complete, thereby contaminating the data.

## 4.4. Procedure

All participants received an email containing an invitation to participate via the Qualtrics survey suite which included a link to one of the six scenario surveys. The emails sent for initial contact and follow-up can be found at the project's OSF page: http://dx.doi.org/10.17605/OSF.IO/B3K75.

Participants were randomly assigned to presentation of one of the scenarios described in the Materials section by the Qualtrics-programmed randomizer. When a researcher chose to participate, he or she clicked on a link, which directed the browser to one of the six conditions in the survey. In the first screen, participants were presented with the instruction message, and clicked through using a 'Next' button to the scenario screen, after which they were presented with the primary research question.

Next, the participants were required to move the on-screen slider to indicate their trust judgement on the 1–9 scale, and click 'Next'. The participants were then given the manipulation check and follow-up questions regarding their opinions about PR and RR. Note that for each participant, the order of the questions was randomized. Finally, the participants were thanked for their time, and were given access to the participation information statement about the study's aims and other relevant information. This statement is available on the project's OSF page: https://osf.io/49d2h/.[4] The survey was not speeded, but participants were asked not to dwell too much on the trustworthiness question, and instead rely on instinct as trained researchers regarding their trust in the results of the scenario presented.

Participants were not permitted to go back to earlier parts of the experiment once they click the 'Next' button. This was to prevent the possibility of going back and changing responses once the aims of the study became more obvious by the follow-up questions. Once data collection was finished, a 'thank you' email was sent, which included an explanation of the study aims.

# 5. Results

With the exception of the sampling strategy (which we describe below) the full study described in this article followed our Phase 1 registered report proposal precisely. All data, code and JASP files are available on our OSF page: https://osf.io/b3k75/.

## 5.1. Sampling strategy deviation

We had planned to collect data such that each cell in our six-cell design would contain $N = 80$. After collecting data for two weeks (with the planned reminder prompt email sent after one week), we had over 900 responses. At this point, the first author checked how many exclusions would be made, finding that many participants failed the manipulation check to the extent that our minimum $N$ per

---

[4]During data collection, the participant information statement was only available via the link shown to participants, not via the OSF page.

**Table 1.** Conditions, expected answer pattern for each manipulation check question (A1, expected answer to question 1, A2, expected answer to question 2) , N per group before and after exclusions, total N excluded per condition, with percentage.

| condition | A1 | A2 | pre-excl. N | post-excl. N | N excl. (%) |
|---|---|---|---|---|---|
| none/fam | no | yes | 110 | 44 | 66 (60) |
| none/unfam | no | no | 120 | 84 | 36 (30) |
| PR/fam | yes | yes | 110 | 12 | 98 (89) |
| PR/unfam | yes | no | 106 | 12 | 94 (89) |
| RR/fam | yes | yes | 115 | 24 | 91 (79) |
| RR/unfam | yes | no | 92 | 33 | 59 (64) |

**Table 2.** Descriptive statistics for each condition pre- and post-exclusions.

| condition | pre-N | pre-mean (s.d.) | post-N | post-mean (s.d.) |
|---|---|---|---|---|
| none/fam | 110 | 4.845(1.823) | 44 | 4.818(1.846) |
| none/unfam | 120 | 4.867(1.796) | 84 | 5.060(1.891) |
| PR/fam | 110 | 5.455(1.816) | 12 | 6.167(2.250) |
| PR/unfam | 106 | 5.208(1.798) | 12 | 5.333(1.557) |
| RR/fam | 115 | 5.357(1.812) | 24 | 5.792(1.774) |
| RR/unfam | 92 | 5.087(1.838) | 33 | 5.182(1.911) |

cell was not met. Hence, data collection stayed open for the remainder of the month as planned; however, only four more responses were collected after this point. As planned, the first author conducted a preliminary analysis, and determined that there was not sufficient evidence to either support or contradict our hypotheses, and contacted the *Royal Society Open Science* editorial office. The editorial office confirmed that more data could be collected, and so, as per our registered report proposal, we once again mined WoS, extracted more email addresses and collected a second wave of data.

At the conclusion of the second data collection phase, we had the complete data of 654 participants. After excluding participants who had failed the manipulation checks, we were left with a total N of 209 meaning that we excluded approximately 68% of the participants. For clarification, please consult figure 1: a schematic of the changes to the sample from recruitment to the data analysis.

## 5.2. Exclusions

As explained above, we included manipulation check questions in the survey, and excluded participants from our analysis based on the participants' answers. We asked: *In the fictional scenario you just read, was the researcher responsible for the study someone with whom you were familiar?*, and *In the fictional scenario you just read, was there either: (a) mention of any previously documented sampling plan or study design? or (b) mention of the study protocol being peer-reviewed and conditionally accepted by the publishing journal?*

We excluded participants who answered incorrectly (i.e. 'yes', when 'no' was expected, or 'no' when 'yes' was expected) or answered that they did not note the manipulation. This resulted in an unexpectedly high number of exclusions, leaving a final sample of 209 participants over our six design cells. Unfortunately, this small sample was distributed very unevenly across the conditions. Table 1 shows this information for each condition. Please consult figure 1 for a visualization of how the sample size has changed from recruitment to analysis, and where the manipulation check exclusion fits in this picture.

## 5.3. Confirmatory analysis

As mentioned in earlier sections, 9019 participants were successfully contacted, but only 209 datasets remained for our planned analysis. Table 2 shows the group Ns, in addition to the cell means and SDs for the data pre- and post-exclusions.

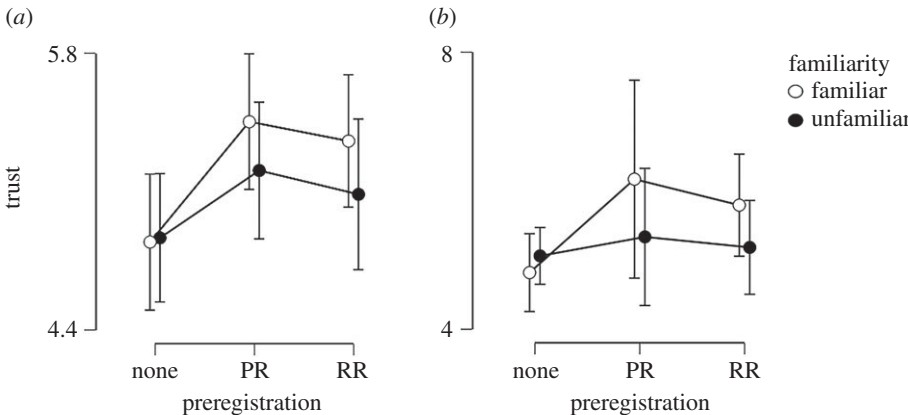

**Figure 3.** Trust ratings for the six experimental conditions from the data *before* (plot *a*) versus *after* (plot *b*) exclusions. Error bars represent 95% credible intervals (which result from, in this case, 2.5% being cut off from each end of the posterior distribution). Being derived from a uniform prior, the intervals are numerically identical to 95% confidence intervals. Note that the *y*-axes do not carry the same range.

**Table 3.** Possible models, their prior probabilities, posterior probabilities and the Bayes factors.

| models | P(M) | P(M\|data) | $BF_M$ | $BF_{10}$ |
|---|---|---|---|---|
| null model | 0.200 | 0.545 | 4.787 | 1.000 |
| preregistration | 0.200 | 0.282 | 1.570 | 0.517 |
| familiarity*Fam | 0.200 | 0.108 | 0.485 | 0.199 |
| PR + Fam | 0.200 | 0.048 | 0.204 | 0.089 |
| PR + Fam + PR*Fam | 0.200 | 0.017 | 0.068 | 0.031 |

It can be seen from figure 3 that the credible intervals are very wide, and there is no discernible pattern in the plotted mean points other than a weak trend towards the same findings as in the pilot study.

As planned, we conducted a between-subjects Bayesian ANOVA choosing all priors according to default settings in JASP[5] ([24,25]) using the statistical software JASP ([24–26]; and see https://static.jasp-stats.org/about-bayesian-anova.html). Table 3 shows the standard Bayesian ANOVA comparisons between individual models. Column *P*(M|data) shows that after observing the data, the most plausible model of those under consideration is the null model.

Perhaps a more informative way to interpret the results is by means of *inclusion* Bayes factors. Inclusion Bayes factors ($BF_{inclusion}$) allow one to compare all models that include any given predictor (e.g. in this context either preregistration status or familiarity) with all those that do not, to determine the relative strength of evidence for each factor on trustworthiness (the dependent variable), based on the data.[6] The JASP inclusion Bayes factor analysis yielded ambiguous results, as shown in table 4. We now describe the results in the context of their support of the hypotheses.

Observing the current data did little to change the odds for a model including the preregistration variable. That is, given our prior distributions on the effect size parameter, there is more support for exclusion of the preregistration variable and the familiarity variable in our model, than for inclusion of them. This said, the absence of any real evidence means that we cannot make claims for anything in terms of the two main variables. We may only conclude that we need more data. Regarding their interaction, however, given our prior distributions over effect size, there is strong support for leaving the interaction term out of the models.

[5]JASP's Bayesian statistics calculations for ANOVA are based on Rouder and Morey's BayesFactor R package. We used the default setting for the prior distributions.

[6]A brief worked example of this approach using the data obtained in the pilot study is shown in appendix A.

**Table 4.** Confirmatory analysis: effects, their prior probabilities, posterior probabilities and the inclusion Bayes factors.

| effects | *P*(incl) | *P*(incl\|data) | BF$_{\text{Incl.}}$ |
|---|---|---|---|
| preregistration | 0.600 | 0.350 | 0.359 |
| familiarity | 0.600 | 0.177 | 0.143 |
| PR*Fam | 0.200 | 0.017 | 0.069 |

**Table 5.** Exploratory analysis: effects, their prior probabilities, posterior probabilities and the inclusion Bayes factors.

| effects | *P*(incl) | *P*(incl\|data) | BF$_{\text{Inclusion}}$ |
|---|---|---|---|
| preregistration | 0.600 | 0.516 | 0.710 |
| familiarity | 0.600 | 0.150 | 0.117 |
| PR*Fam | 0.200 | 0.004 | 0.016 |

## 5.4. Unregistered (exploratory) analysis

A reviewer suggested that, in light of so much data being discarded for the confirmatory analysis, readers may be interested in seeing the results for the full sample. Given this, and uncertainty regarding the validity of the manipulation check questions (which we touch on in the discussion section), we now present the results of the study for all cases (i.e. the data without exclusions) in addition to our preregistered confirmatory analysis. The results of the Bayesian ANOVA on the full data are shown in table 5. As with the confirmatory analysis, the null model (i.e. without the two independent variables included) is the most likely model given the data observed. Comparison of tables 4 and 5 shows that results for the exploratory and confirmatory analysis are qualitatively similar.

# 6. Discussion

This study was designed to assess whether researchers put more trust in the findings of others when these findings have been obtained from a preregistered study or a registered report, as opposed to from a study conducted without any plans stated prior to data collection. We had three questions: (1) *does preregistering a research protocol increase the trust a fellow researcher places in the resulting research findings?*, (2) *does familiarity with the author of a research study increase the trust a fellow researcher places in the resulting research?* and (3) *does familiarity combine with preregistration status such that findings which feature both a familiar author and preregistration are maximally trustworthy?*

After conducting a promising pilot study, we sought to answer these questions by administering an experiment in which we presented fictional study vignettes that differed from one another in the level of preregistration. We asked participants to rate their trust of the findings presented in those scenarios. Our aim was to have at least 80 participants per cell satisfy the inclusion criteria (i.e. based on recognized successful manipulation). Unfortunately, a manipulation check revealed that 68% of participants were not reliably manipulated, leading to group *N*s of less than 50 in all conditions except for one.

Possibly as a result of this, our analysis showed that the evidence for our main hypotheses was ambiguous ('not worth more than a bare mention' according to [27, Appendix B]). We obtained strong evidence in favour of leaving the interaction between the two independent variables out; however, given that the independent variables do not reflect meaningful results, this result too is unlikely to be substantial.

## 6.1. Limitations

### 6.1.1. Exclusions

A surprisingly high percentage of the sample failed the manipulation checks, leading to exclusion of 68% of the sample. It is possible that people did not understand what we were asking due to complexities in the wording of the questions, or a lack of clarity in the link between the questions and the study materials

the questions referred to. These participants may have answered incorrectly as a result (even though they had, in fact, been manipulated successfully) or have just guessed the right answers. In hindsight, it would have been useful to conduct a small pilot on the materials after they were developed for the full study, as we did for the original materials in the pilot study; an approach Houtkoop *et al.* [28] have recently used. This may have flagged potential problems with the manipulations early enough such that we could have adjusted our materials to prevent losing so much data.

This being said, it is possible that the manipulations did in fact work for more people than our checks showed, but that the manipulation was so subtle that it influenced some people's responses without them consciously noticing that influence. If this were the case though, it would be difficult if not impossible to separate those participants from others who were genuinely unsuccessfully manipulated.

Given the concerning number of participants who failed the manipulation checks, we consider it possible that people generally did not understand the study even though they answered as 'expected' in the checks (perhaps they were guessing in the manipulation check questions and happened to answer correctly). Although the lack of power in the study makes it hard to say anything substantial, it could be that we would not find the anticipated effects even if we had sufficient power.

### 6.1.2. Materials

In response to receiving overwhelming suggestions by participants in the pilot study to make the scenario texts more elaborate and realistic, we substantially changed our original materials.[7] The original materials included a few simple paragraphs describing the results of a study (either a preregistered one, a registered report, or a traditional one with neither) and side-by-side box plots; the current study featured a realistic mini-study vignette. With this change we aimed to facilitate participants' immersion into the content, as if they were reviewing the study 'for real'.

The switch to a realistic mini-study vignette may have had an adverse effect on the response pattern, however. Specifically, participants might have glanced over the scenario page, decided that it would take them too much time or effort to respond (either thinking there was too much reading to be done, or that the scenario was too complex for the five minutes we suggested they spend on it) and given up on completing the study. It may be that the more realistic complex materials are best to test our hypotheses, but that the 'quick and easy' 5 min online survey our study features is too simple to capture the effect. It should be noted, however, that the scenario texts were presented to participants *after* the manipulation texts. Therefore, it is unlikely that quick and careless reading by participants directly led to them failing the manipulation check. On the other hand, it is possible that because a large block of text was presented after the manipulations and before the primary experimental question, the information in the manipulations was forgotten by the time the manipulation check questions were presented. This does not mean that the manipulation was ineffective necessarily (because participants may have been manipulated unconsciously), only that our manipulation check is likely to be unreliable.

As discussed earlier, participants were not given a 'Back' button in each page of the survey. This was designed to safeguard the dependent variable responses against being contaminated by the questions that appeared after the manipulations and key trust question. This meant that participants were unable to review their scenario or manipulation texts after having seen the trustworthiness question. Although we consider this choice justifiable, it is possible we should have started out saying 'read the following research scenario, in the end you will be asked to indicate to what extent you trust the results', such that people would be able to prepare themselves for the eventuality of the question, without giving them the option of going back and contaminating their answers.

We also consider that it is possible that explicit mention of the well-known online repository OSF in the preregistration conditions may have increased people's trust in the findings. Perhaps people attach credibility to individuals who use services like the OSF, because when an author chooses to upload study materials, data and code, they are acting with good faith in a very tangible way. In hindsight, it may have been a good idea to add information about materials and data sharing onto the registered report's conditions explicitly, in addition to describing the registered report process, with in principal acceptance. We are not investigating people's faith in OSF of course, but it is conceivable that people's faith in OSF (and people who use it) influenced their faith in the results we presented in the PR conditions.

---

[7]In our pilot, we used open questions to probe why participants might have had trouble answering 'how much do you trust these results?' which helped us understand how people interpreted what we had asked them. A content analysis of these responses is available on OSF (https://osf.io/w6acb/).

Finally, we note our use of a single dependent measure of trust. A reviewer has pointed out that, in light of such unclear results, it would have been useful to have more data on the dependent variable. Although one of the aims for this study was to make the survey as simple, quick and easy as possible for participants to respond, we recognize the issues with validity and reliability that a single-indicator measure might present.

### 6.1.3. Sample size

In general, a larger $N$ leads to a more compelling Bayes factor [29]. It is likely, therefore, that whatever the true state of the world is in terms of our predictions (that is, are preregistered/registered report findings *really* perceived as more trustworthy by peers?), this effect would be reflected in the inclusion Bayes factor with greater participant numbers. That is, our current findings do not teach us very much about the true state of the world, given that our $N$ is so small.

## 6.2. Changing opinions

Support for preregistration and registered reports is growing (over 200 journals now accept the registered report format, for instance: https://cos.io/rr/#journals) to the extent where it may become the norm [30]. Recently, there seems to be an increase of discussions in the scientific community on the complexities that surround using preregistration and registered reports. This is another factor that may have influenced peoples' responses in this study (especially relative to our pilot's compelling pro-effect findings).[8] Twitter hosts discussions about these protocols regularly, demonstrating that researchers are thinking critically about conditions in which preregistration and registered reports are less useful or are inappropriate in some fashion. For instance, there is concern that registered reports/preregistration 'dampens exploratory work' [31,32], restricts authors' freedom [33], or takes too much time to do [34]. Other concerns revolve around the results of articles featuring preregistered plans—people may be hesitant to trust the results because they are unsure of the degree to which authors stuck to their plans. This concern is not entirely without merit. Claesen, Gomes, Tuerlinckx and Vanpaemel recently uploaded a preprint [35] in which they report that none of the 27 studies they scrutinized had completely adhered to their preregistered plans.

We asked participants extra questions to probe their opinions about the protocols and reasons for participation. Of the participants that provided answers to our question about their thoughts on preregistration and registered reports, an overwhelming number (263 of the 545 qualitative responses collected) thought them 'great' and 'useful'. However, 105 indicated that a more complex answer to the question was warranted, and chose to type in their own answer. Several of these answers suggested that the utility of preregistration and registered reports was defined by either the kind of research question, or type of study design. One individual stated of preregistration and registered reporting: ' … it largely depends on the question and its setting. For straightforward experimental work it is certainly a good idea and useful', while another wrote 'Useful in practice when there is a sufficiently focused research Q.' A further 104 participants indicated they were neutral about the protocols. Eight people were of the opinion that preregistration and registered reports are bad initiatives. Even a quick look at the qualitative results to the probe reveals that over half of participants have some reservations about the value of preregistration and registered reports. This, however, does not explain why it appears that so many people have failed to register the relevant manipulation information.

## 6.3. Concluding reflections

The results of this registered study are disappointing in that they are uninformative. This is in sharp contrast to our pilot's results, which were both predicted and compelling. On a more constructive note, however, the experience of conducting this study, and obtaining the findings we did is valuable, providing food for thought on several points. Firstly, the findings being different to what we had expected prompts reflection on what the registered report format offers to good scientific practice. The registered report model for publishing ensures that messy, ugly studies like ours see the light of day (it is no secret that clean, positive results are much more likely to be published [36]), and that the research literature may be an increasingly accurate reflection of the true state of the world.

---

[8]It does not explain the large amount of exclusions, however.

It allows researchers the freedom to focus on several important aspects of studies: the research questions and their theoretical relevance, the quality of the research methodology and its relationship to the hypotheses, and, in terms of the results—what is exploratory and what is confirmatory. We may reflect on these aspects of the experiment without interference from the outcomes (regardless of whether they support the hypotheses or not). We are also free to write up the results of our study faithfully and transparently without fear of rejection by the journal, providing we adhered to our plans (or sufficiently rationalized if we did not). In the case of the current study, we can be transparent about the trouble we had with our sample, our findings, and that we cannot conclude anything from the study. This may then serve as a warning to other researchers who may attempt to study a similar phenomenon.

Although our experiment results failed to meet our expectations, we remain of the opinion that registered reports and studies using preregistration are of higher quality, and therefore more trustworthy. However, the picture is perhaps more complex than our design could capture. It is, of course, true that people can find ways to 'hack' preregistration and registered reporting, and do the wrong thing if they so desire, but our cautiously optimistic view is that, on average, researchers will use these protocols to produce higher-quality science in good faith. That being said, the use of preregistration and registered reporting does not replace critical engagement of researchers with the literature they consume.

Data accessibility. All raw data, code, analysis files and materials associated with this study are available on its OSF page: https://osf.io/b3k75/.

Authors' contributions. D.v.R., E.-J.W., R.H. and A.F.E. conceived of the original idea to study the link between PR/RR and trust; S.M.F., H.A.L.K., D.v.R., E.-J.W., R.H. and A.F.E. developed the idea for testing in the pilot and full study; S.M.F. collected all data, processed and analysed it; D.v.R. wrote all code associated with the study; S.M.F. led the writing and editing of the manuscript; all other authors provided feedback. All authors gave final approval for publication.

Competing interests. We declare we have no competing interests.

Funding. We received no funding for this study.

Acknowledgements. We are grateful for Angelika Stefan's assistance with conducting a custom Bayes Factor Design Analysis for ANOVA.

# Appendix A. An explanation and hypothetical worked example describing inclusion Bayes factors

As discussed in the results section, using inclusion Bayes factors we can compare all models that include any given factor (e.g. preregistration) with all those that do not, to determine the relative strength of evidence for each factor on trustworthiness, based on the data, by means of the inclusion Bayes factor. This allows us to quantify the relative evidence of a factor affecting the outcome, provided by the data, by taking the ratio of the posterior inclusion odds and the prior inclusion odds. Below is a hypothetical worked example of the calculation of inclusion Bayes factors (tables 6 and 7).

**Table 6.** Possible models, their prior probabilities, posterior probabilities and the Bayes factors.

| effects | P(M) | P(M\|D) | $BF_M$ | $BF_{10}$ |
| --- | --- | --- | --- | --- |
| 1. null model | 0.20 | 0.00038 | 0.00 | 1.000 |
| 2. preregistration | 0.20 | 0.812 | 17.33 | 2142.34 |
| 3. familiarity | 0.20 | 0.000093 | 0.00037 | 0.25 |
| 4. familiarity + preregistration | 0.20 | 0.175 | 0.85 | 462.60 |
| 5. familiarity * preregistration | 0.20 | 0.012 | 0.05 | 30.62 |

**Table 7.** Effects, their prior probabilities, posterior probabilities and the inclusion Bayes factors.

| effects | P(incl) | P(incl\|D) | $BF_{incl}$ |
| --- | --- | --- | --- |
| 1. preregistration | 0.60 | 1 | 1409.97 |
| 2. familiarity | 0.60 | 0.19 | 0.15 |
| 3. interaction | 0.20 | 0.01 | 0.05 |

In the absence of data, we can assume *a priori* that each of these five models is equally likely: $p = 0.2$. Using this information, we can calculate the prior inclusion probability of 'preregistration' by adding the prior probabilities of all models that contain the preregistration effect (i.e. models 2, 4 and 5) as $0.20 + 0.20 + 0.20 = 0.60$.

After the data come in, we are able to calculate the posterior inclusion probability. In this example, the posterior probabilities of the three models (2, 4 and 5, in the $P(M \mid D)$) are 0.812, 0.175 and 0.012. The posterior inclusion probability of preregistration is therefore $0.812 + 0.175 + 0.012 = 0.999$, while the posterior probability of models *without* the effect of preregistration is the sum of the posterior probabilities of models 1 and 3.

Once the prior and posterior inclusion probabilities are obtained, we can calculate $BF_{incl}$: The change from prior inclusion odds to posterior inclusion odds. The inclusion Bayes factor for the preregistration effect is therefore:

$$BF_{incl} = \frac{0.999 / (0.0003792 + 0.00009315)}{0.60 / (1 - 0.60)}$$

$$= 1409.97$$

In this context, having observed the data, the odds for a model including preregistration status have increased by a factor of nearly 1410. An inclusion Bayes factor can be calculated for the interaction effect between preregistration status and familiarity in the same manner.

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
