## [Reviewer comments · Royal Society Open Science]

Review History

RSOS-180599.R0 (Original submission)

Review form: Reviewer 1

Is the language acceptable?

Yes

Do you have any ethical concerns with this paper?

No

Have you any concerns about statistical analyses in this paper?

No

Recommendation?

Major revision

Comments to the Author(s)

Summary: The proposed study is exciting and important. I'm highly supportive of the study moving forward; however, more information needs to be provided in the proposal to a) allow

replication (see responses to criteria 3, 4, and 5 below) and b) justify (and operationalize) the familiarity hypothesis (as discussed in criterion 2 below).

1. The importance of the research question(s).

The question this study proposes to answer is incredibly important. I could drone on and on here but suffice to say: The importance of the research question is paramount and is the feature of the proposed study that reviewed the best.

2. The logic, rationale, and plausibility of the proposed hypotheses.

RE: the hypothesis that “preregistration increases researchers’ trust in findings, relative to no preregistration, and that registered reporting increases trust more than preregistration alone,” I find the logic and rationale quite sound and highly plausible.

RE: the hypothesis that “familiarity enhances the effect of preregistration on trustworthiness ratings,” I’m sorry to say that the manuscript doesn’t provide enough information to be convincing. One problem, which I’ll discuss below, is that the description of how familiarity will be operationalized is far too vague in the proposal.

Another problem is that the authors appear to be proposing that familiarity is an additive factor to the preregistration continuum. I’m not sure why familiarity is proposed as an independent, additive factor rather than, say, an interactive factor. If pre-registration increases trust, and if familiarity also increases trust, how do we know that these two features might not cancel one another out or otherwise interact?

Unfortunately, there wasn’t much in the proposal to support the prediction of familiarity as an additive factor – or much in the proposal to rule out the prediction of an interactive factor, for that matter. In fact, the familiarity manipulation felt quite exploratory to me (“Our primary interest is in the effects of preregistration protocols on trust, so we only consider familiarity as a moderating variable on the relationship between trust and preregistration protocols, not as a main effect on its own”).

Exploratory variables are fine. But they should be identified as such.

Related, whereas the three levels of the registration hypothesis (none, preregistered, registered report) are externally established and have a relatively agreed-upon definition, the two levels of the familiarity hypothesis seem to lack similar validity and clarity.

As I’ll mention below, I had to go to OSF, download six separate PDFs, and compare and contrast them, to get any idea how familiarity (or pre-registration for that matter) was being operationalized. That’s a problem for the manuscript that I’ll address below. But here I’ll say that after I pulled out this information, I learned that familiarity is being manipulated by the contrast between participants who read this sentence:

“A researcher with whom you have collaborated previously (and with whom you would collaborate again) has recently conducted a study.”

versus participants who read this sentence:

“A researcher unknown to you has recently conducted a study.”

To me, this manipulation doesn’t seem to be about familiarity. The contrast isn’t between a researcher you know and a researcher you don’t know. Rather, the contrast is between a researcher with whom you’ve collaborated (and would collaborate with again) and a researcher

whom you don't know. That's not what I expected prior to excavating the materials, and at the least needs to be better addressed in the proposal.

Lastly, on the topic of the familiarity hypothesis (which is definitely the weaker of the two hypotheses), I think the authors need to better operationalize what it means that a "researcher is unknown" to participants.

Does that mean you've never heard the researcher's name before; you've heard their name, but you've never read any of their papers; you've read one of their papers, but you wouldn't say you 'know' the researcher; you've read a lot of their work, but you've never met them in person, so you still wouldn't say 'know' the researcher?

I'm by training a language researcher, and a good rule of thumb in language research is that the control condition is specified as well linguistically as the contrast condition. Otherwise, one is inviting all sorts of noise when participants interpret the under-specified control condition in various ways. So, I'd recommend that if the sentence establishing the contrast condition has nearly 15 words and two descriptive clauses, i.e.

"with whom you have collaborated previously (and with whom you would collaborate again)"

the sentence establishing the control condition should have a similar number of words and a similar number of descriptive clauses, e.g.,

"with whom you haven't collaborated previously (and with whom you're unlikely to collaborate in the future)"

3. The soundness and feasibility of the methodology and analysis pipeline (including statistical power analysis where applicable).
4. Whether the clarity and degree of methodological detail would be sufficient to replicate the proposed experimental procedures and analysis pipeline.
5. Whether the authors provide a sufficiently clear and detailed description of the methods to prevent undisclosed flexibility in the experimental procedures or analysis pipeline.

I've grouped together these three criteria because my comments relate to all three. I have little to say about the analysis pipeline, but with regard to the methods, more information needs to be provided to assess its soundness and considerably more information needs to be provided to allow future attempts at replication.

Here's a list:

Why was Web of Science chosen? Do the authors believe the same results would be obtained if Google Scholar (a more egalitarian indexing program) or another indexing program is used for participant selection?

How were "psychology articles" operationalized (i.e., "we used the population of corresponding authors from psychology articles")? For example, was it required that the journal name contain the word "psychology"? If not, what were the criteria?

Which authors were recruited? All authors on each article or only the corresponding author?

In the pilot study, what did the "contact email" message say and who sent it? Similarly, what did the invitation email message say and who sent it? (I ask about who sent it because my guess is that participants who are more open to preregistration would be more likely to respond to both messages if they came from a researcher publicly known for advocating preregistration.)

In the proposed study, what will the invitation email say and who will send it? (I looked on OSF and could find only the six scenarios, no other relevant material.)

In the pilot study, were participants compensated? In the proposed study, will the participants be compensated? If so, what will be the compensation?

In the pilot study, how were the different versions of the scenario assigned to participants? (In the proposed study, the manuscript explains the randomization scheme will be conducted by Qualtrics.)

When is the Trust Question asked? The manuscript implies that the trust question is asked after participants read the scenario: "In the first screen, participants are presented with the instruction message, and can click through using a 'next' button to the scenario screen, after which they are presented with Question 1." (Question 1 is the Trust Question.)

However, the scenario PDFs posted on OSF suggest that the question is asked before participants read the scenario. The first sentence on each scenario PDF says the following: "We aim to measure your direct or 'primary' response to the scenario on the next page. To that end, please attempt to answer the question given without thinking at length about it. You may use any of the information given on the next page to answer the question.") The authors' use of the definite-article expression "the question" implies that that "the question" has already been asked.

If the Trust Question is not asked until after participants read the scenario, then the instructions that appear on the scenario PDFs are a bit confusing and need to be cleaned up.

Regarding the new manipulation check questions, I'm confused why they will be asked only for the scenarios that lead to a "yes" answer. For example, as I understand the authors, only for the familiar condition will participants be asked, "In the fictional study you just read, was the researcher responsible for the study someone with whom you were familiar?" And only in the preregistered or registered reports condition will participants be asked, "whether the study had been preregistered in some way, or whether it was the subject of a registered report."

Why not employ a manipulation check for all conditions? In other words, why not ask the same (or similar) manipulation check question for all scenarios? I also recommend adding in a couple of other neutral questions to better occlude the nature of the manipulation check question.

Returning to the scenario PDFs, it was deeply unfortunate that the only way I could see the manipulation was to (as I whined about above) click over to OSF, download six different files, and then compare each of the six files one to one another. That's asking a lot of reviewers, particularly for information that a) needs to be provided in the manuscript, and b) is simple to provide in the manuscript!

But, from what I could discern, the Familiar versus Unfamiliar conditions differ in that the Familiar scenarios have the sentence, "A researcher with whom you have collaborated previously (and with whom you would collaborate again) has recently conducted a study" whereas the Unfamiliar scenarios have the sentence I wrote about above, "A researcher unknown to you has recently conducted a study."

And, from what I could discern, the Registration conditions differ in that the None scenarios include the sentence, "The paper makes no mention of any previously documented sampling plan or study design," the Preregistered scenarios include the sentence, "Prior to conducting this study, a detailed sampling plan, analysis strategy, and the study hypotheses were posted publicly on the author's website," and the Registered Report scenarios include the sentence, "Prior to conducting the study, its protocol was peer-reviewed and conditionally accepted by the editorial committee at the publishing journal."

If all the above is true, this information must be stated in the manuscript.

Two more concerns, both relatively serious: First, both the pilot study and the proposed study are lacking demographic information about the participants. Given the 12.5% response rate, we really need to know who chose to participate in this type of study. As I alluded to above, different samples could provide different responses (e.g., samples of researchers who are enthusiastic about preregistration are most likely to provide responses that indicate that preregistration boosts trust).

Second, there is no justification provided for designing the study as between- rather than within-subjects. Because of carryover effects and subject demand (fatigue), a between-subjects might be the best choice, but no justification is provided.

6. Whether the authors have considered sufficient outcome-neutral conditions (e.g. absence of floor or ceiling effects; positive controls; other quality checks) for ensuring that the results obtained are able to test the stated hypotheses.

The addition of manipulation checks and some quality control checks is a good move.

One last and small quibble. The manuscript states: “the merits of PR and RR are advocated by a group of scientists often referred to as ‘meta-scientists’.” I know numerous scientists who advocate PR and RR, myself included, and none of us refer to ourselves as ‘meta-scientists.’ We’re just scientists. In other words, it’s not just “meta-scientists” who advocate pre-registration.

Review form: Reviewer 2

Is the language acceptable?

Yes

Do you have any ethical concerns with this paper?

No

Have you any concerns about statistical analyses in this paper?

I do not feel qualified to assess the statistics

Recommendation?

Major revision

Comments to the Author(s)

Comments on points specified by the editor, as well as some additional comments, are all stated in the attachment (Appendix A).

Review form: Reviewer 3

Is the language acceptable?

Yes

Do you have any ethical concerns with this paper?

No

Have you any concerns about statistical analyses in this paper?

No

Recommendation?

Accept with minor revision

Comments to the Author(s)

Review for RSOS-180599: "The Effect of Preregistration on Trust in Empirical Research Findings: A Registered Report Proposal"

Overall, I think that this RR proposal is a valuable study, that promises to shed more light onto the (psychological) effects of preregistration. As the field is moving forward in implementing open science practices, an evaluation of their effectiveness and side-effects is an important step, to which this papers adds. In general I support an IPA of this proposal, but have some issues I'd like to see addressed before:

1. More precise hypotheses

In the research questions you suddenly introduce "objective" trust. What is that? Please define. Furthermore, you write: "We expect that preregistration does increase the trust participants have in research results, and that familiarity increases trust in conjunction with preregistration." What does "in conjunction" mean? Is that a hint for an interaction effect? If yes, in what direction?

2. Description of results of pilot study

You write on p. 5: "Incorporating familiarity appears to led to the expected behavior of the dependent variable- familiar researchers led to a slight increase participant trust, over and above what alone was explained by PR and RR." This description seems misleading, as the BF does not support this claim.

If I read the BF tables correctly, there is no evidence for the inclusion of familiarity (in contrast, there is moderate evidence for not including it). Hence, there is no support for your hypothesis that "familiarity increases trust", and this conclusion should be explicitly written down.

For readers unacquainted with BF, please verbally interpret the BF for the interaction term.

Table 2 seems misplaced at its current position - why not report that as an exploratory result of the pilot study?

3. Proposed study

I guess you want to generalize the results to a variety of preregistered studies - this would actually call for a design where the study content (in the experimental material) is a random factor. I realize that this could lead to power problems, but I think at least a discussion of the issue is warranted.

Is the reported power analysis on p. 7 a frequentist power analysis? Seems quite odd in the context of a purely Bayesian data analysis. Why not report a Bayes factor design analysis (see Schönbrodt & Wagenmakers, 2017) for a fixed-n design, i.e., $\text{prob}(\text{BF} > 10 \mid \text{assumed effect size})$.

I think one underappreciated factor of the replication crisis is the low reliability of our measurements. Many social psychology studies rightly have been criticized for measuring their central variables with ad-hoc single-item measures. I would urge the authors to think about how to increase the reliability of the central DV measure in this study as well; from a psychometric point of view using a single item for a multifaceted construct seems careless.

On p. 9 you write: "In the absence of data, we assume a priori that each of the five models in our two-factor design is equally likely ($p = .20$). Well, you do have data from the pilot study. Why not updating your priors from pilot  main study?"

4. Minor comments:

- p. 3: "Researchers' work is taken more seriously and its academic content trusted more by others when PR or RR have been part of the research process.": Here (and throughout this paragraph) is either a citation needed, or it should be made clearer that this is a hypothesis and not an empirical finding. Do we already have evidence for that claim, or is that what we expect? Isn't the current project just about finding that out?

- In the pilot study, and also the main study: Did/will you explain the difference of PR and RR to participants? In my experience, this is not clear to many.

Here are the journal's check points:

The significance of the research question(s): Relevant.

The logic, rationale, and plausibility of the proposed hypotheses: Makes sense.

The soundness and feasibility of the methodology and analysis pipeline (including statistical power analysis where applicable): Please improve power analysis (see above); anything else is state of the art.

Whether the clarity and degree of methodological detail would be sufficient to replicate exactly the proposed experimental procedures and analysis pipeline: Sufficient.

Whether the authors provide a sufficiently clear and detailed description of the methods to prevent undisclosed flexibility in the experimental procedures or analysis pipeline: Sufficiently clear.

Whether the authors have considered sufficient outcome-neutral conditions (e.g. positive controls) for ensuring that the results obtained are able to test the stated hypotheses: They added a manipulation check.

Review form: Reviewer 4

Is the language acceptable?

Yes

Do you have any ethical concerns with this paper?

No

Have you any concerns about statistical analyses in this paper?

No

Recommendation?

Major revision

Comments to the Author(s)

The current project proposes to examine the effect of study pre-registration (vs. RR vs. control) and collaborator familiarity on trust.

Major concerns:

1. Conceptualization of trust: You propose to add explanation text to the study defining trust (which was not contained in the pilot study): "the word 'trust' in this question can be interpreted in many ways: 'reliable,' 'replicable,' 'a true effect,' 'valid,' or 'high-quality.'" (p. 8). This text was helpful for me, because I do not feel that the construct of trust was fully explained in the introduction. I personally had a strong (negative) reaction to the word "trust." Trust implies that checking someone's work is not needed, but the benefit of pre-registration is that it enables you to check someone's work.

All of these open science initiatives that we have proposed and enacted are designed to increase research/evidence quality. It doesn't matter to me whether we can "trust" a finding more. Trust should be an outcome of quality, but it is not the same thing as quality. A pre-registered or RR finding is more credible than an unregistered study, because we can verify its claims more fully. That is, transparency in detailing the data generating process makes it possible to verify a claim. RR studies may be de facto higher quality because of improvements introduced in the two-stage review process (but again, quality is the desired outcome; trust should be irrelevant). Let me try to be as clear as I can: pre-registered studies should not be trusted more, but on average, the data contained in them are more valuable, because it is possible to verify the claims the authors make. I wouldn't say that value is the same thing as trust, either.

I will add that it is a separate question about whether researchers *do* trust pre-registered or RR studies more than traditional studies (versus whether they *should*), but it seems weird to me to couch higher trust as the desired outcome. We should care about the quality of evidence, not about trust. For instance, you write "Rebuilding trust in our research findings is vital" (p. 3), but I would rephrase this to "Increasing the quality of our evidence base is vital." Similarly, "several new initiatives have been launched which focus on restoring the trust of the scientific community in its research" (p. 2). But this seems to me to be a mischaracterization, and it makes it sound like we are just doing reputation management rather than trying to improve research quality.

2. Measurement of trust: Unless I missed it, I do not see the full text of the question asking about trust in the text or the supplemental materials. It should be provided, and you might consider putting your Qualtrics (.qsf) file on OSF too. I reviewed the materials on OSF and did not see anything like this there. A single item ad hoc measure of the key dependent variable seems to leave a lot to be desired.

3. You write "It is unclear whether the scientific community at large trusts the results borne of the PR or RR process more than studies which have not been subject to these processes" (p. 3). I think this is the question your study is designed to answer. It may be interesting to know this, but perhaps not for the reasons you outline in the introduction.

Additional point:

4. When I was reviewing the materials on OSF, I noticed that you state the following in your ethics application: "As aforementioned, participants will be naive to the exact objectives of the study, and will be only exposed to the stimulus of 1 (of 6) experimental conditions. Naivety is important for this study as social desirability may be expected to play a role in the participants' responses. We want to ensure that the participants' responses are not impacted by what they might think is expected of them." This seems to be inconsistent with the "participant information form" which gives away many of the details of the study including the focal variables. Perhaps I misunderstood and maybe this form is a debriefing form rather than a consent form, but I would urge you to reconsider your framing in the participant information form to try to preserve participant naivety.

Signed,
Katie Corker

Decision letter (RSOS-180599.R0)

23-May-2018

Dear Ms Field,

The Editors assigned to your Stage 1 Registered Report ("The Effect of Preregistration on Trust in Empirical Research Findings: A Registered Report Proposal") have now received comments from reviewers. We would like you to revise your paper in accordance with the referee and editors suggestions which can be found below (not including confidential reports to the Editor). Please note this decision does not guarantee eventual acceptance.

Please note that Royal Society Open Science will introduce article processing charges for all new submissions received from 1 January 2018. Registered Reports submitted and accepted after this date will ONLY be subject to a charge if they subsequently progress to and are accepted as Stage 2 Registered Reports. If your manuscript is submitted and accepted for publication after 1 January 2018 (i.e. as a full Stage 2 Registered Report), you will be asked to pay the article processing charge, unless you request a waiver and this is approved by Royal Society Publishing. You can find out more about the charges at <http://rsos.royalsocietypublishing.org/page/charges>. Should you have any queries, please contact openscience@royalsociety.org.

Kind regards,
Andrew Dunn
Royal Society Open Science
openscience@royalsociety.org

on behalf of Chris Chambers (Registered Reports Editor, Royal Society Open Science)
openscience@royalsociety.org

Associate Editor Comments to Author (Professor Chris Chambers):

Associate Editor: 1

Comments to the Author:

Four reviewers have now assessed your submission. The standard of reviews is high, with all providing a range of constructive and critical suggestions from different perspectives. The general view is that the proposed research question is important and interesting, satisfying the first criteria of Registered Reports. However the reviewers also raise a number of significant

issues that cut across the remaining Stage 1 criteria and will need to be addressed thoroughly in revision.

There are too many issues raised to cover fully in summary, so to provide some guidance I will focus on the main points raised. Foremost is the need to clarify (and re-assess) the rationale, validity and precision of the familiarity hypothesis (Reviewers 1, 2 and 3), address potential confounds in the operationalisation of preregistration and concerns with the sampling plan and falsifiability of the hypotheses (Reviewer 2), consider the reliability of the principal DV and suitability of the power analysis (Reviewer 3), address concerns with the conceptualisation and measurement of trust (Reviewer 4), and with the precision of the hypotheses involving trust (Reviewer 3). Each set of reviews also raises questions about the justification and clarity the research question and methodology.

Substantive work is therefore required to achieve IPA, but the concerns appear readily addressable as part of a major revision, which will be returned to the reviewers.

Comments to Author:

Reviewer: 1

Comments to the Author(s)

Summary: The proposed study is exciting and important. I'm highly supportive of the study moving forward; however, more information needs to be provided in the proposal to a) allow replication (see responses to criteria 3, 4, and 5 below) and b) justify (and operationalize) the familiarity hypothesis (as discussed in criterion 2 below).

1. The importance of the research question(s).

The question this study proposes to answer is incredibly important. I could drone on and on here but suffice to say: The importance of the research question is paramount and is the feature of the proposed study that reviewed the best.

2. The logic, rationale, and plausibility of the proposed hypotheses.

RE: the hypothesis that "preregistration increases researchers' trust in findings, relative to no preregistration, and that registered reporting increases trust more than preregistration alone," I find the logic and rationale quite sound and highly plausible.

RE: the hypothesis that "familiarity enhances the effect of preregistration on trustworthiness ratings," I'm sorry to say that the manuscript doesn't provide enough information to be convincing. One problem, which I'll discuss below, is that the description of how familiarity will be operationalized is far too vague in the proposal.

Another problem is that the authors appear to be proposing that familiarity is an additive factor to the preregistration continuum. I'm not sure why familiarity is proposed as an independent, additive factor rather than, say, an interactive factor. If pre-registration increases trust, and if familiarity also increases trust, how do we know that these two features might not cancel one another out or otherwise interact?

Unfortunately, there wasn't much in the proposal to support the prediction of familiarity as an additive factor - or much in the proposal to rule out the prediction of an interactive factor, for that matter. In fact, the familiarity manipulation felt quite exploratory to me ("Our primary interest is in the effects of preregistration protocols on trust, so we only consider familiarity as a moderating variable on the relationship between trust and preregistration protocols, not as a main effect on its own").

Exploratory variables are fine. But they should be identified as such.

Related, whereas the three levels of the registration hypothesis (none, preregistered, registered report) are externally established and have a relatively agreed-upon definition, the two levels of the familiarity hypothesis seem to lack similar validity and clarity.

As I'll mention below, I had to go to OSF, download six separate PDFs, and compare and contrast them, to get any idea how familiarity (or pre-registration for that matter) was being operationalized. That's a problem for the manuscript that I'll address below. But here I'll say that after I pulled out this information, I learned that familiarity is being manipulated by the contrast between participants who read this sentence:

"A researcher with whom you have collaborated previously (and with whom you would collaborate again) has recently conducted a study."

versus participants who read this sentence:

"A researcher unknown to you has recently conducted a study."

To me, this manipulation doesn't seem to be about familiarity. The contrast isn't between a researcher you know and a researcher you don't know. Rather, the contrast is between a researcher with whom you've collaborated (and would collaborate with again) and a researcher whom you don't know. That's not what I expected prior to excavating the materials, and at the least needs to be better addressed in the proposal.

Lastly, on the topic of the familiarity hypothesis (which is definitely the weaker of the two hypotheses), I think the authors need to better operationalize what it means that a "researcher is unknown" to participants.

Does that mean you've never heard the researcher's name before; you've heard their name, but you've never read any of their papers; you've read one of their papers, but you wouldn't say you 'know' the researcher; you've read a lot of their work, but you've never met them in person, so you still wouldn't say 'know' the researcher?

I'm by training a language researcher, and a good rule of thumb in language research is that the control condition is specified as well linguistically as the contrast condition. Otherwise, one is inviting all sorts of noise when participants interpret the under-specified control condition in various ways. So, I'd recommend that if the sentence establishing the contrast condition has nearly 15 words and two descriptive clauses, i.e.

"with whom you have collaborated previously (and with whom you would collaborate again)"

the sentence establishing the control condition should have a similar number of words and a similar number of descriptive clauses, e.g.,

"with whom you haven't collaborated previously (and with whom you're unlikely to collaborate in the future)"

3. The soundness and feasibility of the methodology and analysis pipeline (including statistical power analysis where applicable).
4. Whether the clarity and degree of methodological detail would be sufficient to replicate the proposed experimental procedures and analysis pipeline.
5. Whether the authors provide a sufficiently clear and detailed description of the methods to prevent undisclosed flexibility in the experimental procedures or analysis pipeline.

I've grouped together these three criteria because my comments relate to all three. I have little to

say about the analysis pipeline, but with regard to the methods, more information needs to be provided to assess its soundness and considerably more information needs to be provided to allow future attempts at replication.

Here's a list:

Why was Web of Science chosen? Do the authors believe the same results would be obtained if Google Scholar (a more egalitarian indexing program) or another indexing program is used for participant selection?

How were "psychology articles" operationalized (i.e., "we used the population of corresponding authors from psychology articles")? For example, was it required that the journal name contain the word "psychology"? If not, what were the criteria?

Which authors were recruited? All authors on each article or only the corresponding author?

In the pilot study, what did the "contact email" message say and who sent it? Similarly, what did the invitation email message say and who sent it? (I ask about who sent it because my guess is that participants who are more open to preregistration would be more likely to respond to both messages if they came from a researcher publicly known for advocating preregistration.)

In the proposed study, what will the invitation email say and who will send it? (I looked on OSF and could find only the six scenarios, no other relevant material.)

In the pilot study, were participants compensated? In the proposed study, will the participants be compensated? If so, what will be the compensation?

In the pilot study, how were the different versions of the scenario assigned to participants? (In the proposed study, the manuscript explains the randomization scheme will be conducted by Qualtrics.)

When is the Trust Question asked? The manuscript implies that the trust question is asked after participants read the scenario: "In the first screen, participants are presented with the instruction message, and can click through using a 'next' button to the scenario screen, after which they are presented with Question 1." (Question 1 is the Trust Question.)

However, the scenario PDFs posted on OSF suggest that the question is asked before participants read the scenario. The first sentence on each scenario PDF says the following: "We aim to measure your direct or 'primary' response to the scenario on the next page. To that end, please attempt to answer the question given without thinking at length about it. You may use any of the information given on the next page to answer the question.") The authors' use of the definite-article expression "the question" implies that that "the question" has already been asked.

If the Trust Question is not asked until after participants read the scenario, then the instructions that appear on the scenario PDFs are a bit confusing and need to be cleaned up.

Regarding the new manipulation check questions, I'm confused why they will be asked only for the scenarios that lead to a "yes" answer. For example, as I understand the authors, only for the familiar condition will participants be asked, "In the fictional study you just read, was the researcher responsible for the study someone with whom you were familiar?" And only in the preregistered or registered reports condition will participants be asked, "whether the study had been preregistered in some way, or whether it was the subject of a registered report."

Why not employ a manipulation check for all conditions? In other words, why not ask the same (or similar) manipulation check question for all scenarios? I also recommend adding in a couple of other neutral questions to better occlude the nature of the manipulation check question.

Returning to the scenario PDFs, it was deeply unfortunate that the only way I could see the manipulation was to (as I whined about above) click over to OSF, download six different files, and then compare each of the six files one to one another. That's asking a lot of reviewers, particularly for information that a) needs to be provided in the manuscript, and b) is simple to provide in the manuscript!

But, from what I could discern, the Familiar versus Unfamiliar conditions differ in that the Familiar scenarios have the sentence, "A researcher with whom you have collaborated previously (and with whom you would collaborate again) has recently conducted a study" whereas the Unfamiliar scenarios have the sentence I wrote about above, "A researcher unknown to you has recently conducted a study."

And, from what I could discern, the Registration conditions differ in that the None scenarios include the sentence, "The paper makes no mention of any previously documented sampling plan or study design," the Preregistered scenarios include the sentence, "Prior to conducting this study, a detailed sampling plan, analysis strategy, and the study hypotheses were posted publicly on the author's website," and the Registered Report scenarios include the sentence, "Prior to conducting the study, its protocol was peer-reviewed and conditionally accepted by the editorial committee at the publishing journal."

If all the above is true, this information must be stated in the manuscript.

Two more concerns, both relatively serious: First, both the pilot study and the proposed study are lacking demographic information about the participants. Given the 12.5% response rate, we really need to know who chose to participate in this type of study. As I alluded to above, different samples could provide different responses (e.g., samples of researchers who are enthusiastic about preregistration are most likely to provide responses that indicate that preregistration boosts trust).

Second, there is no justification provided for designing the study as between- rather than within-subjects. Because of carryover effects and subject demand (fatigue), a between-subjects might be the best choice, but no justification is provided.

6. Whether the authors have considered sufficient outcome-neutral conditions (e.g. absence of floor or ceiling effects; positive controls; other quality checks) for ensuring that the results obtained are able to test the stated hypotheses.

The addition of manipulation checks and some quality control checks is a good move.

One last and small quibble. The manuscript states: "the merits of PR and RR are advocated by a group of scientists often referred to as 'meta-scientists'." I know numerous scientists who advocate PR and RR, myself included, and none of us refer to ourselves as 'meta-scientists.' We're just scientists. In other words, it's not just "meta-scientists" who advocate pre-registration.

Reviewer: 2

Comments to the Author(s)

Comments on points specified by the editor, as well as some additional comments, are all stated in the attachment.

Reviewer: 3

Comments to the Author(s)

Review for RSOS-180599: "The Effect of Preregistration on Trust in Empirical Research Findings: A Registered Report Proposal"

Overall, I think that this RR proposal is a valuable study, that promises to shed more light onto the (psychological) effects of preregistration. As the field is moving forward in implementing open science practices, an evaluation of their effectiveness and side-effects is an important step, to which this papers adds. In general I support an IPA of this proposal, but have some issues I'd like to see addressed before:

1. More precise hypotheses

In the research questions you suddenly introduce "objective" trust. What is that? Please define. Furthermore, you write: "We expect that preregistration does increase the trust participants have in research results, and that familiarity increases trust in conjunction with preregistration." What does "in conjunction" mean? Is that a hint for an interaction effect? If yes, in what direction?

2. Description of results of pilot study

You write on p. 5: "Incorporating familiarity appears to led to the expected behavior of the dependent variable- familiar researchers led to a slight increase participant trust, over and above what alone was explained by PR and RR." This description seems misleading, as the BF does not support this claim.

If I read the BF tables correctly, there is no evidence for the inclusion of familiarity (in contrast, there is moderate evidence for not including it). Hence, there is no support for your hypothesis that "familiarity increases trust", and this conclusion should be explicitly written down.

For readers unacquainted with BF, please verbally interpret the BF for the interaction term.

Table 2 seems misplaced at its current position - why not report that as an exploratory result of the pilot study?

3. Proposed study

I guess you want to generalize the results to a variety of preregistered studies - this would actually call for a design where the study content (in the experimental material) is a random factor. I realize that this could lead to power problems, but I think at least a discussion of the issue is warranted.

Is the reported power analysis on p. 7 a frequentist power analysis? Seems quite odd in the context of a purely Bayesian data analysis. Why not report a Bayes factor design analysis (see Schönbrodt & Wagenmakers, 2017) for a fixed-n design, i.e., $\text{prob}(\text{BF} > 10 \mid \text{assumed effect size})$.

I think one underappreciated factor of the replication crisis is the low reliability of our measurements. Many social psychology studies rightly have been criticized for measuring their central variables with ad-hoc single-item measures. I would urge the authors to think about how to increase the reliability of the central DV measure in this study as well; from a psychometric point of view using a single item for a multifaceted construct seems careless.

On p. 9 you write: "In the absence of data, we assume a priori that each of the five models in our two- factor design is equally likely ($p = .20$)." Well, you do have data from the pilot study. Why not updating your priors from pilot  main study?

4. Minor comments:

- p. 3: "Researchers' work is taken more seriously and its academic content trusted more by others when PR or RR have been part of the research process.": Here (and throughout this paragraph) is either a citation needed, or it should be made clearer that this is a hypothesis and not an empirical

finding. Do we already have evidence for that claim, or is that what we expect? Isn't the current project just about finding that out?

- In the pilot study, and also the main study: Did/will you explain the difference of PR and RR to participants? In my experience, this is not clear to many.

Here are the journal's check points:

The significance of the research question(s): Relevant.

The logic, rationale, and plausibility of the proposed hypotheses: Makes sense.

The soundness and feasibility of the methodology and analysis pipeline (including statistical power analysis where applicable): Please improve power analysis (see above); anything else is state of the art.

Whether the clarity and degree of methodological detail would be sufficient to replicate exactly the proposed experimental procedures and analysis pipeline: Sufficient.

Whether the authors provide a sufficiently clear and detailed description of the methods to prevent undisclosed flexibility in the experimental procedures or analysis pipeline: Sufficiently clear.

Whether the authors have considered sufficient outcome-neutral conditions (e.g. positive controls) for ensuring that the results obtained are able to test the stated hypotheses: They added a manipulation check.

Reviewer: 4

Comments to the Author(s)

The current project proposes to examine the effect of study pre-registration (vs. RR vs. control) and collaborator familiarity on trust.

Major concerns:

1. Conceptualization of trust: You propose to add explanation text to the study defining trust (which was not contained in the pilot study): "the word 'trust' in this question can be interpreted in many ways: 'reliable,' 'replicable,' 'a true effect,' 'valid,' or 'high-quality.'" (p. 8). This text was helpful for me, because I do not feel that the construct of trust was fully explained in the introduction. I personally had a strong (negative) reaction to the word "trust." Trust implies that checking someone's work is not needed, but the benefit of pre-registration is that it enables you to check someone's work.

All of these open science initiatives that we have proposed and enacted are designed to increase research/evidence quality. It doesn't matter to me whether we can "trust" a finding more. Trust should be an outcome of quality, but it is not the same thing as quality. A pre-registered or RR finding is more credible than an unregistered study, because we can verify its claims more fully. That is, transparency in detailing the data generating process makes it possible to verify a claim. RR studies may be de facto higher quality because of improvements introduced in the two-stage review process (but again, quality is the desired outcome; trust should be irrelevant). Let me try to be as clear as I can: pre-registered studies should not be trusted more, but on average, the data contained in them are more valuable, because it is possible to verify the claims the authors make. I wouldn't say that value is the same thing as trust, either.

I will add that it is a separate question about whether researchers *do* trust pre-registered or RR

studies more than traditional studies (versus whether they *should*), but it seems weird to me to couch higher trust as the desired outcome. We should care about the quality of evidence, not about trust. For instance, you write "Rebuilding trust in our research findings is vital" (p. 3), but I would rephrase this to "Increasing the quality of our evidence base is vital." Similarly, "several new initiatives have been launched which focus on restoring the trust of the scientific community in its research" (p. 2). But this seems to me to be a mischaracterization, and it makes it sound like we are just doing reputation management rather than trying to improve research quality.

2. Measurement of trust: Unless I missed it, I do not see the full text of the question asking about trust in the text or the supplemental materials. It should be provided, and you might consider putting your Qualtrics (.qsf) file on OSF too. I reviewed the materials on OSF and did not see anything like this there. A single item ad hoc measure of the key dependent variable seems to leave a lot to be desired.

3. You write "It is unclear whether the scientific community at large trusts the results borne of the PR or RR process more than studies which have not been subject to these processes" (p. 3). I think this is the question your study is designed to answer. It may be interesting to know this, but perhaps not for the reasons you outline in the introduction.

Additional point:

4. When I was reviewing the materials on OSF, I noticed that you state the following in your ethics application: "As aforementioned, participants will be naive to the exact objectives of the study, and will be only exposed to the stimulus of 1 (of 6) experimental conditions. Naivety is important for this study as social desirability may be expected to play a role in the participants' responses. We want to ensure that the participants' responses are not impacted by what they might think is expected of them." This seems to be inconsistent with the "participant information form" which gives away many of the details of the study including the focal variables. Perhaps I misunderstood and maybe this form is a debriefing form rather than a consent form, but I would urge you to reconsider your framing in the participant information form to try to preserve participant naivety.

Signed,
Katie Corker

Author's Response to Decision Letter for (RSOS-180599.R0)

See Appendix B.

RSOS-181351.R0

Review form: Reviewer 1

Is the language acceptable?

Yes

Do you have any ethical concerns with this paper?

No

Have you any concerns about statistical analyses in this paper?

No

Recommendation?

Accept with minor revision

Comments to the Author(s)

Kudos to the authors for being responsive to most of the concerns raised by the reviewers. I think the proposed study is much stronger and better fulfills the requirements of a registered report.

However, I have two remaining concerns, one major and one minor.

Major concern: I still don't have a sense of whether the authors are predicting that familiarity will be an additive or interactive factor – and why.

In response to my previous concern, the authors write (in their response to reviewers):

“Our group had discussed that familiarity should be included because it is a very strong influence on trust by virtue of the recognition heuristic mentioned in the manuscript, however there is an extent to which the expectation of how it would behave in conjunction with the preregistration variable was unknown. At the outset, we believed it should be additive, however recognize that in cases where a personal interaction leading to familiarity is negative (e.g., in the case where one co-author suspects another of poor research practice), then the effect of familiarity would behave differently.”

Alas, that response doesn't answer the question of whether the authors predict that manipulating the familiarity variable will add to or interact with the other variable the authors propose to manipulate (i.e., preregistration status).

Similarly, in response to Reviewer 3, the authors write (in their response to reviewers):

“Yes, indeed this is a hint for an interaction effect. We expected that familiarity would have an additive effect on trust that is, it would increase trust ratings in conjunction with preregistration status even more than preregistration status alone would cause. We see that our descriptions of the interaction and the variable in general are not sufficiently clear, and have amended the manuscript to fix this issue in several different places in pages 5/6 and 16.”

Again, this response doesn't really answer the question and seems to confuse interaction with additivity (“yes, indeed this is a hint for an interaction effect” but then “we expected that familiarity would have an additive effect on trust”).

Similarly, in the manuscript, the authors write, “The results in Figure 1 reveal that familiarity does not increase trust means for unregistered studies, but does increase trust means for PR and RR studies.” Thus, the authors are describing an interactive effect, not an additive effect.

If the authors are considering familiarity a predicted rather than an exploratory variable, then they need to predict whether the effect of this variable will be additive or interactive with the effect of the other predicted variable. Just to be clear, I'm using the terms additive and interactive in the same way that classic experimental psychology does (e.g., http://www.psychwiki.com/wiki/What_is_an_Interaction%3F), because the design of the proposed study (a 3 x 2 factorial) appears to be a classic experimental psychology design.

The minor concern: I think the authors might have misunderstood why I recommended that they better occlude the manipulation check question (with additional questions). The authors seem to believe better occlusion isn't necessary because “the manipulation checks are done at the end of the experiment.” Yes, indeed; that's usually when manipulation checks are conducted. But, alas,

that's not the sole reason for occluding manipulation checks. Another reason is to occlude the purpose of the study until all data have been collected. Participants talk to other participants – often before all participants have participated. This type of participant-to-participant contamination used to be a huge problem with undergraduate participant pools. But it can also be a problem with current day social media and even old-fashioned water-cooler talk.

Review form: Reviewer 2

Is the language acceptable?

Yes

Do you have any ethical concerns with this paper?

No

Have you any concerns about statistical analyses in this paper?

No

Recommendation?

Accept with minor revision

Comments to the Author(s)

Most of my concerns have been adequately addressed, and my outstanding concerns should be relatively easy for the authors to address/implement. I therefore recommend the submission be accepted after some minor revisions. See attached document for outstanding concerns (Appendix C).

Review form: Reviewer 4

Is the language acceptable?

Yes

Do you have any ethical concerns with this paper?

No

Have you any concerns about statistical analyses in this paper?

No

Recommendation?

Accept with minor revision

Comments to the Author(s)

In my previous review, I raised three major concerns:

1. I requested that the introduction be edited to explain the construct of trust in more detail. In particular, I was concerned that it was unclear that perceptions of trust, rather than actual trustworthiness or research quality, was being assessed in this study. In the revised version, it is now clearer what the authors examine. I found footnote 1 particularly helpful.
2. I asked for more details about the study's methods, and I was happy to see that materials have now been shared on OSF. I looked through the Qualtrics preview and a few of the materials, and it appears things are now clear and well organized on OSF.

Regarding the measurement of trust, I believe the authors may have misunderstood my critique. My critique was not that a single item measure of trust might be missing important facets, but rather that a single item ad hoc measure has unknown reliability and validity. A benefit of keeping the item the same as the pilot study is that you can compare to that study. But you still may want to consider adding a few more additional items. You could ask separate questions about trust, reliability, replicability, etc. Presumably, if those items hang together, you have some evidence for internal consistency of your trust construct. If you were being really thorough, you could separately validate your trust measurement in another study. This is not a “must” for me, but something for you to consider. You could (a) keep things as they are with a single item measure, (b) collect your desired single item, but add additional questions to further probe and understand your construct (this part could be exploratory; you could stick with your single item as your focal DV for your pre-reg), or (c) validate the single item measure in a separate study before doing choice a. The paper will still be interesting if you decide not to do (b) or (c), but we could certainly learn more if you had time and energy for either of those options.

3. I asked you to better separate the question of whether researchers do trust pre-registered (or RR) studies more than regular studies from whether they should. As noted above in point 1, the intro does a better job at this now. Yet, I still found some spots where you appeared to argue that open science advocates think trust should be higher (should in the moral sense, not in terms of hypotheses). For instance, you write “it is the opinion of proponents of PR and RR that researchers’ academic work will be trusted more by other academics when PR or RR have been part of the research process” As evidence, you cite a 2013 joint letter, but that letter doesn’t talk about trust at all (in spite of the Guardian’s headline). I am uncertain if there are many who would argue that higher trust, in and of itself, is the goal. As we discussed before and agree on, the goal is higher quality work. If further revisions are required, I would ask that you continue to pay attention to the distinction between trust and quality (these are separable goals and some OS proponents might not have higher trust as a goal).

Additional notes:

4. I am glad that you removed references to “high-impact” journals that were confounding manipulations.

5. You say that “proponents argue that researchers will increase their chances of getting articles accepted by journals, regardless of whether or not the results obtained favor their hypotheses.” Ideally, this would be true for PR, but I fear that it is not, and is instead an exclusive benefit of RR. I’m not sure I hear people arguing that PR will increase the chance of acceptance of null results (and there are anecdotes of rejections of PR’d nulls; come to think of it, I have personally published a replication that was done PR, not RR. It was still rejected at several journals, because results were null, before we went to an OS friendly outlet).

6. Maybe I miss some nuance to the way Bayesian analyses should be described, but is it really correct to say one model is “more likely to be true” (p. 9) than another model? Aren’t these always relative comparisons – i.e., one model is more likely than another, but the likelihood of either model relative to “truth” is unknown/unknowable? I hesitate to even raise the critique because I am a novice at best when it comes to Bayes. Something about “has a true effect” (p. 10) and the aforementioned phrase felt off to me.

Decision letter (RSOS-181351.R0)

18-Sep-2018

Dear Ms Field

On behalf of the Editors, I am pleased to inform you that your Manuscript RSOS-181351 entitled "The Effect of Preregistration on Trust in Empirical Research Findings: A Registered Report Proposal" has been accepted in principle for publication in Royal Society Open Science subject to minor revision in accordance with the referee and editor suggestions. Please find their comments at the end of this email.

The reviewers and handling editors have recommended publication, but also suggest some minor revisions to your manuscript. Therefore, I invite you to respond to the comments and revise your manuscript.

Full author guidelines can be found here
<http://rsos.royalsocietypublishing.org/content/registered-reports>.

Please note that Royal Society Open Science charge article processing charges for all new submissions that are accepted for publication. Charges will also apply to papers transferred to Royal Society Open Science from other Royal Society Publishing journals, as well as papers submitted as part of our collaboration with the Royal Society of Chemistry (<http://rsos.royalsocietypublishing.org/chemistry>). If your manuscript is newly submitted and subsequently accepted for publication, you will be asked to pay the article processing charge, unless you request a waiver and this is approved by Royal Society Publishing. You can find out more about the charges at <http://rsos.royalsocietypublishing.org/page/charges>. Should you have any queries, please contact openscience@royalsociety.org.

on behalf of Chris Chambers (Subject Editor, Royal Society Open Science)
openscience@royalsociety.org

Associate Editor Comments to Author (Professor Chris Chambers):

Associate Editor: 1

Comments to the Author:

The revised manuscript was returned to the three of the original four reviewers (one reviewer was unavailable but will return at Stage 2 in the event of the manuscript achieving IPA). All reviewers viewed the revision favourably but highlight a number of areas that would benefit from further improvement, chiefly with the methodology but also regarding structure (e.g. Reviewer 2's suggestion to present the hypotheses in list format, which I endorse), clarity of hypotheses (e.g. Reviewer 1's query regarding the nature of the predicted interaction) and conceptual framing (e.g. Reviewer 4's concern about the issue of trust vs quality). Provided all remaining points are addressed comprehensively in a revised submission, Stage 1 IPA should be forthcoming without requiring further in-depth review.

Reviewer comments to Author:

Reviewer: 2

Comments to the Author(s)

Most of my concerns have been adequately addressed, and my outstanding concerns should be relatively easy for the authors to address/implement. I therefore recommend the submission be accepted after some minor revisions. See attached document for outstanding concerns.

Reviewer: 1

Comments to the Author(s)

Kudos to the authors for being responsive to most of the concerns raised by the reviewers. I think the proposed study is much stronger and better fulfills the requirements of a registered report.

However, I have two remaining concerns, one major and one minor.

Major concern: I still don't have a sense of whether the authors are predicting that familiarity will be an additive or interactive factor – and why.

In response to my previous concern, the authors write (in their response to reviewers):

“Our group had discussed that familiarity should be included because it is a very strong influence on trust by virtue of the recognition heuristic mentioned in the manuscript, however there is an extent to which the expectation of how it would behave in conjunction with the preregistration variable was unknown. At the outset, we believed it should be additive, however recognize that in cases where a personal interaction leading to familiarity is negative (e.g., in the case where one co-author suspects another of poor research practice), then the effect of familiarity would behave differently.”

Alas, that response doesn't answer the question of whether the authors predict that manipulating the familiarity variable will add to or interact with the other variable the authors propose to manipulate (i.e., preregistration status).

Similarly, in response to Reviewer 3, the authors write (in their response to reviewers):

“Yes, indeed this is a hint for an interaction effect. We expected that familiarity would have an additive effect on trust that is, it would increase trust ratings in conjunction with preregistration status even more than preregistration status alone would cause. We see that our descriptions of the interaction and the variable in general are not sufficiently clear, and have amended the manuscript to fix this issue in several different places in pages 5/6 and 16.”

Again, this response doesn't really answer the question and seems to confuse interaction with

additivity (“yes, indeed this is a hint for an interaction effect” but then “we expected that familiarity would have an additive effect on trust”).

Similarly, in the manuscript, the authors write, “The results in Figure 1 reveal that familiarity does not increase trust means for unregistered studies, but does increase trust means for PR and RR studies.” Thus, the authors are describing an interactive effect, not an additive effect.

If the authors are considering familiarity a predicted rather than an exploratory variable, then they need to predict whether the effect of this variable will be additive or interactive with the effect of the other predicted variable. Just to be clear, I’m using the terms additive and interactive in the same way that classic experimental psychology does (e.g., http://www.psychwiki.com/wiki/What_is_an_Interaction%3F), because the design of the proposed study (a 3 x 2 factorial) appears to be a classic experimental psychology design.

The minor concern: I think the authors might have misunderstood why I recommended that they better occlude the manipulation check question (with additional questions). The authors seem to believe better occlusion isn’t necessary because “the manipulation checks are done at the end of the experiment.” Yes, indeed; that’s usually when manipulation checks are conducted. But, alas, that’s not the sole reason for occluding manipulation checks. Another reason is to occlude the purpose of the study until all data have been collected. Participants talk to other participants – often before all participants have participated. This type of participant-to-participant contamination used to be a huge problem with undergraduate participant pools. But it can also be a problem with current day social media and even old-fashioned water-cooler talk.

Reviewer: 4

Comments to the Author(s)

In my previous review, I raised three major concerns:

1. I requested that the introduction be edited to explain the construct of trust in more detail. In particular, I was concerned that it was unclear that perceptions of trust, rather than actual trustworthiness or research quality, was being assessed in this study. In the revised version, it is now clearer what the authors examine. I found footnote 1 particularly helpful.
2. I asked for more details about the study’s methods, and I was happy to see that materials have now been shared on OSF. I looked through the Qualtrics preview and a few of the materials, and it appears things are now clear and well organized on OSF.

Regarding the measurement of trust, I believe the authors may have misunderstood my critique. My critique was not that a single item measure of trust might be missing important facets, but rather that a single item ad hoc measure has unknown reliability and validity. A benefit of keeping the item the same as the pilot study is that you can compare to that study. But you still may want to consider adding a few more additional items. You could ask separate questions about trust, reliability, replicability, etc. Presumably, if those items hang together, you have some evidence for internal consistency of your trust construct. If you were being really thorough, you could separately validate your trust measurement in another study. This is not a “must” for me, but something for you to consider. You could (a) keep things as they are with a single item measure, (b) collect your desired single item, but add additional questions to further probe and understand your construct (this part could be exploratory; you could stick with your single item as your focal DV for your pre-reg), or (c) validate the single item measure in a separate study before doing choice a. The paper will still be interesting if you decide not to do (b) or (c), but we could certainly learn more if you had time and energy for either of those options.

3. I asked you to better separate the question of whether researchers do trust pre-registered (or RR) studies more than regular studies from whether they should. As noted above in point 1, the

intro does a better job at this now. Yet, I still found some spots where you appeared to argue that open science advocates think trust should be higher (should in the moral sense, not in terms of hypotheses). For instance, you write “it is the opinion of proponents of PR and RR that researchers’ academic work will be trusted more by other academics when PR or RR have been part of the research process” As evidence, you cite a 2013 joint letter, but that letter doesn’t talk about trust at all (in spite of the Guardian’s headline). I am uncertain if there are many who would argue that higher trust, in and of itself, is the goal. As we discussed before and agree on, the goal is higher quality work. If further revisions are required, I would ask that you continue to pay attention to the distinction between trust and quality (these are separable goals and some OS proponents might not have higher trust as a goal).

Additional notes:

4. I am glad that you removed references to “high-impact” journals that were confounding manipulations.

5. You say that “proponents argue that researchers will increase their chances of getting articles accepted by journals, regardless of whether or not the results obtained favor their hypotheses.” Ideally, this would be true for PR, but I fear that it is not, and is instead an exclusive benefit of RR. I’m not sure I hear people arguing that PR will increase the chance of acceptance of null results (and there are anecdotes of rejections of PR’d nulls; come to think of it, I have personally published a replication that was done PR, not RR. It was still rejected at several journals, because results were null, before we went to an OS friendly outlet).

6. Maybe I miss some nuance to the way Bayesian analyses should be described, but is it really correct to say one model is “more likely to be true” (p. 9) than another model? Aren’t these always relative comparisons – i.e., one model is more likely than another, but the likelihood of either model relative to “truth” is unknown/unknowable? I hesitate to even raise the critique because I am a novice at best when it comes to Bayes. Something about “has a true effect” (p. 10) and the aforementioned phrase felt off to me.

Author's Response to Decision Letter for (RSOS-181351.R0)

See Appendix D.

Decision letter (RSOS-181351.R1)

18-Oct-2018

Dear Ms Field

On behalf of the Editor, I am pleased to inform you that your Manuscript RSOS-181351.R1 entitled "The Effect of Preregistration on Trust in Empirical Research Findings: A Registered Report Proposal" has been accepted in principle for publication in Royal Society Open Science.

You may now progress to Stage 2 and complete the study as approved. Before commencing data collection we ask that you:

1) Update the journal office as to the anticipated completion date of your study.

2) Register your approved protocol on the Open Science Framework (<https://osf.io/>) or other recognised repository, either publicly or privately under embargo until submission of the Stage 2 manuscript. Please note that a time-stamped, independent registration of the protocol is mandatory under journal policy, and manuscripts that do not conform to this requirement cannot be considered at Stage 2. The protocol should be registered unchanged from its current approved state, with the time-stamp preceding implementation of the approved study design.

Following completion of your study, we invite you to resubmit your paper for peer review as a Stage 2 Registered Report. Please note that your manuscript can still be rejected for publication at Stage 2 if the Editors consider any of the following conditions to be met:

- The results were unable to test the authors' proposed hypotheses by failing to meet the approved outcome-neutral criteria.
- The authors altered the Introduction, rationale, or hypotheses, as approved in the Stage 1 submission.
- The authors failed to adhere closely to the registered experimental procedures. Please note that any deviations from the approved experimental procedures must be communicated to the editor immediately for approval, and prior to the completion of data collection. Failure to do so can result in revocation of in-principle acceptance and rejection at Stage 2 (see complete guidelines for further information).
- Any post-hoc (unregistered) analyses were either unjustified, insufficiently caveated, or overly dominant in shaping the authors' conclusions.
- The authors' conclusions were not justified given the data obtained.

We encourage you to read the complete guidelines for authors concerning Stage 2 submissions at <http://rsos.royalsocietypublishing.org/content/registered-reports>. Please especially note the requirements for data sharing, reporting the URL of the independently registered protocol, and that withdrawing your manuscript will result in publication of a Withdrawn Registration.

Please note that Royal Society Open Science will introduce article processing charges for all new submissions received from 1 January 2018. Registered Reports submitted and accepted after this date will ONLY be subject to a charge if they subsequently progress to and are accepted as Stage 2 Registered Reports. If your manuscript is submitted and accepted for publication after 1 January 2018 (i.e. as a full Stage 2 Registered Report), you will be asked to pay the article processing charge, unless you request a waiver and this is approved by Royal Society Publishing. You can find out more about the charges at <http://rsos.royalsocietypublishing.org/page/charges>. Should you have any queries, please contact openscience@royalsociety.org.

Once again, thank you for submitting your manuscript to Royal Society Open Science and we look forward to receiving your Stage 2 submission. If you have any questions at all, please do not hesitate to get in touch. We look forward to hearing from you shortly with the anticipated submission date for your stage two manuscript.

Kind regards,

Royal Society Open Science Editorial Office
Royal Society Open Science
openscience@royalsociety.org

on behalf of Professor Chris Chambers (Registered Reports Editor, Royal Society Open Science)
openscience@royalsociety.org

Author's Response to Decision Letter for (RSOS-181351.R1)

See Appendix E.

RSOS-181351.R2 (Revision)

Review form: Reviewer 1

Is the manuscript scientifically sound in its present form?

No

Are the interpretations and conclusions justified by the results?

No

Is the language acceptable?

Yes

Do you have any ethical concerns with this paper?

Yes

Have you any concerns about statistical analyses in this paper?

Yes

Recommendation?

Major revision

Comments to the Author(s)

I applaud the authors for assembling their final report when the data are so confusing. The manuscript is an emblem of bravery.

However, I'm quite concerned that so many participants failed the manipulation checks. I think it would be extremely helpful to ensure that

- a) the conditions were correctly assigned to each participant;
- b) the manipulation check questions were correctly assigned to each participant;
- c) Qualtrics randomizer was working as desired;
- d) the data codes were interpreted correctly; and
- e) the manipulation check questions were interpreted correctly.

Therefore, first, I recommend that the authors return to Qualtrics and take precise screenshots of each and every screen of material that was shown to participants in each of the conditions. Examine those screenshots to ensure that the study proceeded as the authors envisioned it would and that the Qualtrics randomizer worked the way the authors envisioned it would (if not, that could easily explain why nearly 90% of the participants in one condition failed the manipulation check -- they received the wrong condition).

Moreover, such screenshots from Qualtrics should be available in the supplementary materials. They are best practice for research transparency.

Second, I recommend that the authors download their data from Qualtrics both as "numeric values" and as "choice text" and compare the two downloads. Although this might seem pedantic,

given the high profile recent instances in which data codes were erroneously interpreted (e.g., a JAMA article in which conditions codes were reversed 1 versus 2), this second check is important.

Third, I encourage the authors to go through the Qualtrics study numerous times, noting each time what condition they appear to be receiving (from Qualtrics) and then make sure that condition is correctly coded in Qualtrics. Again, this might seem pedantic, but it is important.

Fourth, I recommend that the authors ask 10 other native speakers of English to read the manipulation check questions and report their interpretation of them. I found the questions to be worded complexly; therefore, one problem might be that the manipulation check questions were difficult to understand.

The bottom line: I encourage the authors to painstakingly examine every aspect of the study to ensure that there were no undetected (and of course unintentional) mixups. Otherwise, it is hard to explain a near 90% exclusion rate for this type of study -- particularly given that the participants are other researchers.

Also, I noticed that participants' email addresses are included, along with their responses, in one of the files uploaded on OSF. However, according to the ethics consent, "The research results of this study will be treated confidentially and anonymously. Your data will be processed by means of a participant number. This code is disconnected from your personal data." Therefore, participants' email addresses should most likely be removed.

Review form: Reviewer 2

Is the manuscript scientifically sound in its present form?

No

Are the interpretations and conclusions justified by the results?

Yes

Is the language acceptable?

No

Do you have any ethical concerns with this paper?

No

Have you any concerns about statistical analyses in this paper?

Yes

Recommendation?

Major revision

Comments to the Author(s)

I noted a few issues that seemed quite important to me to address before the paper is accepted. In reality, they may not constitute more than minor revisions however. See comments in attached Word document for details (Appendix F).

Review form: Reviewer 4

Is the manuscript scientifically sound in its present form?

Yes

Are the interpretations and conclusions justified by the results?

Yes

Is the language acceptable?

Yes

Do you have any ethical concerns with this paper?

No

Have you any concerns about statistical analyses in this paper?

No

Recommendation?

Major revision

Comments to the Author(s)

The current paper reports the results of an in principle accepted manuscript concerning the effects of pre-registration and familiarity on a single item measuring trust in a paper's findings. The research team had a low response rate to their invitation to participate in the study (6%), and they consequently had to modify their sampling strategy. Additionally, many participants did not pass the manipulation checks that the authors planned to use. The final sample was much smaller than the goal (N = 209, planned N = 480). To complete this review, I read the current version of the manuscript alongside my comments on the proposal draft. I also reviewed the files posted on OSF.

1. As a first note, I am unable to locate a copy of the Qualtrics questionnaire for the main study (.qsf or .docx). I can find it for the pilot study but not for the main study. I wanted to check the coding for the manipulation check questions to make sure that they were executed as planned. The high failure rate for the manipulation check questions is really striking.
2. The data file that is posted has participant IP addresses and email addresses. I imagine these should be removed. There is also no codebook (that I could find) to interpret the data file labels.
3. I think the manuscript should analyze and present results from all available cases (i.e., before exclusions). These should of course be labeled as exploratory follow up, but given how much data is being discarded in the main analysis, I think it would be useful for readers to be able to consider the results without these exclusions. These results for the full sample are of course hard to interpret, but they are an important part of what we have.
4. I previously worried about the validity/reliability of a single item ad hoc measure of trust. Now that the results are in, I find myself wishing we had more data to consider about this outcome. There are many open questions that the current results leave unsettled, but even if the full intended sample had been collected, there would still be unresolved issues related to what exactly is being measured as the DV here.
5. I would be supportive of a more detailed analysis of the qualitative/open-ended data to help better understand the failed manipulation.
6. The current study varied in a number of ways from the pilot study (many at our direction). Comparisons to the pilot study should be very cautious and consider these differences. For

instance, the pilot study manipulation of RR was confounded with a reference to journal prestige/impact. That confound was (appropriately) removed here. Hence, the results are not directly comparable. (Note: Another way in which the current study differed from the pilot study was that explanatory text to clarify the meaning of trust was added. Perhaps this text altered the way in which participants responded to the trust question.)

7. The authors appear to have followed the pre-registered research plan and disclosed deviations from that plan.

Decision letter (RSOS-181351.R2)

18-Dec-2019

Dear Ms Field,

The editors assigned to your Stage 2 RR ("The Effect of Preregistration on Trust in Empirical Research Findings: Results of a Registered Report") has now received comments from reviewers. We would like you to revise your paper in accordance with the referee and Subject Editor suggestions which can be found below (not including confidential reports to the Editor). Please note this decision does not guarantee eventual acceptance.

Please submit a copy of your revised paper within 6 weeks (i.e. by the 09-Jan-2020). If deemed necessary by the Editors, your manuscript will be sent back to one or more of the original reviewers for assessment.

- Data accessibility

It is a condition of publication that all supporting data are made available either as supplementary information or preferably in a suitable permanent repository. The data accessibility section should state where the article's supporting data can be accessed. This section should also include details, where possible of where to access other relevant research materials such as statistical tools, protocols, software etc can be accessed. If the data has been deposited in an external repository this section should list the database, accession number and link to the DOI

for all data from the article that has been made publicly available. Data sets that have been deposited in an external repository and have a DOI should also be appropriately cited in the manuscript and included in the reference list.

<http://datadryad.org/submit?journalID=RSOS&manu=RSOS-181351.R2>

- Competing interests

- Authors' contributions

- Acknowledgements

- Funding statement

Kind regards,

Anita Kristiansen

Editorial Coordinator

on behalf of Chris Chambers

Subject Editor, Royal Society Open Science

Associate Editor's comments (Professor Chris Chambers):

Associate Editor: 1

Comments to the Author:

Three of the four original reviewers who assessed the Stage 1 manuscript returned to review the Stage 2 submission. The reviews overall indicate a number of concerns that will need to be addressed to achieve full acceptance. All reviewers note with the concern the failure of the

manipulation check, with Reviewer 1 outlining a helpful point-by-point list of actions to check that the study was in fact administered correctly (see also Reviewer 4 point 1), and with Reviewers 2 and 4 suggesting additional exploratory analyses. The RR policy does not require authors to conduct additional analyses, but especially given the failure of the manipulation checks, these suggestions are sensible and very well advised. The reviewers also note a potential breach of confidentiality in the OSF data that should be corrected, as well as a range of other revisions to improve structure, clarity and comprehensiveness, both of the manuscript and the materials stored on the OSF.

Given the concerns with manipulation success, Reviewer 2 makes an additional point: "I think the RSOS editorial team should take note of this issue and consider whether more flexible RR format could be offered in cases where follow-up pilot studies are warranted after changes to the experimental design at stage 1 (e.g. the authors could be offered to submit a stage 2 [second pilot] and a stage 3 [main study] version of the report)." This point is well made, and should the authors prefer to do so, I am open to them conducting a modified follow-up study as part of an Incremental Registration, and then including both the original study and the follow up study in the current RR. If the authors wish to do so, please contact the journal office and we will take the necessary administrative steps to return the manuscript to Stage 1. However, I am not going to require this, as to do so would undermine the commitment of the journal as part of Stage 1 in-principle acceptance.

Comments to Author:

Reviewers' Comments to Author:

Reviewer: 1

Comments to the Author(s)

I applaud the authors for assembling their final report when the data are so confusing. The manuscript is an emblem of bravery.

However, I'm quite concerned that so many participants failed the manipulation checks. I think it would be extremely helpful to ensure that

- a) the conditions were correctly assigned to each participant;
- b) the manipulation check questions were correctly assigned to each participant;
- c) Qualtrics randomizer was working as desired;
- d) the data codes were interpreted correctly; and
- e) the manipulation check questions were interpreted correctly.

Therefore, first, I recommend that the authors return to Qualtrics and take precise screenshots of each and every screen of material that was shown to participants in each of the conditions. Examine those screenshots to ensure that the study proceeded as the authors envisioned it would and that the Qualtrics randomizer worked the way the authors envisioned it would (if not, that could easily explain why nearly 90% of the participants in one condition failed the manipulation check -- they received the wrong condition).

Moreover, such screenshots from Qualtrics should be available in the supplementary materials. They are best practice for research transparency.

Second, I recommend that the authors download their data from Qualtrics both as "numeric values" and as "choice text" and compare the two downloads. Although this might seem pedantic, given the high profile recent instances in which data codes were erroneously interpreted (e.g., a JAMA article in which conditions codes were reversed 1 versus 2), this second check is important.

Third, I encourage the authors to go through the Qualtrics study numerous times, noting each

time what condition they appear to be receiving (from Qualtrics) and then make sure that condition is correctly coded in Qualtrics. Again, this might seem pedantic, but it is important.

Fourth, I recommend that the authors ask 10 other native speakers of English to read the manipulation check questions and report their interpretation of them. I found the questions to be worded complexly; therefore, one problem might be that the manipulation check questions were difficult to understand.

The bottom line: I encourage the authors to painstakingly examine every aspect of the study to ensure that there were no undetected (and of course unintentional) mixups. Otherwise, it is hard to explain a near 90% exclusion rate for this type of study -- particularly given that the participants are other researchers.

Also, I noticed that participants' email addresses are included, along with their responses, in one of the files uploaded on OSF. However, according to the ethics consent, "The research results of this study will be treated confidentially and anonymously. Your data will be processed by means of a participant number. This code is disconnected from your personal data." Therefore, participants' email addresses should most likely be removed.

Reviewer: 2

Comments to the Author(s)

I noted a few issues that seemed quite important to me to address before the paper is accepted. In reality, they may not constitute more than minor revisions however. See comments in attached Word document for details.

Reviewer: 4

Comments to the Author(s)

The current paper reports the results of an in principle accepted manuscript concerning the effects of pre-registration and familiarity on a single item measuring trust in a paper's findings. The research team had a low response rate to their invitation to participate in the study (6%), and they consequently had to modify their sampling strategy. Additionally, many participants did not pass the manipulation checks that the authors planned to use. The final sample was much smaller than the goal (N = 209, planned N = 480). To complete this review, I read the current version of the manuscript alongside my comments on the proposal draft. I also reviewed the files posted on OSF.

1. As a first note, I am unable to locate a copy of the Qualtrics questionnaire for the main study (.qsf or .docx). I can find it for the pilot study but not for the main study. I wanted to check the coding for the manipulation check questions to make sure that they were executed as planned. The high failure rate for the manipulation check questions is really striking.

2. The data file that is posted has participant IP addresses and email addresses. I imagine these should be removed. There is also no codebook (that I could find) to interpret the data file labels.

3. I think the manuscript should analyze and present results from all available cases (i.e., before exclusions). These should of course be labeled as exploratory follow up, but given how much data is being discarded in the main analysis, I think it would be useful for readers to be able to consider the results without these exclusions. These results for the full sample are of course hard to interpret, but they are an important part of what we have.

4. I previously worried about the validity/reliability of a single item ad hoc measure of trust.

Now that the results are in, I find myself wishing we had more data to consider about this outcome. There are many open questions that the current results leave unsettled, but even if the full intended sample had been collected, there would still be unresolved issues related to what exactly is being measured as the DV here.

5. I would be supportive of a more detailed analysis of the qualitative/open-ended data to help better understand the failed manipulation.

6. The current study varied in a number of ways from the pilot study (many at our direction). Comparisons to the pilot study should be very cautious and consider these differences. For instance, the pilot study manipulation of RR was confounded with a reference to journal prestige/impact. That confound was (appropriately) removed here. Hence, the results are not directly comparable. (Note: Another way in which the current study differed from the pilot study was that explanatory text to clarify the meaning of trust was added. Perhaps this text altered the way in which participants responded to the trust question.)

7. The authors appear to have followed the pre-registered research plan and disclosed deviations from that plan.

Author's Response to Decision Letter for (RSOS-181351.R2)

See Appendix G.

Decision letter (RSOS-181351.R3)

19-Feb-2020

Dear Ms Field,

It is a pleasure to accept your revised Stage 2 RR entitled "The Effect of Preregistration on Trust in Empirical Research Findings: Results of a Registered Report" in its current form for publication in Royal Society Open Science.

Kind regards,
Lianne Parkhouse

on behalf of Professor Chris Chambers (Subject Editor)
openscience@royalsociety.org

Appendix A

NB! This reviewer is not sufficiently experienced with Bayesian model selection and comparison to fully evaluate the soundness of the planned analyses.

Please comment explicitly on each of the following points in your comments to the authors

- ***The significance of the research question(s)***

The research question seems highly important to answer. The perceived trustworthiness of preregistration and registered reports are a crucial component of their usefulness, and the extent to which they are increasing perceived trustworthiness would be important information for funders considering whether to prioritize such research in grants, for journals considering such research for publication, and for individual researchers considering implementing the practice of preregistration in their own research.

- ***The logic, rationale, and plausibility of the proposed hypotheses***

The rationale for the expected correlation between Preregistration Status and Trustworthiness is clearly laid out and justified. The hypothesis concerning this correlation is also in my opinion highly plausible (both given the results of the pilot data, and my subjective prior).

The rationale for the moderating effect of familiarity is less clear to me. While the inclusion of the moderator in and of itself is rather straight-forward, I have the following concerns:

1. It is not obvious to me why the authors are only interested in familiarity as a moderator on the relationship between preregistration status and trustworthiness, and not as a main effect.
2. It is not obvious to me why familiarity is the only moderating variable considered. I do not expect the authors to lay out every possible moderator of interest of course, and I certainly see that there is a trade-off between including potential moderators and maintaining sufficient measurement precision. However, I would at least like to see a brief discussion of potential moderators, and a justification for why author familiarity could be considered the *most* important moderator.

- ***The soundness and feasibility of the methodology and analysis pipeline (including statistical power analysis where applicable)***

The proposed study is certainly feasible as currently planned, which is evidenced by results from the pilot study. However, I believe there are several major conceptual issues that needs to be resolved and/or clarified in the preregistration before data collection can begin. I list these below in order of importance.

1. When looking at the proposed scenarios, I think there might be important confounds in the current operationalization of the preregistration status variable:
 - a. The RR study condition is explicitly stated as peer reviewed, while the other conditions are not.
 - b. The RR study condition is stated as accepted for publication, while the other conditions are not.

- c. The PR scenario only deals with preregistration on a personal website, but this is arguably one of the less credible forms of preregistration. One could just as easily have phrased it as preregistration in a recognized third-party repository such as OSF preprints, or AsPredicted.org. This method is a more credible, and perhaps also a more common, form of preregistration. The authors should at least justify why they have chosen this specific form of preregistration. They could perhaps consider randomly presenting one of several operationalizations of preregistration in a “Radical randomization” paradigm proposed by Baribault et al. (2018).
 - d. Apart from these confounds, it is also not possible for the participants to evaluate whether the fictional study actually followed the protocol that was preregistered (in the PR/RR conditions). This is not necessarily part of what the authors wish to measure of course, but it seems like an important aspect of the trustworthiness of PR/RR in real life.
2. I think there are a few major problems with the sampling plan of the study:
 - a. I would urge the authors to consider that they may be oversampling from the population. From reproducing the Web of Science mining procedure, it seems that the authors plan on emailing every single email included in their specified search (around 15 000). They state that this should yield 99% power for the expected effect. While highly powered designs are desirable, there is also a cost implied for later meta-researchers, in that researchers who have already agreed to participate in one study may be less inclined to participate in another (at least for a while). The current authors should of course seek to achieve a high level of precision, but given that researcher participants are a limited resource pool, I think a more rigorous justification for the chosen sample size is in order, perhaps including an a priori desired level of evidence as proposed for sequential testing with Bayes factors (Schönbrodt et al. 2017).
 - b. The sampling plan section does not clearly specify whether the authors will stop after three weeks regardless of the sample size obtained, or if they will extend the data collection period until 480 subjects have been collected (It also does not specify further email addresses should be included in the unlikely case that 480 responses cannot be obtained from the current pool of emails).
 - c. The power analysis is not provided with sufficient detail to understand for what it has been calculated, and the alpha level used is not stated.
 - d. The pilot study scenarios are phrased differently than the proposed study scenarios. Due to the way they are phrased, I would guess that the effect size in the original study might be higher than in the proposed study (as the pilot RR scenario was confounded with journal being high-impact). The authors might consider using a more conservative estimate.
3. Some of the changes made to the design from pilot to proposed study does not seem to be mentioned in the main text, such as the changes to the phrasing of the author profiles.
4. If the authors are not interested in the main effect of familiarity, why is there no model included which only includes Preregistration Status + Interaction? Apologies if this is

a naïve question, but I think several other naïve readers might have the same question.

- ***Whether the clarity and degree of methodological detail would be sufficient to replicate exactly the proposed experimental procedures and analysis pipeline***

I believe the authors have provided a good amount of detail concerning the experimental protocol and analysis pipeline to allow for direct replication. However, I think the actual experimental run needs to be documented much more clearly. Ideally there would be a link to a copy of the Qualtrics questionnaire so that interested researchers could run through the experiment themselves. In addition, the manipulation check questions are not spelled out.

- ***Whether the authors provide a sufficiently clear and detailed description of the methods to prevent undisclosed flexibility in the experimental procedures or analysis pipeline***

The experimental procedures are described in sufficient detail that I am confident that any deviation would be obvious and easy to document. I see two major problems with the preregistration of the analyses:

1. The authors do not specify what results they would consider a falsification of their hypotheses. Since BFs are used, I would like to know the minimum amount of evidence in favor of H1 or H0 the authors feel they require to sufficiently corroborate or falsify their hypothesis.
2. The hypotheses are being numbered in the introduction, but this numbering is not followed up on in the analysis plan section, where it should be specified what aspects of the analyses are addressing each hypothesis, and what the falsification criteria are for each hypothesis. I would also recommend that the research questions/hypotheses should be put under a separate heading after the introduction, and structured as a numbered list.

- ***Whether the authors have considered sufficient outcome-neutral conditions (e.g. positive controls) for ensuring that the results obtained are able to test the stated hypotheses***

The authors plan to include sound manipulation checks, and they have already modified the proposed study following information from pilot data on the quality of their test instrument. I believe these measures are sufficient quality checks. However, see my issues with confounds in the preregistration status variable above.

Additional comments

Comments on the structure

- The pilot study “materials” section appears to be a mix of information about materials, participants, and statistical analyses. I think this section needs to be structured more clearly.
- I would urge the authors to refrain from referring back to the pilot in their “Proposed Study” section, and rather spell out the methodological details twice. This would in my opinion make it easier for readers to move back and forth between the preregistration and published report, if they want to verify that the protocol has been followed.
- P8. Line 11-14: Please spell out all the questions you plan to ask in which condition.

Minor comments

- In “Content analysis, Web of Science mining procedure and other study-related information.pdf” page 12, on OSF, the “subject” field in the initial and reminder email seems to have gotten switched around.
- The expected N for the study is 1100, but the target N in the ethics proposal is set to 1748. Where does the latter number come from, and why is it not mentioned in the RR?
- In your pilot email you state that it should not take more than 5 minutes to respond. With the added material in the proposed study, this will probably change. Consider quickly piloting the time required to complete the new study, and to use a slightly conservative estimate in the email.
- In the “Analysis plan” section, I think “Table 1” is referring to Table 2, and “Table 2” is referring to Table 3. Is this correct? Table 2 also seems to be a duplicate of table 1. Is it inserted twice to enhance readability?
- P2. Line 52: The sentence does not clearly express what p-hacking entails.
- P3. Line 32-33: Sentence structure is a bit clunky
- P3. Line 34-36: Since this is what you set out to test in this study, it may be a bit confusing that you are stating it as a fact in the introduction.
- P4. Line 38-39: Please link to the preregistration of the pilot here.
- P3. Line 46-48: This sentence feels a bit vague. Perhaps the authors could consider referencing some researchers or work explicitly advocating for PR and RR here?
- P4. Line 32-33: I recommend creating a DOI for this OSF project and cite the doi.org link instead, for link stability in the article over time.
- P5. Line 42: “Led” should be “lead”.
- P5. Line 43: Grammar error in sentence.
- P8. Line 45-46: Question marks in the JASP reference.

Appendix B

Dear Mr. Dunn,

My coauthors and I are pleased to resubmit our registered report: “The Effect of Preregistration on Trust in Empirical Research Findings: A Registered Report Proposal”. We were pleased with how thorough and thoughtful the reviews were and have done our best to address the reviewers’ concerns.

We copied the texts provided by reviewers below. Please find our responses to each comment in bold.

Warm regards,

Sarahanne M. Field,

And on behalf of E.-J. Wagenmakers, Rink Hoekstra, Anja Ernst, Henk Kiers and Don van Ravenzwaaij.

Associate Editor

Comments to the Author:

Four reviewers have now assessed your submission. The standard of reviews is high, with all providing a range of constructive and critical suggestions from different perspectives. The general view is that the proposed research question is important and interesting, satisfying the first criteria of Registered Reports. However the reviewers also raise a number of significant issues that cut across the remaining Stage 1 criteria and will need to be addressed thoroughly in revision.

There are too many issues raised to cover fully in summary, so to provide some guidance I will focus on the main points raised. Foremost is the need to clarify (and re-assess) the rationale, validity and precision of the familiarity hypothesis (Reviewers 1, 2 and 3), address potential confounds in the operationalisation of preregistration and concerns with the sampling plan and falsifiability of the hypotheses (Reviewer 2), consider the reliability of the principal DV and suitability of the power analysis (Reviewer 3), address concerns with the conceptualisation and measurement of trust (Reviewer 4), and with the precision of the hypotheses involving trust (Reviewer 3). Each set of reviews also raises questions about the justification and clarity the research question and methodology.

Substantive work is therefore required to achieve IPA, but the concerns appear readily addressable as part of a major revision, which will be returned to the reviewers.

Reviewer 1

Summary: The proposed study is exciting and important. I’m highly supportive of the study moving forward; however, more information needs to be provided in the proposal to a) allow replication (see responses to criteria 3, 4, and 5 below) and b) justify (and operationalize) the familiarity hypothesis (as discussed in criterion 2 below).

1. The importance of the research question(s).

The question this study proposes to answer is incredibly important. I could drone on and on here but

suffice to say: The importance of the research question is paramount and is the feature of the proposed study that reviewed the best.

2. The logic, rationale, and plausibility of the proposed hypotheses.

RE: the hypothesis that “preregistration increases researchers’ trust in findings, relative to no preregistration, and that registered reporting increases trust more than preregistration alone,” I find the logic and rationale quite sound and highly plausible.

RE: the hypothesis that “familiarity enhances the effect of preregistration on trustworthiness ratings,” I’m sorry to say that the manuscript doesn’t provide enough information to be convincing. One problem, which I’ll discuss below, is that the description of how familiarity will be operationalized is far too vague in the proposal.

Another problem is that the authors appear to be proposing that familiarity is an additive factor to the preregistration continuum. I’m not sure why familiarity is proposed as an independent, additive factor rather than, say, an interactive factor. If pre-registration increases trust, and if familiarity also increases trust, how do we know that these two features might not cancel one another out or otherwise interact?

Unfortunately, there wasn’t much in the proposal to support the prediction of familiarity as an additive factor – or much in the proposal to rule out the prediction of an interactive factor, for that matter. In fact, the familiarity manipulation felt quite exploratory to me (“Our primary interest is in the effects of preregistration protocols on trust, so we only consider familiarity as a moderating variable on the relationship between trust and preregistration protocols, not as a main effect on its own”).

Exploratory variables are fine. But they should be identified as such.

We can see how the familiarity variable appears to be somewhat exploratory in nature. Our group had discussed that familiarity should be included because it is a very strong influence on trust by virtue of the recognition heuristic mentioned in the manuscript, however there is an extent to which the expectation of how it would behave in conjunction with the preregistration variable was unknown. At the outset, we believed it should be additive, however recognize that in cases where a personal interaction leading to familiarity is negative (e.g., in the case where one co-author suspects another of poor research practice), then the effect of familiarity would behave differently. We believe this to be more of the exception than the rule, though.

Familiarity as a construct is difficult, because it is more accurately a continuous construct than a dichotomous one. For instance, we experience many different ‘levels’ of acquaintance in academia:

- 1. A direct colleague with whom you have published before**
- 2. A direct colleague with whom you regularly interact with in lab meetings (but have not published with before)**
- 3. A colleague you frequently encounter at conferences**
- 4. A colleague you have once interacted with at a conference**
- 5. A colleague you have never met, but have read some papers of**
- 6. A colleague you have never met, but have read one or two papers of**

7. A researcher you have never heard of before

We are not interested in examining all these levels in detail. Instead, we are interested in observing if this spectrum matters at all in conjunction with trust, so we plan to take two of the extremes (which is what we have now). In doing so, we believe we capture the most important aspect of familiarity, without overly complicating our design.

Despite this, it is an important issue to address that is now directly discussed in the manuscript, on pages 5 and 6: “Familiarity is a nuanced and continuous construct and is seen to varying extents in the academic setting. For instance, one is familiar with someone in their department with whom they regularly collaborate, and may even see in social settings. In contrast, one is familiar with someone in that they have talk with them at an annual conference, or have read several of their articles. We attempted to capture the extremes of this range of familiarity, and so familiarity was described in a binary way in the pilot study- the participant was said to either have collaborated previously with the fictional study's author, or not. This manipulation was intended to simulate a commonly encountered situation in the academic setting, whereby collaboration often coincides with one researcher being familiar with another to the extent that they would possibly look upon their work as being more credible and trustworthy in comparison with the work of a complete stranger.”

Related, whereas the three levels of the registration hypothesis (none, preregistered, registered report) are externally established and have a relatively agreed-upon definition, the two levels of the familiarity hypothesis seem to lack similar validity and clarity.

As I'll mention below, I had to go to OSF, download six separate PDFs, and compare and contrast them, to get any idea how familiarity (or pre-registration for that matter) was being operationalized. That's a problem for the manuscript that I'll address below. But here I'll say that after I pulled out this information, I learned that familiarity is being manipulated by the contrast between participants who read this sentence:

“A researcher with whom you have collaborated previously (and with whom you would collaborate again) has recently conducted a study.”

versus participants who read this sentence:

“A researcher unknown to you has recently conducted a study.”

To me, this manipulation doesn't seem to be about familiarity. The contrast isn't between a researcher you know and a researcher you don't know. Rather, the contrast is between a researcher with whom you've collaborated (and would collaborate with again) and a researcher whom you don't know. That's not what I expected prior to excavating the materials, and at the least needs to be better addressed in the proposal.

Lastly, on the topic of the familiarity hypothesis (which is definitely the weaker of the two hypotheses), I think the authors need to better operationalize what it means that a “researcher is unknown” to participants.

Does that mean you've never heard the researcher's name before; you've heard their name, but you've never read any of their papers; you've read one of their papers, but you wouldn't say you 'know' the

researcher; you've read a lot of their work, but you've never met them in person, so you still wouldn't say 'know' the researcher?

I'm by training a language researcher, and a good rule of thumb in language research is that the control condition is specified as well linguistically as the contrast condition. Otherwise, one is inviting all sorts of noise when participants interpret the under-specified control condition in various ways. So, I'd recommend that if the sentence establishing the contrast condition has nearly 15 words and two descriptive clauses, i.e.

“with whom you have collaborated previously (and with whom you would collaborate again)”

the sentence establishing the control condition should have a similar number of words and a similar number of descriptive clauses, e.g.,

“with whom you haven't collaborated previously (and with whom you're unlikely to collaborate in the future)”

Thank you for the suggestion- we agree that a change is needed to better operationalize the familiarity variable, and make the two descriptions much more linguistically comparable. We describe this change explicitly: “we have adapted the original familiarity manipulations, such that the conditions are more linguistically comparable. Rather than the familiar/unfamiliar conditions involving “a researcher with whom you have collaborated (and with whom you would collaborate with again)” or “a researcher with whom you are unfamiliar (and with whom you are unlikely to collaborate in the future)”, we now refer to past collaboration or a lack thereof. The familiar condition's wording is now the same, but the unfamiliar condition is now: “a researcher with whom you have never collaborated (and with whom you are unlikely to collaborate in the future)” (p. 15/16).

We discuss the operationalization of the familiarity construct on pages 5/6 of the manuscript, and have provided a discussion of this issue in detail above.

3. The soundness and feasibility of the methodology and analysis pipeline (including statistical power analysis where applicable).
4. Whether the clarity and degree of methodological detail would be sufficient to replicate the proposed experimental procedures and analysis pipeline.
5. Whether the authors provide a sufficiently clear and detailed description of the methods to prevent undisclosed flexibility in the experimental procedures or analysis pipeline.

I've grouped together these three criteria because my comments relate to all three. I have little to say about the analysis pipeline, but with regard to the methods, more information needs to be provided to assess its soundness and considerably more information needs to be provided to allow future attempts at replication.

Here's a list:

Why was Web of Science chosen? Do the authors believe the same results would be obtained if Google Scholar (a more egalitarian indexing program) or another indexing program is used for participant selection?

WoS was chosen because of how easy it is for a user to extract author email addresses from article records, and because it contains a very large selection of articles from psychology subdisciplines. Additionally, there are many filter options which allow one to select carefully which records to search, and also allow for others to select the same records at another time if need be (providing, that is, that the original searcher saved their search, or recorded their search in detail).

Our aim was to capture a representative sample of active psychology researchers, and although we recognize that WoS is not a comprehensive database, it is our opinion that comparable results would be obtained if one chose another indexing system.

How were “psychology articles” operationalized (i.e., “we used the population of corresponding authors from psychology articles”)? For example, was it required that the journal name contain the word “psychology”? If not, what were the criteria?

On our OSF page for this RR submission we include a document which sets out exactly how the database was searched and what terms and filters were used (found at <https://osf.io/s7a3d/>).

We recognize, however, that this document was not indexed in the OSF files for the project very clearly. We have now clearly labelled the document in OSF, however we will also respond directly here to this comment for the reviewer’s convenience.

Below is the process we went through to select the sample used:

1. A basic search for articles was initialized via the WoS ‘General Search’ page: <https://apps.webofknowledge.com>, in which the topic ‘psychology’ was used as a search term. All settings and search filters were as default at this point. In May of 2017, approximately 122,000 search results were found using this initial search term.
2. Several filters were then applied to refine this broad search, leaving approximately 43,500 articles left.
 - a. “Web of Science Categories” allowed for filtering of the search results via categories. In total, ten filters were selected to refine these results:
 - Psychology Multidisciplinary
 - Psychology Applied
 - Psychology Clinical
 - Psychology Social
 - Psychology Educational
 - Psychology Experimental
 - Psychology Developmental
 - Behavioral Sciences
 - Psychology Mathematical
 - b. Using “Document Types” filters, further refinement was possible through requesting typical academic document types. “Article” and “Review” were selected in this case, leaving out other types, such as “Poetry” and “Art Exhibit Review”.
 - c. “Publication Years” was used to select a range of years that was deemed wide enough to draw a large target sample, but recent enough such that most researchers in the sample would still be likely to be active in academia. The year 2013 to the year 2017 was the selected year range for the current target sample.

- d. Selecting 'English' using "Language" filters provided articles that were only in English.
3. Once these filters were applied, articles 1-5,000 were selected and placed onto a "Marked List" (5,000 is the upper limit of records allowed in a Marked List on WoS), at which point they could be exported as a .txt file with several variable columns, and saved onto the working disk. As only 500 records can be exported at a time, the process is extensive and must be done many times to achieve a large database of addresses. At export, several variables can be selected. For the current sample, the following were:
 - Authors
 - Addresses
 - Title
 - Source
 - Keywords
 - Research areas
4. A total of 10,158 records were exported.
5. Before the email addresses could be extracted from the database, it was necessary that any duplicates be removed, including duplicates that did not appear on a row on their own, but on a row along with other names as well (this is relevant, as many records imported with several email addresses, rather than just one for the corresponding author). For this purpose, a code was written in R, utilizing simple base-package functions such as unique(), and automation scripts such as for loops. Once this code was executed on the dataset, 9,992 unique addresses were obtained. Additionally, each address remained associated with the other information gathered (such as publication year and source journal), such that information about the sample could be derived.
6. As this study forms one of two studies planned (one being the pilot mentioned in the registered report proposal, and the other being the proposed full study), this sample was split in half, such that comparable samples are used for each study. In order for this to be done randomly and relatively equally, either 1 or 2 was assigned to each row in the database randomly with replacement. The database was then sorted by number, and split on that basis. The two samples obtained using this method were 4,998 and 4,994 rows long. The current study will use the email addresses of the first sample, leading to the 4,998 mentioned in the pilot study.

We link to this document directly in the manuscript now also.

Which authors were recruited? All authors on each article or only the corresponding author?

We recruited any authors whose email addresses were found in the address field. Typically this was the corresponding author of the article, however some cases included several email addresses (>10% of cases). This is now clear in the manuscript (footnote 2, p. 4): "The original aim was to use only corresponding authors, however in approximately 9 percent of cases, more than one author's email address was given. In these cases, all author email addresses were used."

In the pilot study, what did the "contact email" message say and who sent it? Similarly, what did the invitation email message say and who sent it? (I ask about who sent it because my guess is that

participants who are more open to preregistration would be more likely to respond to both messages if they came from a researcher publicly known for advocating preregistration.)

The contents of the contact email are also on OSF, but as with the WoS email address mining procedure, the document could have been more clearly indicated. The email texts are now easy to find in OSF, and are also shown below for the reviewer's convenience. The contact emails were sent and signed by the first author. Given that the first author is a first-year PhD student, and not well known, we think it is unlikely that this would have influenced the results meaningfully for the pilot, or for the proposed study.

Initial email:

“Subject: A Quick University of Groningen Questionnaire: What makes you trust someone's findings?”

Text: Dear researcher,

My name is Sarahanne Field, affiliated with the University of Groningen, and I am conducting a study on the factors that are important to researchers when considering the validity and reliability of the findings of other researchers. As part of this study, I am contacting researchers who have published in high-quality research outlets. To that end, I would like to request your participation in a brief questionnaire. I value your time, so your response is expected to take a maximum of 5 minutes of it. Please note that your responses will not be linked to you personally. Your participation is greatly appreciated.

Thank you in advance,
Sarahanne M. Field.”

If you agree to participate, please click on the following link: [LINK]
Or copy and paste the URL below into your internet browser: [LINK]
[Follow the link to opt out of future emails: Click here to unsubscribe]

Reminder email:

“Subject: Reminder for survey response: What makes you trust someone's findings?”

Text: Dear researcher,

Recently, I invited you to respond on a brief survey. I would like to remind you that the survey remains open for another seven days, should you still wish to respond. Your participation in this study is greatly appreciated, and is expected to take a maximum of five minutes of your time. Please note: If you have already participated- thank you, and please disregard this email.

Thank you, and enjoy the rest of your day!
Sarahanne M. Field”

In the proposed study, what will the invitation email say and who will send it? (I looked on OSF and could find only the six scenarios, no other relevant material.)

We intend on sending the same email text for the proposed study's data collection. The email text document is now more clearly labeled on the OSF page and an explicit link to this document is now provided in the manuscript (<https://osf.io/spjwu/>).

In the pilot study, were participants compensated? In the proposed study, will the participants be compensated? If so, what will be the compensation?

In the pilot, participants were not remunerated for participation. In the proposed study, participation will also be on a voluntary basis. This is now explicitly mentioned under both the pilot and proposed study headings.

In the pilot study, how were the different versions of the scenario assigned to participants? (In the proposed study, the manuscript explains the randomization scheme will be conducted by Qualtrics.)

In the pilot study, randomization was conducted by Qualtrics. This is now clear in the manuscript.

When is the Trust Question asked? The manuscript implies that the trust question is asked after participants read the scenario: “In the first screen, participants are presented with the instruction message, and can click through using a ‘next’ button to the scenario screen, after which they are presented with Question 1.” (Question 1 is the Trust Question.)

However, the scenario PDFs posted on OSF suggest that the question is asked before participants read the scenario. The first sentence on each scenario PDF says the following: “We aim to measure your direct or ‘primary’ response to the scenario on the next page. To that end, please attempt to answer the question given without thinking at length about it. You may use any of the information given on the next page to answer the question.”) The authors’ use of the definite-article expression “the question” implies that that “the question” has already been asked.

If the Trust Question is not asked until after participants read the scenario, then the instructions that appear on the scenario PDFs are a bit confusing and need to be cleaned up.

We agree that the wording was misleading. The key question of trust was shown *after* participants read the scenario. This has been changed in the manuscript on page 16: “In the first screen, participants are presented with... Finally, the participants are thanked for their time, and are given access to the participation information statement, where they can read about the study's aims and other relevant information.”

Additionally, a .qsf file of the proposed materials, and a link to a trial survey is now on OSF at <https://osf.io/ywq9t/>, which means that the experiment as participants will see it can now be looked at more closely by reviewers or interested others, in future.

Regarding the new manipulation check questions, I’m confused why they will be asked only for the scenarios that lead to a “yes” answer. For example, as I understand the authors, only for the familiar condition will participants be asked, “In the fictional study you just read, was the researcher responsible for the study someone with whom you were familiar?” And only in the preregistered or registered reports condition will participants be asked, “whether the study had been preregistered in some way, or whether it was the subject of a registered report.”

Why not employ a manipulation check for all conditions? In other words, why not ask the same (or similar) manipulation check question for all scenarios?

We agree with the suggestion. We will use a manipulation check for all scenarios.

The details of this are now in the manuscript on page 16: “After participants answer the primary experimental question about trust, we will pose two manipulation questions-- one for each of the independent variables. For instance, we would ask “In the fictional scenario you just read, was the researcher responsible for the study someone with whom you were familiar?” A participant in any of the three ‘unfamiliar’ conditions would be expected to answer “no” to such a question, while the other three conditions would be expected to answer in the affirmative. We will also ask participants whether the study had been preregistered in some way, or whether it was the subject of a registered report, or whether neither of these conditions are true. A participant in either of the ‘none’ conditions would be expected to answer “no”, while participants in the other conditions would be expected to indicate that there was either PR or RR present. The data of people who indicate that they did not notice the manipulations in the scenario they were given, or indicate something unexpected (e.g., familiarity with the study author if they are in the ‘unfamiliar’ condition) as revealed by the manipulation check, will be excluded from the analysis.”

The questions also appear in the experiment provided on OSF as a .qsf file (at <http://dx.doi.org/10.17605/OSF.IO/B3K75>, under the ‘Preregistration Proposed Study’ folder).

I also recommend adding in a couple of other neutral questions to better occlude the nature of the manipulation check question.

We respectfully disagree with this suggestion. The manipulation checks are done at the end of the experiment, when there is no risk of them influencing the answers of the participant. Note too, that the survey will not permit participants to go back once they have completed the crucial question. Given this, we do not think it necessary to occlude the manipulation check.

Returning to the scenario PDFs, it was deeply unfortunate that the only way I could see the manipulation was to (as I whined about above) click over to OSF, download six different files, and then compare each of the six files one to one another. That’s asking a lot of reviewers, particularly for information that a) needs to be provided in the manuscript, and b) is simple to provide in the manuscript!

We apologize for not providing these- we agree it would have been good for them all to be included. These are now included attached to this response to reviews letter in an appendix, for the convenience of the reviewer. For the final manuscript, we will include one of the six in an appendix as an example, and will provide a direct link to all six on the OSF page. We also now explicitly show in text the differences between the conditions in terms of what is said, such that the need for going back and forth between the six scenario texts will not be necessary for readers of the final manuscript. Additionally, the OSF .qsf file (or link to a trial, for those who do not have a Qualtrics account) shows the materials as participants would see it in the experiment.

But, from what I could discern, the Familiar versus Unfamiliar conditions differ in that the Familiar scenarios have the sentence, “A researcher with whom you have collaborated previously (and with whom you would collaborate again) has recently conducted a study” whereas the Unfamiliar scenarios have the sentence I wrote about above, “A researcher unknown to you has recently conducted a study.”

And, from what I could discern, the Registration conditions differ in that the None scenarios include the sentence, “The paper makes no mention of any previously documented sampling plan or study design,” the Preregistered scenarios include the sentence, “Prior to conducting this study, a detailed sampling plan, analysis strategy, and the study hypotheses were posted publicly on the author’s website,” and the Registered Report scenarios include the sentence, “Prior to conducting the study, its

protocol was peer-reviewed and conditionally accepted by the editorial committee at the publishing journal.”

If all the above is true, this information must be stated in the manuscript.

This is indeed true and is now described clearly in the manuscript. For both the pilot and proposed study, we have clearly shown differences in the wording, as well as writing a paragraph detailing these changes on page 16 (as quoted above).

Two more concerns, both relatively serious: First, both the pilot study and the proposed study are lacking demographic information about the participants. Given the 12.5% response rate, we really need to know who chose to participate in this type of study. As I alluded to above, different samples could provide different responses (e.g., samples of researchers who are enthusiastic about preregistration are most likely to provide responses that indicate that preregistration boosts trust).

We had not asked such information in the pilot because we did not want responding to take more time than we thought was necessary for the participant. That being said, we agree it is valuable to know about that 12%. We will ask some simple questions, and have specified those we plan to ask (and the use of the information they provide) in the manuscript. They are also in the experiment file on OSF (at <http://dx.doi.org/10.17605/OSF.IO/B3K75>, in the ‘Preregistration Proposed Study’ folder). Specifically, we ask about familiarity with the PR and RR protocols, we ask about participants’ opinions on these protocols, and finally, we ask whether the participant chose to completely respond to the study because of their opinions of the protocols. These questions are pertinent because they relate directly to the validity of participants’ responses for the dependent variable.

Second, there is no justification provided for designing the study as between- rather than within-subjects. Because of carryover effects and subject demand (fatigue), a between-subjects might be the best choice, but no justification is provided.

Indeed- we chose the between-subjects design for the reasons you mention here, but also because we do not want people to guess the aim of the study as we fear they would in a within-subjects setup. We agree that our justification for this choice should be provided- this rationale is now explained in the manuscript on page 5: “We chose the between-subjects design for two reasons, first, we wanted to minimize the demand of participation on the subjects. Being in all six conditions would require a participant to read six sets of materials in detail, and would cost a great deal of time. Second, we did not want any carry-over effects between the different conditions. Specifically, a within-subjects design could alert participants to the goals of the study, which could lead to bias in the form of socially desirable answers.”

6. Whether the authors have considered sufficient outcome-neutral conditions (e.g. absence of floor or ceiling effects; positive controls; other quality checks) for ensuring that the results obtained are able to test the stated hypotheses.

The addition of manipulation checks and some quality control checks is a good move.

One last and small quibble. The manuscript states: “the merits of PR and RR are advocated by a group of scientists often referred to as ‘meta-scientists’.” I know numerous scientists who advocate PR and RR, myself included, and none of us refer to ourselves as ‘meta-scientists.’ We’re just scientists. In other words, it’s not just “meta-scientists” who advocate pre-registration.

We see how the wording can be improved- we did not intend to imply that meta-scientists are the *only* ones to advocate for preregistration and registered reporting for this is indeed not the case! We have now changed the phrase you quoted above to: "...the merits of PR and RR are advocated by many researchers worldwide..."

Reviewer 2.

NB! This reviewer is not sufficiently experienced with Bayesian model selection and comparison to fully evaluate the soundness of the planned analyses.

Please comment explicitly on each of the following points in your comments to the authors

- The significance of the research question(s)

The research question seems highly important to answer. The perceived trustworthiness of preregistration and registered reports are a crucial component of their usefulness, and the extent to which they are increasing perceived trustworthiness would be important information for funders considering whether to prioritize such research in grants, for journals considering such research for publication, and for individual researchers considering implementing the practice of preregistration in their own research.

- The logic, rationale, and plausibility of the proposed hypotheses

The rationale for the expected correlation between Preregistration Status and Trustworthiness is clearly laid out and justified. The hypothesis concerning this correlation is also in my opinion highly plausible (both given the results of the pilot data, and my subjective prior).

The rationale for the moderating effect of familiarity is less clear to me. While the inclusion of the moderator in and of itself is rather straight-forward, I have the following concerns:

1. It is not obvious to me why the authors are only interested in familiarity as a moderator on the relationship between preregistration status and trustworthiness, and not as a main effect.

2. It is not obvious to me why familiarity is the only moderating variable considered. I do not expect the authors to lay out every possible moderator of interest of course, and I certainly see that there is a trade-off between including potential moderators and maintaining sufficient measurement precision. However, I would at least like to see a brief discussion of potential moderators, and a justification for why author familiarity could be considered the most important moderator.

We agree that familiarity makes sense to be a main effect, as well as included in an interaction. This has now been made clear in the manuscript.

We also agree that it is beneficial to discuss other potential moderators. The manuscript now contains a brief discussion of the candidate variables that we considered in the designing of the study beginning on page 5, in a section discussing the rationale for our choices of variable. Additionally, we will explicitly discuss other potential moderators in the discussion section of the final manuscript that may also be relevant to the research question.

- The soundness and feasibility of the methodology and analysis pipeline (including statistical power analysis where applicable)

The proposed study is certainly feasible as currently planned, which is evidenced by results from the pilot study. However, I believe there are several major conceptual issues that needs to be resolved

and/or clarified in the preregistration before data collection can begin. I list these below in order of importance.

1. When looking at the proposed scenarios, I think there might be important confounds in the current operationalization of the preregistration status variable:

a. The RR study condition is explicitly stated as peer reviewed, while the other conditions are not.

A strength of registered reporting is that the protocol is peer reviewed before data collection is carried out. It is for this reason that the fictional results in the RR condition are stated as peer reviewed.

b. The RR study condition is stated as accepted for publication, while the other conditions are not.

The RR study conditions (RR/unfamiliar and RR/familiar) do not mention actual publication, just in principal acceptance: “Prior to conducting the study, its protocol was peer-reviewed and conditionally accepted by the editorial committee at the publishing journal”. Again, this scenario contains this information, as it is a part of describing the RR protocol.

c. The PR scenario only deals with preregistration on a personal website, but this is arguably one of the less credible forms of preregistration. One could just as easily have phrased it as preregistration in a recognized third-party repository such as OSF preprints, or AsPredicted.org. This method is a more credible, and perhaps also a more common, form of preregistration. The authors should at least justify why they have chosen this specific form of preregistration. They could perhaps consider randomly presenting one of several operationalizations of preregistration in a “Radical randomization” paradigm proposed by Baribault et al. (2018).

We agree that our original choice was not as credible as it could be- we have changed ‘author’s website’ to ‘author’s OSF page’ and also explicitly refer to this change on page 16. While there are different ways to preregister, the OSF ‘route’ is currently very popular and defensible.

We very much like your idea of using the radical randomization paradigm for the operationalizations of preregistration, though we believe it is perhaps a little ambitious and elaborate for the current project’s design and aim.

d. Apart from these confounds, it is also not possible for the participants to evaluate whether the fictional study actually followed the protocol that was preregistered (in the PR/RR conditions). This is not necessarily part of what the authors wish to measure of course, but it seems like an important aspect of the trustworthiness of PR/RR in real life.

We do not agree that this (participants being unable to evaluate whether the protocol was actually followed) is necessarily a problem. It is our opinion that whether someone believes that PR/RR protocols will be followed is directly related to whether or not they trust the results of these studies. This being said, it is worth mentioning in the discussion of the final manuscript, because it is indeed a point of contention regarding PR/RR protocols, and deserves specific mention as one of the potential mechanisms underlying trust judgments.

2. I think there are a few major problems with the sampling plan of the study: a. I would urge the authors to consider that they may be oversampling from the population. From reproducing the Web of Science mining procedure, it seems that the authors plan on emailing every single email included in their specified search (around 15 000). They state that this should yield 99% power for the expected

effect. While highly powered designs are desirable, there is also a cost implied for later meta-researchers, in that researchers who have already agreed to participate in one study may be less inclined to participate in another (at least for a while). The current authors should of course seek to achieve a high level of precision, but given that researcher participants are a limited resource pool, I think a more rigorous justification for the chosen sample size is in order, perhaps including an a priori desired level of evidence as proposed for sequential testing with Bayes factors (Schönbrodt et al. 2017).

We agree generally that this can be an issue. The power analyses (BFDA, actually) are based on an assumption of effect size that may not apply with the changes to our methodology. It is your opinion, expressed in a point below, that this might be the case, and we seriously consider this to be a possibility. We wish to provide as high a level of precision possible in the full study as a result, and so prefer to stick to the original sampling plan.

b. The sampling plan section does not clearly specify whether the authors will stop after three weeks regardless of the sample size obtained, or if they will extend the data collection period until 480 subjects have been collected (It also does not specify further email addresses should be included in the unlikely case that 480 responses cannot be obtained from the current pool of emails).

We have extended the sampling plan section to discuss this.

c. The power analysis is not provided with sufficient detail to understand for what it has been calculated, and the alpha level used is not stated.

The frequentist power analysis we included originally has been removed. At the suggestion of yourself and another reviewer, we have included a Bayes factor design analysis instead. We used the means and standard deviations from the pilot data to simulate from, and have now updated the manuscript with this power calculation.

d. The pilot study scenarios are phrased differently than the proposed study scenarios. Due to the way they are phrased, I would guess that the effect size in the original study might be higher than in the proposed study (as the pilot RR scenario was confounded with journal being high-impact). The authors might consider using a more conservative estimate.

We agree that it is possible that the modifications to the original methodology may lead to a smaller effect size than the pilot results showed. Since the original power analysis has been removed, however, this is no longer problematic for the originally conducted power analysis. That being said, our BFDA does rely on the pilot's ES. We refer to the possibility of the pilot and proposed effect magnitudes not matching in a few places now in the manuscript; most prominently in footnote 6 on page 12: "We recognize that in using the means and SDs of the pilot we assume that they reflect the true population means and SDs for this effect. Naturally, we cannot assume this, however do so for the purposes of providing a reasonable estimate of power for the proposed study."

3. Some of the changes made to the design from pilot to proposed study does not seem to be mentioned in the main text, such as the changes to the phrasing of the author profiles.

We agree this should be more clearly stated. We have now edited the manuscript with greater detail about the texts and changes. We describe the differences in wording explicitly in the materials section (p. 13), as well as highlighting the changes we have made in a specific section, which spans pages 16 and 17.

We have additionally added a .qsf file to the OSF page for each the pilot and proposed studies (at <http://dx.doi.org/10.17605/OSF.IO/B3K75>, under the ‘Preregistration Pilot Study’ and ‘Preregistration Proposed Study’ folders respectively), such that the materials may be viewed as part of the survey and compared.

4. If the authors are not interested in the main effect of familiarity, why is there no model included which only includes Preregistration Status + Interaction? Apologies if this is a naïve question, but I think several other naïve readers might have the same question.

Familiarity will be considered a main effect in the full study, so this is no longer a problem. If we had have kept the original setup without familiarity as a main effect, though, it would still make sense to include both familiarity and preregistration with the interaction, because the interaction is a lower order effect that could be related to a higher order effect. Therefore it should still be included in any model featuring an interaction between familiarity and preregistration (except in some circumstances where omitting the main effect would make sense- e.g., in the case where we would expect to see a perfect crossover interaction).

- Whether the clarity and degree of methodological detail would be sufficient to replicate exactly the proposed experimental procedures and analysis pipeline I believe the authors have provided a good amount of detail concerning the experimental protocol and analysis pipeline to allow for direct replication. However, I think the actual experimental run needs to be documented much more clearly. Ideally there would be a link to a copy of the Qualtrics questionnaire so that interested researchers could run through the experiment themselves. In addition, the manipulation check questions are not spelled out.

We agree that access to the Qualtrics file could be useful. The .qsf file is now available on OSF (more stable than the survey link) as mentioned above, and will be also for the proposed study, should our proposal be accepted. A trial version of the proposed study is included already, for reviewers’ use, also on the OSF page. These .qsf files can be accessed through Qualtrics itself for anyone with access to an account.

- Whether the authors provide a sufficiently clear and detailed description of the methods to prevent undisclosed flexibility in the experimental procedures or analysis pipeline

The experimental procedures are described in sufficient detail that I am confident that any deviation would be obvious and easy to document. I see two major problems with the preregistration of the analyses:

1. The authors do not specify what results they would consider a falsification of their hypotheses. Since BFs are used, I would like to know the minimum amount of evidence in favor of H1 or H0 the authors feel they require to sufficiently corroborate or falsify their hypothesis.

An explanation of our expectations is now in the manuscript on page 19: “Based on the pilot results and the BFDA conducted, we expect that the inclusion Bayes factor for the preregistration status main effect will be over 10 in favor of the alternative hypothesis by the time N has reached 530. We consider this to be strong corroborating evidence of our expectation that preregistration greatly increases feelings of trust in research results, relative to cases in which preregistration has not taken place. We also consider an inclusion Bayes factor between 3 and 10 in favor of the alternative hypothesis corroborative of our hypothesis, however such evidence would be less compelling.

In the case of the familiarity main effect, and the interaction between the two main effects, we consider our hypotheses to be corroborated in the case that we see an inclusion Bayes factor above 3 in favor of the alternative. As with the other main effect, any result above 10 indicates strong corroboration. If, instead, the inclusion Bayes factor for the main effects or interaction is smaller than 1/10 (that is, it actually indicates strong evidence in favor of the null hypothesis), we consider our hypotheses falsified. Any inclusion Bayes factor between 1/3 and 3 (i.e., 'anecdotal evidence', according to Jeffreys' 1961 thresholds) suggests that we cannot either corroborate or falsify any of the hypotheses until more data has been collected.”

2. The hypotheses are being numbered in the introduction, but this numbering is not followed up on in the analysis plan section, where it should be specified what aspects of the analyses are addressing each hypothesis, and what the falsification criteria are for each hypothesis. I would also recommend that the research questions/hypotheses should be put under a separate heading after the introduction, and structured as a numbered list.

We agree and have improved the structure of this content in the manuscript.

- Whether the authors have considered sufficient outcome-neutral conditions (e.g. positive controls) for ensuring that the results obtained are able to test the stated hypotheses

The authors plan to include sound manipulation checks, and they have already modified the proposed study following information from pilot data on the quality of their test instrument. I believe these measures are sufficient quality checks. However, see my issues with confounds in the preregistration status variable above.

Additional comments

Comments on the structure

- The pilot study “materials” section appears to be a mix of information about materials, participants, and statistical analyses. I think this section needs to be structured more clearly.

The pilot section of the manuscript is now more clearly structured.

- I would urge the authors to refrain from referring back to the pilot in their “Proposed Study” section, and rather spell out the methodological details twice. This would in my opinion make it easier for readers to move back and forth between the preregistration and published report, if they want to verify that the protocol has been followed.

Originally we were concerned about information redundancy, but it is indeed easier for readers to compare with the changes you suggest. We have now supplied detailed information about the pilot and proposed methodologies in the appropriate sections, rather than referring readers back and forth.

- P8. Line 11-14: Please spell out all the questions you plan to ask in which condition.

A more detailed description of the planned materials is now in the manuscript, as mentioned in response to an earlier point.

Minor comments

- In “Content analysis, Web of Science mining procedure and other study-related information.pdf” page 12, on OSF, the “subject” field in the initial and reminder email seems to have gotten switched around.

The amended .pdf is on OSF at <https://osf.io/spjwu/>

- The expected N for the study is 1100, but the target N in the ethics proposal is set to 1748. Where does the latter number come from, and why is it not mentioned in the RR?

This is an odd oversight on the part of the first author. The ethics form has been amended and re-approved by the appropriate ethics committee. Please find the updated form on the OSF page: <https://osf.io/pz68y/>

- In your pilot email you state that it should not take more than 5 minutes to respond. With the added material in the proposed study, this will probably change. Consider quickly piloting the time required to complete the new study, and to use a slightly conservative estimate in the email.

We intentionally greatly overestimated this time in the pilot such that we did not mislead anyone- we overestimated it to such an extent, in fact, that the time frame mentioned should still be enough time in which to respond. The first author has already piloted the new material in two of the conditions (the none, unfamiliar condition, and the RR, familiar condition). For the first condition, the time was 3 minutes and for the second it was just under 4 minutes. We believe that few would require longer than the time we specified, especially given our explicit instruction to not dwell on the content too long.

- In the “Analysis plan” section, I think “Table 1” is referring to Table 2, and “Table 2” is referring to Table 3. Is this correct? Table 2 also seems to be a duplicate of table 1. Is it inserted twice to enhance readability? **Yes, Table 2 is a duplicate, it is included again for readability, but indeed the headings are confusing. This is fixed now.**

- P2. Line 52: The sentence does not clearly express what p-hacking entails. **Fixed.**

- P3. Line 32-33: Sentence structure is a bit clunky. **Fixed.**

- P3. Line 34-36: Since this is what you set out to test in this study, it may be a bit confusing that you are stating it as a fact in the introduction. **We agree- Fixed.**

- P4. Line 38-39: Please link to the preregistration of the pilot here. **Fixed.**

- P3. Line 46-48: This sentence feels a bit vague. Perhaps the authors could consider referencing some researchers or work explicitly advocating for PR and RR here? **Fixed.**

- P4. Line 32-33: I recommend creating a DOI for this OSF project and cite the doi.org link instead, for link stability in the article over time. **Great idea:**
<http://dx.doi.org/10.17605/OSF.IO/B3K75> **We still occasionally refer to the specific sections of OSF links though, given that there are many documents in the index in several different folders.**

- P5. Line 42: “Led” should be “lead”. **Fixed (whole paragraph changed, so mistake no longer there).**

- P5. Line 43: Grammar error in sentence. **Fixed, same reason as point above.**

- P8. Line 45-46: Question marks in the JASP reference. **Fixed.**

Reviewer 3

Overall, I think that this RR proposal is a valuable study, that promises to shed more light onto the (psychological) effects of preregistration. As the field is moving forward in implementing open science practices, an evaluation of their effectiveness and side-effects is an important step, to which this papers adds. In general I support an IPA of this proposal, but have some issues I'd like to see addressed before:

1. More precise hypotheses

In the research questions you suddenly introduce "objective" trust. What is that? Please define.

The word 'objective' in this case is superfluous. We have removed it from the manuscript, and now simply refer to and describe 'trust'.

Furthermore, you write: "We expect that preregistration does increase the trust participants have in research results, and that familiarity increases trust in conjunction with preregistration." What does "in conjunction" mean? Is that a hint for an interaction effect? If yes, in what direction?

Yes, indeed this is a hint for an interaction effect. We expected that familiarity would have an additive effect on trust, that is, it would increase trust ratings in conjunction with preregistration status even more than preregistration status alone would cause. We see that our descriptions of the interaction and the variable in general are not sufficiently clear, and have amended the manuscript to fix this issue in several different places (e.g., in pages 5/6 and 16).

2. Description of results of pilot study

You write on p. 5: "Incorporating familiarity appears to led to the expected behavior of the dependent variable– familiar researchers led to a slight increase participant trust, over and above what alone was explained by PR and RR." This description seems misleading, as the BF does not support this claim. If I read the BF tables correctly, there is no evidence for the inclusion of familiarity (in contrast, there is moderate evidence for not including it). Hence, there is no support for your hypothesis that "familiarity increases trust", and this conclusion should be explicitly written down.

We agree that it is misleading- please find a more accurate interpretation in the manuscript (p. 8/9).

For readers unacquainted with BF, please verbally interpret the BF for the interaction term.

We now verbally interpret all reported BFs in the manuscript.

Table 2 seems misplaced at its current position - why not report that as an exploratory result of the pilot study?

Table 2 is a reproduction of Table 1 (which is in the pilot study section, relating to the results we report there)- we reproduced it for readability. This was not clear in the manuscript, however, so we have added a note to clarify.

3. Proposed study

I guess you want to generalize the results to a variety of preregistered studies - this would actually call

for a design where the study content (in the experimental material) is a random factor. I realize that this could lead to power problems, but I think at least a discussion of the issue is warranted.

We are not exactly sure what you mean with this. If “a variety of preregistered studies” means studies in different disciplines, for instance, this is our response: While we do not see obvious reasons as to why researchers would trust PR/RR protocols differently as a function of their field of study, we certainly agree that a discussion of this issue is necessary. If instead you refer to the variety of different preregistration *types* available (e.g., is it preregistered on OSF, is it preregistered on AsPredicted.org...), we certainly agree that we would encounter power problems. We will discuss these two issues in detail in the discussion of the final manuscript.

Is the reported power analysis on p. 7 a frequentist power analysis? Seems quite odd in the context of a purely Bayesian data analysis. Why not report a Bayes factor design analysis (see Schönbrodt & Wagenmakers, 2017) for a fixed-n design, i.e., $\text{prob}(\text{BF} > 10 \mid \text{assumed effect size})$.

We agree that a power analysis for Bayesian results (as much as the concept of power *can* be defined in Bayesian terms, that is) is always superior to a frequentist power analysis in the context of a purely Bayesian analysis such as this. Although no current BFDA implementation exists for a 2x3 ANOVA, Angelika Stefan kindly provided some R code which helped us conduct a BFDA for our design. Hopefully the results along with the description in the manuscript are sufficient.

I think one underappreciated factor of the replication crisis is the low reliability of our measurements. Many social psychology studies rightly have been criticized for measuring their central variables with ad-hoc single-item measures. I would urge the authors to think about how to increase the reliability of the central DV measure in this study as well; from a psychometric point of view using a single item for a multifaceted construct seems careless.

We agree that it is very simplistic to use a single Likert measure for a process as complex as that involved with making a trust judgment on the research of others. That being said, it is our opinion that pinning down this complex process is too ambitious for this study. What we intend to offer with the proposed study is a look at how PR and RR protocols affect trust judgments in a very pared-down way.

We discuss this in detail now in the manuscript on pages 5 and 6 in a section discussing our choice of variables: “Preregistration status reflected whether or not the fictional study protocol had been preregistered in some way before data collection. In the scenario texts, as you will see below, preregistration status was described in terms of the process involved. With the descriptions used, we hoped to broadly capture the main differences between different gross ‘levels’ of preregistration: none at all (the classic procedure by which one simply carries out a study without pre-specifying any plans), preregistration (by which one somehow logs the major plans for a study including, but not limited to hypotheses and sampling plan), and registered report (the process in which one submits a detailed study plan to a journal for peer-review before the study is carried out).

Familiarity is a nuanced and continuous construct and is seen to varying extents in the academic setting. For instance, one is familiar with someone in their department with whom they regularly collaborate, and may even see in social settings. In contrast, one is familiar with someone in that they have talk with them at an annual conference, or have read several of their articles. We attempted to capture the extremes of this range of familiarity, and so familiarity was described in a binary way in the pilot study-- the participant was said to either have collaborated

previously with the fictional study's author, or not. This manipulation was intended to simulate a commonly encountered situation in the academic setting, whereby collaboration often coincided with one researcher being familiar with another to the extent that they would possibly look upon their work as being more credible and trustworthy in comparison with the work of a complete stranger. ”

On p. 9 you write: "In the absence of data, we assume a priori that each of the five models in our two-factor design is equally likely ($p = .20$)." Well, you do have data from the pilot study. Why not updating your priors from pilot  main study?

Although we recognize that this might be a desirable or even obvious choice for some, we prefer to stay with our original plan of using the default prior. It is our opinion that once one begins to change prior odds, it is necessary to open a discussion about why certain prior odds were chosen and not others. Although this doesn't necessarily pose a problem, it is a direction we have chosen against.

4. Minor comments:

- p. 3: "Researchers' work is taken more seriously and its academic content trusted more by others when PR or RR have been part of the research process.": Here (and throughout this paragraph) is either a citation needed, or it should be made clearer that this is a hypothesis and not an empirical finding. Do we already have evidence for that claim, or is that what we expect? Isn't the current project just about finding that out?

This is now worded slightly differently in the manuscript to reflect the fact that although proponents of PR and RR expect that “researchers' work is taken more seriously and its academic content trusted more by others when PR or RR have been part of the research process”, this is an assumption without an empirical basis.

- In the pilot study, and also the main study: Did/will you explain the difference of PR and RR to participants? In my experience, this is not clear to many.

We intentionally did not explain the distinction between PR and RR. We did not even describe these in specific terms at all, in fact. This was deliberate: considering the fact that PR and RR are becoming more popular and discussed in psychology, we thought of the possibility that describing them by name would introduce two problems. First, and most obviously to us, we thought it might make people inclined to respond in a socially acceptable manner, that is, trust PR or RR more simply because they might feel like they should. The second is that there are differences in how one can PR a study, and differences in how different journals approach RR submissions. While there are certainly basic definitions applying to both (e.g., that you get pre-data-collection peer review with RR and not with PR), we did not want variability in people's knowledge and experience with PR and RR to lead to unnecessary noise in the data.

That being said, we agree that the distinctions between PR and RR are not clear to many, and that such a distinction is important! We will include an explanation of the differences in the debriefing note accessible to participants after completion. The amended participant information statement can be found on the OSF site for the project: <https://osf.io/49d2h/>

Here are the journal's check points:

The significance of the research question(s): Relevant.

The logic, rationale, and plausibility of the proposed hypotheses: Makes sense.

The soundness and feasibility of the methodology and analysis pipeline (including statistical power analysis where applicable): Please improve power analysis (see above); anything else is state of the art.

Whether the clarity and degree of methodological detail would be sufficient to replicate exactly the proposed experimental procedures and analysis pipeline: Sufficient.

Whether the authors provide a sufficiently clear and detailed description of the methods to prevent undisclosed flexibility in the experimental procedures or analysis pipeline: Sufficiently clear.

Whether the authors have considered sufficient outcome-neutral conditions (e.g. positive controls) for ensuring that the results obtained are able to test the stated hypotheses: They added a manipulation check.

Reviewer 4

Comments to the Author(s)

The current project proposes to examine the effect of study pre-registration (vs. RR vs. control) and collaborator familiarity on trust.

Major concerns:

1. Conceptualization of trust: You propose to add explanation text to the study defining trust (which was not contained in the pilot study): "the word 'trust' in this question can be interpreted in many ways: 'reliable,' 'replicable,' 'a true effect,' 'valid,' or 'high-quality.'" (p. 8). This text was helpful for me, because I do not feel that the construct of trust was fully explained in the introduction. I personally had a strong (negative) reaction to the word "trust." Trust implies that checking someone's work is not needed, but the benefit of pre-registration is that it enables you to check someone's work.

All of these open science initiatives that we have proposed and enacted are designed to increase research/evidence quality. It doesn't matter to me whether we can "trust" a finding more. Trust should be an outcome of quality, but it is not the same thing as quality. A pre-registered or RR finding is more credible than an unregistered study, because we can verify its claims more fully. That is, transparency in detailing the data generating process makes it possible to verify a claim. RR studies may be de facto higher quality because of improvements introduced in the two-stage review process (but again, quality is the desired outcome; trust should be irrelevant). Let me try to be as clear as I can: pre-registered studies should not be trusted more, but on average, the data contained in them are more valuable, because it is possible verify the claims the authors make. I wouldn't say that value is the same thing as trust, either.

We agree with this point. We mean a similar thing with 'trust' as you do with 'credible'. That being said, we have edited the introduction section in several places to reflect the important distinction between trust and quality/value of data and to more elaborately describe what we mean by our use of the word 'trust'.

I will add that it is a separate question about whether researchers *do* trust pre-registered or RR studies more than traditional studies (versus whether they *should*), but it seems weird to me to couch higher trust as the desired outcome. We should care about the quality of evidence, not about trust. For instance, you write "Rebuilding trust in our research findings is vital" (p. 3), but I would rephrase this to "Increasing the quality of our evidence base is vital." Similarly, "several new initiatives have been launched which focus on restoring the trust of the scientific community in its research" (p. 2). But this seems to me to be a mischaracterization, and it makes it sound like we are just doing reputation management rather than trying to improve research quality.

Again, we agree and have amended the manuscript's introduction to reflect this perspective.

2. Measurement of trust: Unless I missed it, I do not see the full text of the question asking about trust in the text or the supplemental materials. It should be provided, and you might consider putting your Qualtrics (.qsf) file on OSF too. I reviewed the materials on OSF and did not see anything like this there.

We agree that the original materials as they appeared on Qualtrics should be available, and using the .qsf file makes it very easy for others to access the materials as they appeared to participants. You will now find the .qsf file for the pilot on OSF. Additionally, we have created a trial version of the proposed study for reviewers to see, which is now also in .qsf form on the OSF website (at <http://dx.doi.org/10.17605/OSF.IO/B3K75>, under the 'Preregistration Pilot Study' and 'Preregistration Proposed Study' folders respectively).

A single item ad hoc measure of the key dependent variable seems to leave a lot to be desired.

We recognize that our dependent variable measure is in many senses too simplistic to adequately represent what is probably quite a complex process (that is, how researchers decide how much faith to put in the findings of others). That being said, it can still give us a good look at how the PR and RR protocols can influence trust. We think it would be too ambitious for this proposal to attempt to address all facets of the 'trust process', but we acknowledge that some readers might take issue with our choice.

We will discuss this issue in detail in the discussion section of the final manuscript.

3. You write "It is unclear whether the scientific community at large trusts the results borne of the PR or RR process more than studies which have not been subject to these processes" (p. 3). I think this is the question your study is designed to answer. It may be interesting to know this, but perhaps not for the reasons you outline in the introduction.

This comes back to your earlier points about the distinction between quality and trust- we agree, and have adjusted the writing to better reflect what we mean (that is, it is not: "the main motivation for PR/RR should not be because your findings are trusted more", and is instead "trust is part of what will bring PR/RR into common practice in psych science, and the wide use of the protocols will hopefully also increase research quality"), and also to highlight what we changed in response to your earlier suggestions.

Additional point:

4. When I was reviewing the materials on OSF, I noticed that you state the following in your ethics application: "As aforementioned, participants will be naive to the exact objectives of the study, and

will be only exposed to the stimulus of 1 (of 6) experimental conditions. Naivety is important for this study as social desirability may be expected to play a role in the participants' responses. We want to ensure that the participants' responses are not impacted by what they might think is expected of them." This seems to be inconsistent with the "participant information form" which gives away many of the details of the study including the focal variables. Perhaps I misunderstood and maybe this form is a debriefing form rather than a consent form, but I would urge you to reconsider your framing in the participant information form to try to preserve participant naivety.

We see how this would seem confusing. Whilst data collection was underway for the original pilot study, the participant information statement was not available online. It was only made available to participants after they completed the study via a link to the PDF in Qualtrics. This would be the same for the proposed study- the information statement would not be publicly available online during data collection, except to participants who have completed the study, and would only become publicly available again after data collection had ended.

We are explicit about the order of the procedure for participants in the manuscript now: "Once participants had completed the survey, they could follow a url to a PDF of the participant information statement, which provided an explanation of the study's aims, as well as some information in general about the context of the study and related content (such as an explanation of preregistration and the registered reports format). At the time of data collection, this statement was only available to those who had completed the study via the link, and not on the project's OSF page. This was to ensure that participants were indeed naive to the study, and less likely to show undesirable bias characteristics during participation."

Signed,
Katie Corker

We aim to measure your direct or 'primary' response to the scenario on the next page. To that end, please attempt to answer the question on the following page without thinking at length about it. You may use any of the information given on the next page to answer the question.

Author Profile:

A researcher with whom you have collaborated previously (and with whom you would collaborate again) has recently conducted a study. The paper makes no mention of any previously documented sampling plan or study design.

An excerpt of the relevant study information is given on the next page.

1 Introduction

We aimed to investigate the relationship between caffeine and visual representation in the brain. We hypothesized that participants given a high dosage of caffeine would make more errors in the mental rotation task, compared with participants administered a placebo.

2 Methods

2.1 Participants

The experimental sample comprised 100 participants— 52 females (mean age = 26.5) and 48 males (mean age = 25) that had been recruited from the general public in London, UK. Normal or corrected-to-normal vision and full use of both hands was a requirement for participation. Participants with a known allergy to caffeine were excluded, as were participants who reported drinking an excessive amount of coffee daily (> 4 cups per day).

2.2 Design

To ensure gender balance in the experiment, males and females were separately randomly assigned to the 'caffeine' and 'placebo' conditions: both conditions contained 26 females and 24 males. The caffeine group were given a 'high-dose' (400mg) caffeine tablet with a 8-ounce glass of water 45 minutes before the mental rotation task, while the placebo group were given a placebo pill with an 8-ounce glass of water 45 minutes before the task.

The experiment was double-blind, and the pills administered were visually indistinguishable. The dependent variable was error rate on a mental rotation task, quantified as total percentage incorrect in 40 trials.

2.3 Materials

In each of the 40 trials, two objects were presented alongside one another on a computer monitor (display dimensions: 336.3mm x 597.9mm) in one of 8 rotation angles (0°, 45°, 90°, etc). The trial presentation order was randomized. Each trial stimulus appeared as black 3D figures on a white background as in Figure 1. Participants indicated whether or not the right-hand object was the same as the left-hand object ('same') or different ('different'). To respond to each object pair, participants pressed either the 'S' key for 'same', or the 'D' key for 'different'. The task was not speeded.

Fig. 1 Two mental rotation objects: the one on the right is *not* a rotated version of the one on the left. The correct response in this trial is therefore 'different'.

3 Results

A one-tailed independent samples t-test was conducted to compare mental rotation error rate between the two groups. As predicted, the caffeine group had a significantly higher error rate than the placebo group: $t(98) = 2, p = .024, d = 0.40$. The data are shown in Figure 2.

Fig. 2 The boxplots and histograms show the mental rotation error rates for the two experimental conditions **after caffeine administration**.

We aim to measure your direct or 'primary' response to the scenario on the next page. To that end, please attempt to answer the question on the following page without thinking at length about it. You may use any of the information given on the next page to answer the question.

Author Profile:

A researcher with whom you have never collaborated (and with whom you are unlikely to collaborate in the future) has recently conducted a study. The paper makes no mention of any previously documented sampling plan or study design.

An excerpt of the relevant study information is given on the next page.

1 Introduction

We aimed to investigate the relationship between caffeine and visual representation in the brain. We hypothesized that participants given a high dosage of caffeine would make more errors in the mental rotation task, compared with participants administered a placebo.

2 Methods

2.1 Participants

The experimental sample comprised 100 participants— 52 females (mean age = 26.5) and 48 males (mean age = 25) that had been recruited from the general public in London, UK. Normal or corrected-to-normal vision and full use of both hands was a requirement for participation. Participants with a known allergy to caffeine were excluded, as were participants who reported drinking an excessive amount of coffee daily (> 4 cups per day).

2.2 Design

To ensure gender balance in the experiment, males and females were separately randomly assigned to the 'caffeine' and 'placebo' conditions: both conditions contained 26 females and 24 males. The caffeine group were given a 'high-dose' (400mg) caffeine tablet with a 8-ounce glass of water 45 minutes before the mental rotation task, while the placebo group were given a placebo pill with an 8-ounce glass of water 45 minutes before the task.

The experiment was double-blind, and the pills administered were visually indistinguishable. The dependent variable was error rate on a mental rotation task, quantified as total percentage incorrect in 40 trials.

2.3 Materials

In each of the 40 trials, two objects were presented alongside one another on a computer monitor (display dimensions: 336.3mm x 597.9mm) in one of 8 rotation angles (0°, 45°, 90°, etc). The trial presentation order was randomized. Each trial stimulus appeared as black 3D figures on a white background as in Figure 1. Participants indicated whether or not the right-hand object was the same as the left-hand object ('same') or different ('different'). To respond to each object pair, participants pressed either the 'S' key for 'same', or the 'D' key for 'different'. The task was not speeded.

Fig. 1 Two mental rotation objects: the one on the right is *not* a rotated version of the one on the left. The correct response in this trial is therefore 'different'.

3 Results

A one-tailed independent samples t-test was conducted to compare mental rotation error rate between the two groups. As predicted, the caffeine group had a significantly higher error rate than the placebo group: $t(98) = 2, p = .024, d = 0.40$. The data are shown in Figure 2.

Fig. 2 The boxplots and histograms show the mental rotation error rates for the two experimental conditions **after caffeine administration**.

We aim to measure your direct or 'primary' response to the scenario on the next page. To that end, please attempt to answer the question on the following page without thinking at length about it. You may use any of the information given on the next page to answer the question.

Author Profile:

A researcher with whom you have collaborated previously (and with whom you would collaborate again) has recently conducted a study. Prior to conducting this study, a detailed sampling plan, analysis strategy, and the study hypotheses were posted publicly on the author's OSF page.

An excerpt of the relevant study information is given on the next page.

1 Introduction

We aimed to investigate the relationship between caffeine and visual representation in the brain. We hypothesized that participants given a high dosage of caffeine would make more errors in the mental rotation task, compared with participants administered a placebo.

2 Methods

2.1 Participants

The experimental sample comprised 100 participants— 52 females (mean age = 26.5) and 48 males (mean age = 25) that had been recruited from the general public in London, UK. Normal or corrected-to-normal vision and full use of both hands was a requirement for participation. Participants with a known allergy to caffeine were excluded, as were participants who reported drinking an excessive amount of coffee daily (> 4 cups per day).

2.2 Design

To ensure gender balance in the experiment, males and females were separately randomly assigned to the 'caffeine' and 'placebo' conditions: both conditions contained 26 females and 24 males. The caffeine group were given a 'high-dose' (400mg) caffeine tablet with a 8-ounce glass of water 45 minutes before the mental rotation task, while the placebo group were given a placebo pill with an 8-ounce glass of water 45 minutes before the task.

The experiment was double-blind, and the pills administered were visually indistinguishable. The dependent variable was error rate on a mental rotation task, quantified as total percentage incorrect in 40 trials.

2.3 Materials

In each of the 40 trials, two objects were presented alongside one another on a computer monitor (display dimensions: 336.3mm x 597.9mm) in one of 8 rotation angles (0°, 45°, 90°, etc). The trial presentation order was randomized. Each trial stimulus appeared as black 3D figures on a white background as in Figure 1. Participants indicated whether or not the right-hand object was the same as the left-hand object ('same') or different ('different'). To respond to each object pair, participants pressed either the 'S' key for 'same', or the 'D' key for 'different'. The task was not speeded.

Fig. 1 Two mental rotation objects: the one on the right is *not* a rotated version of the one on the left. The correct response in this trial is therefore 'different'.

3 Results

A one-tailed independent samples t-test was conducted to compare mental rotation error rate between the two groups. As predicted, the caffeine group had a significantly higher error rate than the placebo group: $t(98) = 2, p = .024, d = 0.40$. The data are shown in Figure 2.

Fig. 2 The boxplots and histograms show the mental rotation error rates for the two experimental conditions **after caffeine administration**.

We aim to measure your direct or 'primary' response to the scenario on the next page. To that end, please attempt to answer the question on the following page without thinking at length about it. You may use any of the information given on the next page to answer the question.

Author Profile:

A researcher with whom you have never collaborated (and with whom you are unlikely to collaborate in the future) has recently conducted a study. Prior to conducting this study, a detailed sampling plan, analysis strategy, and the study hypotheses were posted publicly on the author's OSF page.

An excerpt of the relevant study information is given on the next page.

1 Introduction

We aimed to investigate the relationship between caffeine and visual representation in the brain. We hypothesized that participants given a high dosage of caffeine would make more errors in the mental rotation task, compared with participants administered a placebo.

2 Methods

2.1 Participants

The experimental sample comprised 100 participants— 52 females (mean age = 26.5) and 48 males (mean age = 25) that had been recruited from the general public in London, UK. Normal or corrected-to-normal vision and full use of both hands was a requirement for participation. Participants with a known allergy to caffeine were excluded, as were participants who reported drinking an excessive amount of coffee daily (> 4 cups per day).

2.2 Design

To ensure gender balance in the experiment, males and females were separately randomly assigned to the 'caffeine' and 'placebo' conditions: both conditions contained 26 females and 24 males. The caffeine group were given a 'high-dose' (400mg) caffeine tablet with a 8-ounce glass of water 45 minutes before the mental rotation task, while the placebo group were given a placebo pill with an 8-ounce glass of water 45 minutes before the task.

The experiment was double-blind, and the pills administered were visually indistinguishable. The dependent variable was error rate on a mental rotation task, quantified as total percentage incorrect in 40 trials.

2.3 Materials

In each of the 40 trials, two objects were presented alongside one another on a computer monitor (display dimensions: 336.3mm x 597.9mm) in one of 8 rotation angles (0°, 45°, 90°, etc). The trial presentation order was randomized. Each trial stimulus appeared as black 3D figures on a white background as in Figure 1. Participants indicated whether or not the right-hand object was the same as the left-hand object ('same') or different ('different'). To respond to each object pair, participants pressed either the 'S' key for 'same', or the 'D' key for 'different'. The task was not speeded.

Fig. 1 Two mental rotation objects: the one on the right is *not* a rotated version of the one on the left. The correct response in this trial is therefore 'different'.

3 Results

A one-tailed independent samples t-test was conducted to compare mental rotation error rate between the two groups. As predicted, the caffeine group had a significantly higher error rate than the placebo group: $t(98) = 2, p = .024, d = 0.40$. The data are shown in Figure 2.

Fig. 2 The boxplots and histograms show the mental rotation error rates for the two experimental conditions **after caffeine administration**.

We aim to measure your direct or 'primary' response to the scenario on the next page. To that end, please attempt to answer the question on the following page without thinking at length about it. You may use any of the information given on the next page to answer the question.

Author Profile:

A researcher with whom you have collaborated previously (and with whom you would collaborate again) has recently conducted a study. Prior to conducting the study, its protocol was peer-reviewed and conditionally accepted by the editorial committee at the publishing journal.

An excerpt of the relevant study information is given on the next page.

1 Introduction

We aimed to investigate the relationship between caffeine and visual representation in the brain. We hypothesized that participants given a high dosage of caffeine would make more errors in the mental rotation task, compared with participants administered a placebo.

2 Methods

2.1 Participants

The experimental sample comprised 100 participants— 52 females (mean age = 26.5) and 48 males (mean age = 25) that had been recruited from the general public in London, UK. Normal or corrected-to-normal vision and full use of both hands was a requirement for participation. Participants with a known allergy to caffeine were excluded, as were participants who reported drinking an excessive amount of coffee daily (> 4 cups per day).

2.2 Design

To ensure gender balance in the experiment, males and females were separately randomly assigned to the 'caffeine' and 'placebo' conditions: both conditions contained 26 females and 24 males. The caffeine group were given a 'high-dose' (400mg) caffeine tablet with a 8-ounce glass of water 45 minutes before the mental rotation task, while the placebo group were given a placebo pill with an 8-ounce glass of water 45 minutes before the task.

The experiment was double-blind, and the pills administered were visually indistinguishable. The dependent variable was error rate on a mental rotation task, quantified as total percentage incorrect in 40 trials.

2.3 Materials

In each of the 40 trials, two objects were presented alongside one another on a computer monitor (display dimensions: 336.3mm x 597.9mm) in one of 8 rotation angles (0°, 45°, 90°, etc). The trial presentation order was randomized. Each trial stimulus appeared as black 3D figures on a white background as in Figure 1. Participants indicated whether or not the right-hand object was the same as the left-hand object ('same') or different ('different'). To respond to each object pair, participants pressed either the 'S' key for 'same', or the 'D' key for 'different'. The task was not speeded.

Fig. 1 Two mental rotation objects: the one on the right is *not* a rotated version of the one on the left. The correct response in this trial is therefore 'different'.

3 Results

A one-tailed independent samples t-test was conducted to compare mental rotation error rate between the two groups. As predicted, the caffeine group had a significantly higher error rate than the placebo group: $t(98) = 2, p = .024, d = 0.40$. The data are shown in Figure 2.

Fig. 2 The boxplots and histograms show the mental rotation error rates for the two experimental conditions **after caffeine administration**.

We aim to measure your direct or 'primary' response to the scenario on the next page. To that end, please attempt to answer the question on the following page without thinking at length about it. You may use any of the information given on the next page to answer the question.

Author Profile:

A researcher with whom you have never collaborated (and with whom you are unlikely to collaborate in the future) has recently conducted a study. Prior to conducting the study, its protocol was peer-reviewed and conditionally accepted by the editorial committee at the publishing journal.

An excerpt of the relevant study information is given on the next page.

1 Introduction

We aimed to investigate the relationship between caffeine and visual representation in the brain. We hypothesized that participants given a high dosage of caffeine would make more errors in the mental rotation task, compared with participants administered a placebo.

2 Methods

2.1 Participants

The experimental sample comprised 100 participants— 52 females (mean age = 26.5) and 48 males (mean age = 25) that had been recruited from the general public in London, UK. Normal or corrected-to-normal vision and full use of both hands was a requirement for participation. Participants with a known allergy to caffeine were excluded, as were participants who reported drinking an excessive amount of coffee daily (> 4 cups per day).

2.2 Design

To ensure gender balance in the experiment, males and females were separately randomly assigned to the 'caffeine' and 'placebo' conditions: both conditions contained 26 females and 24 males. The caffeine group were given a 'high-dose' (400mg) caffeine tablet with a 8-ounce glass of water 45 minutes before the mental rotation task, while the placebo group were given a placebo pill with an 8-ounce glass of water 45 minutes before the task.

The experiment was double-blind, and the pills administered were visually indistinguishable. The dependent variable was error rate on a mental rotation task, quantified as total percentage incorrect in 40 trials.

2.3 Materials

In each of the 40 trials, two objects were presented alongside one another on a computer monitor (display dimensions: 336.3mm x 597.9mm) in one of 8 rotation angles (0°, 45°, 90°, etc). The trial presentation order was randomized. Each trial stimulus appeared as black 3D figures on a white background as in Figure 1. Participants indicated whether or not the right-hand object was the same as the left-hand object ('same') or different ('different'). To respond to each object pair, participants pressed either the 'S' key for 'same', or the 'D' key for 'different'. The task was not speeded.

Fig. 1 Two mental rotation objects: the one on the right is *not* a rotated version of the one on the left. The correct response in this trial is therefore 'different'.

3 Results

A one-tailed independent samples t-test was conducted to compare mental rotation error rate between the two groups. As predicted, the caffeine group had a significantly higher error rate than the placebo group: $t(98) = 2, p = .024, d = 0.40$. The data are shown in Figure 2.

Fig. 2 The boxplots and histograms show the mental rotation error rates for the two experimental conditions **after caffeine administration**.

Appendix C

Comments to authors

Please comment explicitly on each of the following points in your comments to the authors

- **The significance of the research question(s)**

No concerns.

- **The logic, rationale, and plausibility of the proposed hypotheses**

Page 10. Line 52: It would be helpful to see an example version of this discussion now, as in-principle acceptance is given before reviewers can see the final manuscript. I am not familiar enough with the RR review format to know if this is considered a major issue however. I will leave it to the editor to judge whether discussion points for the final manuscript can be “preregistered” at stage 1.

- **The soundness and feasibility of the methodology and analysis pipeline (including statistical power analysis where applicable)**

Page 11. Line 17-23: I may have been too imprecise here. My concern is that the RR conditions are the only ones that assure the participants that some peer review has occurred. Since no condition specifies whether the paper is published, participants could interpret all conditions as examples of an *unpublished* study. If so, a difference in trust may arise from the fact that participants care about whether a study has been peer reviewed at all, regardless of whether this peer review happens before or after data collection.

I appreciate that the preregistration and registered report formats should increase trust in results even if they do not end up published. I also note that your hypotheses does not concern published research findings per se, so I do not view this concern as critical for the ability of your design to address the stated hypotheses. However, I worry that the current wording could inflate trust in the RR condition relative to the prereg and no-prereg condition, simply because more features of the normal review process are guaranteed in your RR condition as currently stated. This would decrease the study’s ability to draw conclusions about trust in the real-world published literature, where both RR and non-RR studies normally are peer-reviewed. I therefore recommend specifying in each condition that the study in question has been published.

Page 11. Line 33-41: The change in description from personal website to OSF page is much appreciated. I would now only recommend the authors to consider whether every participant will understand what the abbreviation “OSF” refers to.

I appreciate the added complexity that the Radical Randomization method would impose on the study design and analyses, and I do not think it is critical to implement. However, I do think that this kind of experiment would be an excellent use-case for the method, as there are many ways you could phrase the scenarios in each individual condition. I would be very curious to see how stable reported trust would be across several operationalizations of the conditions. This would probably influence power somewhat (assuming that the effect would actually vary across phrasings), but given the precision you are currently aiming for, I would seriously consider trading some precision to get more information about the potential for variability in this research design.

- **Whether the clarity and degree of methodological detail would be sufficient to replicate exactly the proposed experimental procedures and analysis pipeline**

All concerns have been sufficiently dealt with.

- **Whether the authors provide a sufficiently clear and detailed description of the methods to prevent undisclosed flexibility in the experimental procedures or analysis pipeline**

Page 14. Line 20: I appreciate the restatement of the hypotheses in the analysis plan section. However, here I would recommend the hypotheses be more explicitly related to the measured variables and to the specific statistical results expected (e.g. “we expect a main effect of imagined familiarity with the study author, where imagined familiarity should increase self-reported trust [Expected inclusion $BF_{10} > 10$]”).

I still recommend that the research questions/hypotheses be put under a separate sub-heading, and structured as a numbered list. Ideally, a reader should only need a quick visual scan to locate the sections containing the proposed hypotheses, planned statistical tests, results and conclusions.

- **Whether the authors have considered sufficient outcome-neutral conditions (e.g. positive controls) for ensuring that the results obtained are able to test the stated hypotheses**

All concerns have been sufficiently dealt with.

- **Other major comments**

After completing the main experimental task, participants should be explicitly asked to guess the research hypothesis!

- **Minor comments**
 - The new phrasing of the "unfamiliar" condition (“... *and with whom you are unlikely to collaborate in the future*”) could be interpreted as being about whether you like the researcher or not.
 - The response options to the survey question "*what is your opinion on these protocols*" seem overly positively charged. I also miss a specific response option for people who think RRs are a bad initiative.
 - Page 33. Line 51-56. This section is phrased rather confusingly, and could benefit from some revision: "*Familiarity is a nuanced and continuous construct and is seen to varying extents in the academic setting. For instance, one is familiar with someone in their department with whom they regularly collaborate, and may even see in social settings. In contrast, one is familiar with someone in that they have talk with them at an annual conference, or have read several of their articles.*"

Appendix D

Dear Mr. Dunn,

My coauthors and I are pleased to submit the requested minor revisions on our registered report: “The Effect of Preregistration on Trust in Empirical Research Findings: A Registered Report Proposal”. Once again, we thank the reviewers for the time and care taken with reviewing our work. As with our previous resubmission, we have done the best we can to respond thoroughly to their comments.

We copied the texts provided by reviewers below. Please find our responses to each comment in bold.

Warm regards,

Sarahanne M. Field,

And on behalf of E.-J. Wagenmakers, Rink Hoekstra, Anja Ernst, Henk Kiers and Don van Ravenzwaaij.

Associate Editor

Comments to the Author:

The revised manuscript was returned to the three of the original four reviewers (one reviewer was unavailable but will return at Stage 2 in the event of the manuscript achieving IPA). All reviewers viewed the revision favourably but highlight a number of areas that would benefit from further improvement, chiefly with the methodology but also regarding structure (e.g. Reviewer 2's suggestion to present the hypotheses in list format, which I endorse), clarity of hypotheses (e.g. Reviewer 1's query regarding the nature of the predicted interaction) and conceptual framing (e.g. Reviewer 4's concern about the issue of trust vs quality).

Provided all remaining points are addressed comprehensively in a revised submission, Stage 1 IPA should be forthcoming without requiring further in-depth review.

Reviewer 1.

Kudos to the authors for being responsive to most of the concerns raised by the reviewers. I think the proposed study is much stronger and better fulfills the requirements of a registered report.

However, I have two remaining concerns, one major and one minor.

Major concern: I still don't have a sense of whether the authors are predicting that familiarity will be an additive or interactive factor – and why.

In response to my previous concern, the authors write (in their response to reviewers):

“Our group had discussed that familiarity should be included because it is a very strong influence on trust by virtue of the recognition heuristic mentioned in the manuscript, however there is an extent to which the expectation of how it would behave in conjunction with the preregistration variable was unknown. At the outset, we believed it should be additive, however recognize that in cases where a personal interaction leading to familiarity is negative (e.g., in the case where one co-author suspects another of poor research practice), then the effect of familiarity would behave differently.”

Alas, that response doesn't answer the question of whether the authors predict that manipulating the

familiarity variable will add to or interact with the other variable the authors propose to manipulate (i.e., preregistration status).

Similarly, in response to Reviewer 3, the authors write (in their response to reviewers):

“Yes, indeed this is a hint for an interaction effect. We expected that familiarity would have an additive effect on trust that is, it would increase trust ratings in conjunction with preregistration status even more than preregistration status alone would cause. We see that our descriptions of the interaction and the variable in general are not sufficiently clear, and have amended the manuscript to fix this issue in several different places in pages 5/6 and 16.”

Again, this response doesn't really answer the question and seems to confuse interaction with additivity (“yes, indeed this is a hint for an interaction effect” but then “we expected that familiarity would have an additive effect on trust”).

Similarly, in the manuscript, the authors write, “The results in Figure 1 reveal that familiarity does not increase trust means for unregistered studies, but does increase trust means for PR and RR studies.” Thus, the authors are describing an interactive effect, not an additive effect.

If the authors are considering familiarity a predicted rather than an exploratory variable, then they need to predict whether the effect of this variable will be additive or interactive with the effect of the other predicted variable. Just to be clear, I'm using the terms additive and interactive in the same way that classic experimental psychology does (e.g., http://www.psychwiki.com/wiki/What_is_an_Interaction%3F), because the design of the proposed study (a 3 x 2 factorial) appears to be a classic experimental psychology design.

We apologize for the confusion our response caused. Thank you for clarifying your request for us.

We make two points here: First- all we expect is that familiarity will enhance trust ratings to some extent for each level of PR/RR. The helpful interactions with the reviewers made clear to us that we do not have specific predictions about the behavior of the familiarity variable. As such, we now consider familiarity an exploratory variable. This is now clear in the manuscript body text, as well as in the abstract.

Second- the results in Figure 1 do show an ordinal interaction. Because we have changed the materials in response to participant feedback in the pilot study, we do not necessarily anticipate the same pattern of results.

The minor concern: I think the authors might have misunderstood why I recommended that they better occlude the manipulation check question (with additional questions). The authors seem to believe better occlusion isn't necessary because “the manipulation checks are done at the end of the experiment.” Yes, indeed; that's usually when manipulation checks are conducted. But, alas, that's not the sole reason for occluding manipulation checks. Another reason is to occlude the purpose of the study until all data have been collected. Participants talk to other participants – often before all participants have participated. This type of participant-to-participant contamination used to be a huge problem with undergraduate participant pools. But it can also be a problem with current day social media and even old-fashioned water-cooler talk.

We now understand your recommendation. Indeed, it is important to ensure that no contamination of the participant pool occurs. In any case, there are now a few more questions that probe for the participants' knowledge about the PR and RR protocols that are asked, which appear after the study with the manipulation check questions. This should help occlude the

manipulation checks. Additionally, we will randomize the order of all of these questions (there are now 8 in total for each person, including the manipulation checks) in the final study, which should prevent people from being too suspicious, as they might if the manipulation checks were directly after the primary question about trust.

Reviewer 2.

Most of my concerns have been adequately addressed, and my outstanding concerns should be relatively easy for the authors to address/implement. I therefore recommend the submission be accepted after some minor revisions.

- The significance of the research question(s)

No concerns.

- The logic, rationale, and plausibility of the proposed hypotheses

Page 10. Line 52: It would be helpful to see an example version of this discussion now, as in-principle acceptance is given before reviewers can see the final manuscript. I am not familiar enough with the RR review format to know if this is considered a major issue however. I will leave it to the editor to judge whether discussion points for the final manuscript can be “preregistered” at stage 1.

We are happy with the reviewer's suggestion to let the editor decide whether or not we should preregister our discussion point, but are willing to go on the record that we will discuss the fact that authors might deviate from their own preregistration documents and that this is not always easy to verify in the Discussion section of our eventual manuscript.

- The soundness and feasibility of the methodology and analysis pipeline (including statistical power analysis where applicable)

Page 11. Line 17-23: I may have been too imprecise here. My concern is that the RR conditions are the only ones that assure the participants that some peer review has occurred. Since no condition specifies whether the paper is published, participants could interpret all conditions as examples of an unpublished study. If so, a difference in trust may arise from the fact that participants care about whether a study has been peer reviewed at all, regardless of whether this peer review happens before or after data collection.

I appreciate that the preregistration and registered report formats should increase trust in results even if they do not end up published. I also note that your hypotheses does not concern published research findings per se, so I do not view this concern as critical for the ability of your design to address the stated hypotheses. However, I worry that the current wording could inflate trust in the RR condition relative to the prereg and no-prereg condition, simply because more features of the normal review process are guaranteed in your RR condition as currently stated. This would decrease the study's ability to draw conclusions about trust in the real-world published literature, where both RR and non-RR studies normally are peer-reviewed. I therefore recommend specifying in each condition that the study in question has been published.

Your concerns about this are now clear- thank you for this extra explanation. Indeed, we can see how this issue you raise could confound the results. We now describe the scenarios more consistently. Below are the changes we have made to the manuscript manipulation texts, in italics:

...a participant in the none/unfamiliar condition would receive the following author profile text:

“A researcher with whom you have never collaborated (and with whom you are unlikely to collaborate in the future) has recently conducted a study. *The study was subsequently published in a peer-reviewed journal.* The paper makes no mention of any previously documented sampling plan or study design.”

In the none/familiar condition, the author profile text becomes:

“A researcher with whom you have collaborated previously (and with whom you would collaborate again) has recently conducted a study. *The study was subsequently published in a peer-reviewed journal.* The paper makes no mention of any previously documented sampling plan or study design.”

All receive either the unfamiliar or familiar condition text as described above, depending on which condition they are assigned to. The difference between none, PR and RR condition texts is as follows. A participant in the PR/unfamiliar condition would receive:

“A researcher with whom you have never collaborated (and with whom you are unlikely to collaborate in the future) has recently conducted a study. *The study was subsequently published in a peer-reviewed journal.* Prior to conducting this study, a detailed sampling plan, analysis strategy, and the study hypotheses were posted publicly on the author’s Open Science Framework (OSF) page.”

For those in the RR/familiar condition, this text is presented instead:

“A researcher with whom you have collaborated previously (and with whom you would collaborate again) has recently conducted a study. Prior to conducting the study, its protocol was peer-reviewed and conditionally accepted by the editorial committee at the publishing journal. *The study was subsequently published.*”

Page 11. Line 33-41: The change in description from personal website to OSF page is much appreciated. I would now only recommend the authors to consider whether every participant will understand what the abbreviation “OSF” refers to.

We acknowledge this possibility. We have expanded the acronym in the text now (as seen in the penultimate manipulation text copied above). Additionally, we have added a question that will appear after the experimental question about whether participants know about the OSF.

I appreciate the added complexity that the Radical Randomization method would impose on the study design and analyses, and I do not think it is critical to implement. However, I do think that this kind of experiment would be an excellent use-case for the method, as there are many ways you could phrase the scenarios in each individual condition. I would be very curious to see how stable reported trust would be across several operationalizations of the conditions. This would probably influence power somewhat (assuming that the effect would actually vary across phrasings), but given the precision you are currently aiming for, I would seriously consider trading some precision to get more information about the potential for variability in this research design.

The reviewer may be aware that one of the co-authors on the present proposal, DvR, is a co-author on the radical randomization paper. It is a powerful method for detecting how different nuances in the operationalization can affect the presence and magnitude of the main effect. However, the method relies on each participant carrying out several micro-experiments. Put differently, the method depends on manipulating facets within-subjects. As also indicated in our

response to the final reviewer, we worry that by asking participants too many questions probing the same construct, we might make them inadvertently aware of the intent of the study.

- Whether the clarity and degree of methodological detail would be sufficient to replicate exactly the proposed experimental procedures and analysis pipeline

All concerns have been sufficiently dealt with.

- Whether the authors provide a sufficiently clear and detailed description of the methods to prevent undisclosed flexibility in the experimental procedures or analysis pipeline

Page 14. Line 20: I appreciate the restatement of the hypotheses in the analysis plan section. However, here I would recommend the hypotheses be more explicitly related to the measured variables and to the specific statistical results expected (e.g. “we expect a main effect of imagined familiarity with the study author, where imagined familiarity should increase self-reported trust [Expected inclusion $BF_{10} > 10$]”).

We acknowledge that being so specific can be useful. We have now numerically specified our expectations in terms of Bayes factors in the section you mention.

I still recommend that the research questions/hypotheses be put under a separate sub- heading, and structured as a numbered list. Ideally, a reader should only need a quick visual scan to locate the sections containing the proposed hypotheses, planned statistical tests, results and conclusions.

We agree. A specific section has now been created at the end of the introduction dedicated to research questions and hypotheses, and each hypothesis is numbered. This numbering is reflected in the analysis plan section also, to improve structure and readability. When the results section is written, the same numbering will also be used.

- Whether the authors have considered sufficient outcome-neutral conditions (e.g. positive controls) for ensuring that the results obtained are able to test the stated hypotheses

All concerns have been sufficiently dealt with.

- Other major comments

After completing the main experimental task, participants should be explicitly asked to guess the research hypothesis!

Great suggestion. We will include this as an item in the follow-up question list.

- Minor comments
 - The new phrasing of the "unfamiliar" condition (“ and with whom you are unlikely to collaborate in the future”) could be interpreted as being about whether you like the researcher or not.

Wording of any crucial manipulation text is always tricky, and we have spent much time thinking about what is best (or, at least, least problem-prone). We have changed the particular phrasing you mention in response to another reviewer’s suggestion, and are happy with the new text. Specifically, although we agree the interpretation of liking the researcher is possible, we do not think it necessarily follows from the present formulation.

- The response options to the survey question "what is your opinion on these protocols" seem overly positively charged. I also miss a specific response option for people who think RRs are a bad initiative.

We agree that it is vital to ensure that response options cover all possibilities. We have changed the response options according to your comment.

- Page 33. Line 51-56. This section is phrased rather confusingly, and could benefit from some revision: “Familiarity is a nuanced and continuous construct and is seen to varying extents in the academic setting. For instance, one is familiar with someone in their department with whom they regularly collaborate, and may even see in social settings. In contrast, one is familiar with someone in that they have talk with them at an annual conference, or have read several of their articles.”

We have attempted to make this explanation clearer in the manuscript:

“Familiarity is a nuanced and continuous construct. In academic settings, familiarity with another researcher can take many forms, and is seen to varying extents. For instance, in one case, Researcher A may be familiar with Researcher B in their department because they regularly collaborate. Such colleagues may even socialize with one another, and thus, probably be very familiar with one another, to the extent where they may even call one another a friend. In contrast, Researcher A may not even have heard Researcher C's name, even if they work in the same field of research. If we contrast the first and second cases, we would expect Researcher A to trust the work of Researcher B in the first case more than that of Researcher C, due to the familiarity of Researcher A with Researcher B.”

N/A

Reviewer 3.

Reviewer 4.

In my previous review, I raised three major concerns:

1. I requested that the introduction be edited to explain the construct of trust in more detail. In particular, I was concerned that it was unclear that perceptions of trust, rather than actual trustworthiness or research quality, was being assessed in this study. In the revised version, it is now clearer what the authors examine. I found footnote 1 particularly helpful.

2. I asked for more details about the study's methods, and I was happy to see that materials have now been shared on OSF. I looked through the Qualtrics preview and a few of the materials, and it appears things are now clear and well organized on OSF.

Regarding the measurement of trust, I believe the authors may have misunderstood my critique. My critique was not that a single item measure of trust might be missing important facets, but rather that a single item ad hoc measure has unknown reliability and validity. A benefit of keeping the item the same as the pilot study is that you can compare to that study. But you still may want to consider adding a few more additional items. You could ask separate questions about trust, reliability, replicability, etc. Presumably, if those items hang together, you have some evidence for internal consistency of your trust construct. If you were being really thorough, you could separately validate your trust measurement in another study. This is not a “must” for me, but something for you to consider. You could (a) keep things as they are with a single item measure, (b) collect your desired single item, but add additional questions to further probe and understand your construct (this part could be exploratory); you could stick with your single item as your focal DV for your pre-reg, or (c) validate the single item measure in a separate study before doing choice a. The paper will still be interesting if you decide not to do (b) or (c), but we could certainly learn more if you had time and energy for either of those options.

Thank you for clarifying this, we prefer to stick to option (a). Aside from potentially making the analysis needlessly complex and leading to chance capitalization (if we have 10 items, are we

going to fish for the highest Bayes factor?), in our opinion having more items would increase the risk of the participant becoming aware of what we are measuring.

3. I asked you to better separate the question of whether researchers do trust pre-registered (or RR) studies more than regular studies from whether they should. As noted above in point 1, the intro does a better job at this now. Yet, I still found some spots where you appeared to argue that open science advocates think trust should be higher (should in the moral sense, not in terms of hypotheses). For instance, you write “it is the opinion of proponents of PR and RR that researchers’ academic work will be trusted more by other academics when PR or RR have been part of the research process” As evidence, you cite a 2013 joint letter, but that letter doesn’t talk about trust at all (in spite of the Guardian’s headline). I am uncertain if there are many who would argue that higher trust, in and of itself, is the goal. As we discussed before and agree on, the goal is higher quality work. If further revisions are required, I would ask that you continue to pay attention to the distinction between trust and quality (these are separable goals and some OS proponents might not have higher trust as a goal).

We understand your concerns. We have further amended the manuscript to more strongly reflect the point you make. For instance, here’s an excerpt of the same area of text you mention, which has been thus edited: “It is thought that researchers will produce higher-quality, more reproducible research. This will ultimately benefit them as scientists, for different reasons. For one thing, it is possible that researchers’ academic work will be trusted more by other academics when PR or RR have been part of the research process...”

Additional notes:

4. I am glad that you removed references to “high-impact” journals that were confounding manipulations.

5. You say that “proponents argue that researchers will increase their chances of getting articles accepted by journals, regardless of whether or not the results obtained favor their hypotheses.” Ideally, this would be true for PR, but I fear that it is not, and is instead an exclusive benefit of RR. I’m not sure I hear people arguing that PR will increase the chance of acceptance of null results (and there are anecdotes of rejections of PR’d nulls; come to think of it, I have personally published a replication that was done PR, not RR. It was still rejected at several journals, because results were null, before we went to an OS friendly outlet).

We agree that acceptance of null findings can still be a challenge. Here we merely echo what proponents argue. This being said, we have added a footnote at the point you indicate to flag the possibility of PR users not reaping the same benefits of null-finding acceptance as RR users.

6. Maybe I miss some nuance to the way Bayesian analyses should be described, but is it really correct to say one model is “more likely to be true” (p. 9) than another model? Aren’t these always relative comparisons – i.e., one model is more likely than another, but the likelihood of either model relative to “truth” is unknown/unknowable? I hesitate to even raise the critique because I am a novice at best when it comes to Bayes. Something about “has a true effect” (p. 10) and the aforementioned phrase felt off to me.

The statement of one model being more likely to be true than another, given the data, seems to us to indicate exactly that: *relative* plausibility of one model over another. This statement is a fairly standard way of interpreting Bayes factors and does not speak to the absolute likelihood of a model being true.

Appendix E

The Effect of Preregistration on Trust in Empirical Research Findings: Results of a Registered Report

Sarahanne M. Field¹, E.-J. Wagenmakers², Henk A. L. Kiers¹, Rink Hoekstra¹, Anja F. Ernst,¹ and Don van Ravenzwaaij¹
¹Rijksuniversiteit Groningen; ²University of Amsterdam

Corresponding author: Sarahanne M. Field
University of Groningen, Department of Psychometrics and Statistics
E-mail should be sent to s.m.field@rug.nl

Abstract

The crisis of confidence has undermined the trust that researchers place in the findings of their peers. In order to increase trust in research, initiatives such as preregistration have been suggested, which aim to prevent various questionable research practices. As it stands, however, no empirical evidence exists that preregistration does increase perceptions of trust. The picture may be complicated by a researcher's familiarity with the author of the study, regardless of the preregistration status of the research. This registered report presents an empirical assessment of the extent to which preregistration increases the trust of 209 active academics in the reported outcomes, and how familiarity with another researcher influences that trust. Contrary to our expectations, we report ambiguous Bayes factors and conclude that we do not have strong evidence towards answering our research questions. Our findings are presented along with evidence that our manipulations were ineffective for many participants, leading to the exclusion of 86% of the data, and an underpowered design as a consequence. We discuss other limitations and confounds which may explain why the findings of the study deviate from a previously conducted pilot study. We reflect on the benefits of using the registered report submission format in light of our results. The OSF page for this registered report proposal and its pilot can be found here: <http://dx.doi.org/10.17605/OSF.IO/B3K75>

Keywords: Preregistration, registered reporting, trustworthiness, QRP

We are grateful for Angelika Stefan's assistance with conducting a custom Bayes Factor Design Analysis for ANOVA.

The crisis of confidence in psychology has given rise to a wave of doubt within the scientific community: we now question the quality and trustworthiness of several findings upon which researchers have built entire careers, and the trust of our peers is more difficult to earn than it once was [1, 2, 3]. In response to the crisis, in what can be called a ‘methodological revolution’ [4, 5], several new initiatives have been launched which focus on improving the quality of published research findings, and on restoring the trust of the scientific community in that higher-quality research. Preregistration and registered reporting (henceforth PR and RR, respectively) are two such initiatives.

PR is the process in which a researcher articulates her plans for a research project—including study rationale, hypotheses, design, and analysis and sampling plans—before the data are collected and analyzed. A growing number of researchers choose to upload preregistration documents, which lay out these plans, onto sites such as the Open Science Framework (OSF), [AsPredicted.org](https://aspredicted.org), or onto their personal websites. Others choose to send their plans to colleagues on a departmental mailing list [6, 7]. The plans can be made public, be released only to reviewers, or be kept private, depending on the preference of the author. RR extends the preregistration process, involving the peer-review of the preregistration document through a journal, just as in the review process of a complete research report. Once the preregistration plan has been accepted, the study has been accepted in principle, and the researcher can begin conducting the study. Crucially, a journal’s acceptance of the RR comes with outcome-independent publication [8].

A “critical part of urgent wider reform” [9], both PR and RR are explicitly geared toward lending credibility to research findings. They benefit the researcher, the quality of their output, and the integrity of the entire field’s literature. They may lead to increased study reproducibility [10]. The popularity of preregistration and RR appears to be gaining momentum in the scientific community. For many, it is now part of the regular research process [4]. This move is popular with individual researchers and academic publishers alike. In 2013, 80 researchers authored an open letter to *The Guardian*, petitioning others to take up preregistration practice [9]; to date, more than 140 major journals accept RRs. These include *Royal Society Open Science*, *Cortex*, *Perspectives on Psychological Science*, *Attention, Perception, & Psychophysics*, *Nature Human Behavior*, and *Comprehensive Results in Social Psychology* (see <https://cos.io/rr/> for an up-to-date list of participating journals).

The increasing adoption of PR and RR is accompanied by a growing body of literature that argues for its benefits (see [11, 12]). One major benefit is that the author can be transparent about the research process. This lends credibility to research because it allows for the claims contained in that research to be more thoroughly verified (as opposed to unregistered studies) [13]. Another benefit is that they provide researchers with a means of distinguishing between exploratory and confirmatory research. Authors often unwittingly present exploratory research as confirmatory. This is problematic because it heavily influences the validity of the statistical testing procedure, and changes the interpretation of ‘statistical significance’. Hypothesizing after the Results are Known (HARKing; [14]) is a questionable research practice (QRP), in which researchers blur this crucial distinction. As PR and RR protocols require the researcher to register their hypotheses before they collect data, the researcher is inoculated against the effects of their own hindsight and confirmation biases [15].

P-hacking, when researchers influence the data collection or analysis process until a statistically significant result appears, and cherry-picking, in which one chooses to use or report only those dependent variables which were statistically significant, are two more well-known examples of QRP [16]. They undermine the reliability of research findings, in that they produce results that misrepresent the true nature of a study's findings, typically making them look more compelling than they actually are. P-hacking gets counteracted by the explicit description of the sampling plan and the procedure for outlier removal provided by the author. Any deviation from this in the final manuscript has to be justified. Cherry-picking is prevented by specification of the hypotheses and expected results before the data are seen.

The File Drawer phenomenon [17] refers to another common QRP, in which researchers do not write up their null findings for publication. This QRP is similar to publication bias, a QRP perpetrated by academic journals, whereby the results of articles influence whether or not an outlet publishes them. The file drawer problem and publication bias are jointly responsible for the construction of a literature that does not faithfully represent all the research findings that have been produced. RR targets these QRPs specifically: when null results are found, authors will not automatically relegate them to the file drawer, as participating RR journals typically guarantee outcome-independent publication. When they accept RRs, journals commit to outcome-independent publishing, and are usually bound to that agreement, providing the author has adequately adhered to the originally reviewed and accepted study methodology and sampling plan. There are often other stipulations made by the journal, such as making the data public.

Simmons and colleagues [3] state that QRPs are usually committed in good faith. They typically arise from ambiguity as to which methodological and statistical procedures to follow, in conjunction with a desire to report interesting and eye-catching findings, which is fueled by the incentive structures in place for publication. Unfortunately, QRPs are not uncommon: an estimated 94% of researchers admit to having committed at least one QRP (possibly a conservative estimate, see [16]). This high rate is not the preserve of psychological research: a similar report on QRP has recently emerged for ecology and evolution biology research based on data from 807 participants [18].

Further benefits to adopting PR and RR are directly relevant to the researcher's career and academic reputation [19]. It is thought that researchers will produce higher-quality, more reproducible research. This will ultimately benefit them as scientists, for different reasons. It is possible that researchers' academic work will be trusted more by other academics when PR or RR have been part of the research process. By adopting PR and RR, proponents argue that researchers will increase their chances of getting articles accepted by journals, regardless of whether or not the results obtained favor their hypotheses¹. Finally, they may be more confident in trusting the work of colleagues in their own fields if they know that others' work has been preregistered, or is a registered report. In the specific case of RR, authors will benefit from extra review and input on their methodology before they conduct the study. This allows them to save their time and resources for the highest-quality studies.

¹Whether this benefit is actually reaped by those who use PR as well as for those who submit RRs has yet to be established – it is possible that acceptance of null results remains challenging for publications even when they have been subject to preregistration.

The merits of PR and RR are advocated by many researchers worldwide: by high-profile meta-science researchers such as Chambers [20], Wagenmakers [19] and Nosek [4], as well as by the general population of academics. Despite this, it is unclear whether the scientific community at large trusts the quality of the results borne of the PR or RR process more than studies which have not been subject to these processes. Indeed, we are unaware of any published direct empirical assessment. It is vital to establish whether researchers do trust PR and RR protocols more than those unregistered, because without trust, PR and RR will not truly become part of standard practice in psychology, and the quality of our studies may not continue to increase².

While rebuilding the quality of our evidence base is vital, so too is the trust researchers place in other researchers [21]. One's judgment of information credibility is directly related to one's perception of the credibility of its source [22], and it is often the case that researchers must evaluate the quality of a research article from an author (or author's body of work) with whom they are familiar. As such, these elements within the research process—familiarity and trust go hand in hand, and should be evaluated in conjunction with one another. But what is the mechanism that determines the extent to which we trust our academic colleagues, and their research output? This mechanism may be understood through the recognition or familiarity heuristic; a cognitive heuristic that can act to assist us in making judgments about the credibility of information based on the familiarity of its source [23]. Another possible explanatory mechanism may be ingroup bias – we may favour the research output of ingroup others, leading us to trust those findings relatively more than those of outgroup members. In the context of the proposed study, familiarity with a publication's authors is expected to lead to increased trust in the findings of that study. We cannot predict the precise nature of this relationship, however.

Research Questions and Hypotheses

The research questions under investigation were: (1) *does preregistering a research protocol increase the trust a fellow researcher places in the resulting research findings?*, (2) *does familiarity with the author of a research study increase the trust a fellow researcher places in the resulting research?* and (3) *does familiarity combine with preregistration status such that findings which feature both a familiar author and preregistration are maximally trustworthy?*

We expected that PR increases the trust participants have in research results over no preregistration at all, and expected that RR evokes the most trust in participants overall. We had less clear expectations about familiarity: we expected that familiarity increases trust to some extent, relative to no familiarity overall. We also expected familiarity would garner higher trust in conjunction with preregistration, relative to no familiarity.

This article contains the results of our registered report. The methodology we describe was peer reviewed, and subsequently granted 'in principal acceptance'. The experimental materials, participant information statement, and data associated with the current study

²It is important to note that determining whether or not registered findings are actually more credible, due to higher quality and a decreased risk of QRP is not our aim. We are focusing on how researchers *perceive* registered studies. We assume that researchers expect registered studies to be of higher quality than their non-registered counterparts, but we emphasize that the actual credibility of a study is ultimately more important than how it is perceived.

and a previously conducted pilot, described briefly below, are available on the project's OSF page: <http://dx.doi.org/10.17605/OSF.IO/B3K75>.

Pilot Study

Prior to submitting the plan for the full study described in this article, we conducted a pilot study. It featured simpler stimuli, and was conducted on a smaller sample, compared with the full study. The pilot led us to further develop our methods and provided compelling support for our hypotheses, therefore reinforcing our expectations. The pilot strongly supported the first of our hypotheses: our analysis yielded extreme evidence in favor of Preregistration Status having a true effect on trust, given the data. Familiarity did not compellingly influence trust for unregistered studies, however it did increase trust for the preregistration and registered report conditions. Finally, the pilot results showed that the model including the two main effects and their interaction was also strongly supported. The experimental design and analysis for the pilot study are the same as described in the full study. A detailed description of the pilot appears in the Phase 1 registered report proposal document. It is available on the OSF page, along with the materials used, the raw and processed data, the analysis code and other associated documentation: <http://dx.doi.org/10.17605/OSF.IO/B3K75>.

Current Study

We designed this study to attempt to empirically determine the role of preregistration status in influencing perceptions of trust in research findings, and to determine if familiarity with the author of the research reinforces those trust perceptions.

Method

Participants and Recruitment

This study aimed to obtain information from all actively publishing researchers in the broad field of psychology. As a proxy, we used the population of authors from psychology articles on the Web of Science (WoS) database for a period of several years.³

The recruitment procedure we followed is complex due to recruitment for the pilot study, and our later decision to collect extra data for the full study. For transparency, we break down the procedure as follows:

1. The first part of the sample was originally obtained as part of recruitment for the pilot study. A total of 9,996 email addresses were initially extracted from the WoS, from article records between January 2013 and May 2017, and the resulting sample was randomly split into two groups. One group of 4,998 was used in the pilot, and the other for the full study.

2. In addition, we extracted unique addresses from 2011, 2012 and from May 2017 to December 2017. This yielded 3,267 additional email addresses.

These two steps resulted in the sample of researchers contacted in the first sweep of data collection. Via the Qualtrics browser-based survey software suite, we sent a total of

³The original aim was to use only corresponding authors, however in approximately 9 percent of cases, more than one author's email address was given. In these cases, all author email addresses were used.

8,265 emails for the first sweep. Of these, 1,084 emails were bounced due to expired or incorrect email addresses or inbox spam filters.

3. As we describe later, we chose to collect more data, mining the WoS for records in 2009 and 2010. Mining from this time period yielded 2,449 email addresses, which we used to contact further participants. Of the 2,449 emails sent, 611 bounced due to expired or incorrect email addresses or inbox spam filters.

We attempted to contact a total of 10,714 individuals overall, and observed a response rate of just over 6%. The response rate for the first sweep was approximately 6.7%, while the rate for the second was around 4%. The total N for this study (i.e., participants that *completed* the survey) before exclusions is 652. The participants were not remunerated for their participation.

Design

To investigate the influence of preregistration status and familiarity on trust we employed a 2 x 3 ANOVA design. Our two independent variables were thus preregistration status (with three levels— none, PR and RR) and familiarity (with two levels— familiar and unfamiliar). We asked participants: “How much do you trust the results of this study?”, and measured trust through the rating participants entered on a 1-9 Likert-scale.

Materials. Six fictional research scenarios were constructed for this study according to the study’s six design cells, in which preregistration status and familiarity were manipulated. Six conditions resulted from the factorial combination of 3 preregistration statuses (none, PR, RR) and 2 familiarity types (unfamiliar, familiar).

Construct Operationalization.

Independent Variables: Preregistration and Familiarity. In each hypothetical study scenario, preregistration status and familiarity were manipulated in the text. Preregistration status reflected whether or not the fictional study protocol had been preregistered in some way before data collection. In the scenario texts preregistration status was described in terms of the process involved. With the descriptions used, we hoped to broadly capture the main differences between different levels of preregistration: none at all, preregistration, and registered report. To avoid the use of potentially loaded terminology, we described preregistration and registered reporting in terms of their processes.

The familiarity manipulation was intended to simulate a commonly encountered situation in the academic setting, whereby collaboration often coincides with one researcher being familiar with another to the extent that they would possibly look upon their work as being more credible and trustworthy in comparison with the work of a complete stranger. This study features familiarity as an exploratory variable, included as a main effect in the analysis we describe later in this manuscript.

Dependent Variable: Trust. A measurement of trust was taken for each participant once they had read their fictional scenario. The word ‘trust’ was chosen, as it best encompassed the dimensions we wanted to test. We intended trust to capture the judgment people made based on their perceptions of the credibility of the fictional study, but is a broad enough word such that people can interpret it in different ways. We considered that it could be interpreted as ‘valid’, ‘reliable’, ‘reproducible’ or to indicate the robustness of the effect presented in the fictional scenario. We found the possibility of different interpretations of the word ‘trust’ desirable, as people use different heuristics and definitions to

establish a trust judgment when it comes to evaluating the research output of others: we wanted to capture as many of these interpretations as possible.

All scenario texts and a trial of the study as it appeared to participants can be found on the project's OSF page: <http://dx.doi.org/10.17605/OSF.IO/B3K75>.

All participants received the same instructional text:

“We aim to measure your direct or ‘primary’ response to the scenario on the next page. To that end, please attempt to answer the question on the following page without thinking at length about it. You may use any of the information given on the next page to answer the question.”

In addition to this initial text, a participant in the **none/unfamiliar** condition received the following author profile text:

“A researcher with whom you have never collaborated (and with whom you are unlikely to collaborate in the future) has recently conducted a study. The study was subsequently published in a peer-reviewed journal. The paper makes no mention of any previously documented sampling plan or study design.”

In the **none/familiar** condition, the author profile text becomes (changes in bold):

*“A researcher **with whom you have collaborated previously (and with whom you would collaborate again)** has recently conducted a study. The study was subsequently published in a peer-reviewed journal. The paper makes no mention of any previously documented sampling plan or study design.”*

All receive either the unfamiliar or familiar condition text as described above, depending on which condition they are assigned to. The difference between none, PR and RR condition texts is as follows. A participant in the **PR/unfamiliar** condition received:

“A researcher with whom you have never collaborated (and with whom you are unlikely to collaborate in the future) has recently conducted a study. The study was subsequently published in a peer-reviewed journal. Prior to conducting this study, a detailed sampling plan, analysis strategy, and the study hypotheses were posted publicly on the author's Open Science Framework (OSF) page.”

For those in the **RR/familiar** condition, this text was presented instead:

“A researcher with whom you have collaborated previously (and with whom you would collaborate again) has recently conducted a study. Prior to conducting the study, its protocol was peer-reviewed and conditionally accepted by the editorial committee at the publishing journal. The study was subsequently published.”

Once the instruction and author profile has been read, participants were presented with the following scenario text and data plot:

“Introduction We aimed to investigate the relationship between caffeine and visual representation in the brain. We hypothesized that participants given a high dosage of caffeine would make more errors in the mental rotation task, compared with participants administered a placebo.

Methods

Participants. The experimental sample comprised 100 participants— 52 females (mean age = 26.5) and 48 males (mean age = 25) that had been recruited from the general public in London, UK. Normal or corrected-to-normal vision and full use of both hands was a requirement for participation. Participants with a known allergy to caffeine were excluded, as were participants who reported drinking an excessive amount of coffee daily (> 4 cups per day).

Design. To ensure gender balance in the experiment, males and females were separately randomly assigned to the ‘caffeine’ and ‘placebo’ conditions: both conditions contained 26 females and 24 males. The caffeine group were given a ‘high-dose’ (400mg) caffeine tablet with a 8-ounce glass of water 45 minutes before the mental rotation task, while the placebo group were given a placebo pill with an 8-ounce glass of water 45 minutes before the task.

The experiment was double-blind, and the pills administered were visually indistinguishable. The dependent variable was error rate on a mental rotation task, quantified as total percentage incorrect in 40 trials.

Materials

In each of the 40 trials, two objects were presented alongside one another on a computer monitor (display dimensions: 336.3mm x 597.9mm) in one of 8 rotation angles (0°, 45°, 90°, etc). The trial presentation order was randomized. Each trial stimulus appeared as black 3D figures on a white background as in Figure 1. Participants indicated whether or not the right-hand object was the same as the left-hand object (‘same’) or different (‘different’). To respond to each object pair, participants pressed either the ‘S’ key for ‘same’, or the ‘D’ key for ‘different’. The task was not speeded.

Results

A one-tailed independent samples t-test was conducted to compare mental rotation error rate between the two groups. As predicted, the caffeine group had a significantly higher error rate than the placebo group: $t(98) = 2$, $p = .024$, $d = 0.40$. The data are shown in Figure 4.”

Once the participant read the scenario, they clicked to the next page, on which they found the key experimental question:

How much do you trust the results of this study? (Please note that the word ‘trust’ in this question can be interpreted in many ways: ‘reliable’, ‘replicable’, ‘a true effect’, ‘valid’ or ‘high quality’. Any such interpretations of trust are acceptable for the purposes of

Figure 1. The boxplots and histograms show the mental rotation error rates for the two experimental conditions after caffeine administration.

answering the question.)

A sliding scale accompanied this question, which allowed possible responses ranging from 1 “*Not at all*” to 9 “*Completely*”. Responses recorded on the sliding scale numerically captured participants ratings of how trustworthy they thought the results presented to be.

Manipulation Checks and Followup Questions

After participants answered the primary experimental question about trust, we posed two manipulation questions – one for each of the independent variables.

The questions were:

- 1) *In the fictional scenario you just read, was the researcher responsible for the study someone with whom you were familiar?*
- 2) *In the fictional scenario you just read, was there either: a) mention of any previously documented sampling plan or study design? or b) mention of the study protocol being peer-reviewed and conditionally accepted by the publishing journal?*

A participant in any of the three unfamiliar conditions was expected to answer “no” to such a question, while the other three conditions is expected to answer in the affirmative. We also asked participants whether the study had been preregistered in some way, or whether it was the subject of a registered report, or whether neither of these conditions are true. A

participant in either of the none conditions was expected to answer “no”, while participants in the other conditions was expected to indicate that there was either PR or RR present.

The data of people who indicated that they did not notice the manipulations in the scenario they were given, or indicate something unexpected (e.g., familiarity with the study author if they are in the unfamiliar condition) as revealed by the manipulation check, were excluded from the analysis. We describe the exclusion procedure in detail shortly.

At the recommendation of a reviewer in the first phase of the registered report, we included some follow-up questions. These were intended to help occlude the manipulation check questions, and by extension, the purpose of the study, ensuring that people did not discuss the study before data collection was complete, thereby contaminating the data.

Procedure

All participants received an email containing an invitation to participate via the Qualtrics survey suite which included a link to one of the six scenario surveys. The emails sent for initial contact and follow-up can be found at the project’s OSF page: <http://dx.doi.org/10.17605/OSF.IO/B3K75>.

Participants were randomly assigned to presentation of one of the scenarios described in the Materials section by the Qualtrics-programmed randomizer. When a researcher chose to participate, he or she clicked on a link, which directed the browser to one of the six conditions in the survey. In the first screen, participants were presented with the instruction message, and clicked through using a ‘next’ button to the scenario screen, after which they were presented with the primary research question.

Next, the participants were required to move the on-screen slider to indicate their trust judgment on the 1 – 9 scale, and click ‘Next’. The participants were then given the manipulation check questions, and after that, the follow-up questions regarding their opinions about PR and RR. Finally, the participants were thanked for their time, and were given access to the participation information statement about the study’s aims and other relevant information. This statement is available on the project’s OSF page: <https://osf.io/49d2h/4>. The survey was not speeded, but participants were asked not to dwell too much on the trustworthiness question, and instead rely on instinct as trained researchers regarding their trust in the results of the scenario presented.

Participants were not permitted to go back to earlier parts of the experiment once they click the ‘Next’ button. This was to prevent the possibility of going back and changing responses once the aims of the study became more obvious by the follow-up questions. Once data collection was finished, a ‘thank you’ email was sent, which included an explanation of the study aims.

Results

With the exception of the sampling strategy (which we describe below) the full study described in this article followed our Phase 1 registered report proposal precisely. All data, code, and JASP files are available on our OSF page: <https://osf.io/b3k75/>.

⁴During data collection, the participant information statement was only available via the link shown to participants, not via the OSF page.

Table 1

Conditions, expected answer pattern for each manipulation check question (A1 = expected answer to question 1, A2 = expected answer to question 2) , N per group before and after exclusions, total N excluded per condition, with percentage

Condition	A1	A2	Pre-excl. N	Post-excl. N	N excl. (%)
None/Fam	No	Yes	110	44	66 (60)
None/Unfam	No	No	120	84	36 (30)
PR/Fam	Yes	Yes	110	12	98 (89)
PR/Unfam	Yes	No	106	12	94 (89)
RR/Fam	Yes	Yes	115	24	91 (79)
RR/Unfam	Yes	No	92	33	59 (64)

Sampling Strategy Deviation

We had planned to collect data such that each cell in our six-cell design would contain $N = 80$. After collecting data for two weeks (with the planned reminder prompt email sent after one week), we had over 900 responses. At this point, the first author checked how many exclusions would be made, finding that many participants failed the manipulation check to the extent that our minimum N per cell was not met. Hence, data collection stayed open for the remainder of the month as planned; however only four more responses were collected after this point. As planned, the first author conducted a preliminary analysis, and determined that there was not sufficient evidence to either support or contradict our hypotheses, and contacted the *Royal Society Open Science* editorial office. The editorial office confirmed that more data could be collected, and so, as per our registered report proposal, we once again mined WoS, extracted more email addresses and collected a second wave of data. This yielded only 100 more complete data sets.

At the conclusion of the second data collection phase, we had the complete data of 654 participants. After excluding participants who had failed the manipulation checks we set, we were left with a total N of 209.

Exclusions

As explained above, we included manipulation check questions in the survey, and excluded participants from our analysis based on the participants' answers. We asked: "*In the fictional scenario you just read, was the researcher responsible for the study someone with whom you were familiar?*", and "*In the fictional scenario you just read, was there either: a) mention of any previously documented sampling plan or study design? or b) mention of the study protocol being peer-reviewed and conditionally accepted by the publishing journal?*"

We excluded participants who answered incorrectly (i.e., "yes", when "no" was expected, or "no" when "yes" was expected) or answered that they did not notice the manipulation. This resulted in an unexpectedly high number of exclusions, leaving a final sample of 209 participants over our six design cells. Unfortunately, this small sample was distributed very unevenly across the conditions. Table 1 shows this information for each condition.

Figure 2. Trust ratings for the six experimental conditions from the data *before* (plot a) versus *after* (plot b) exclusions. Error bars represent 95% credible intervals (which result from, in this case, 2.5% being cut off from each end of the posterior distribution). Being derived from a uniform prior, the intervals are numerically identical to 95% confidence intervals. Note that the y-axes do not carry the same range.

Confirmatory Analysis

Table 2 shows the group Ns, in addition to the cell means and SDs for the data pre- and post-exclusions.

Table 2

Descriptive statistics for each condition pre- and post-exclusions

Condition	Pre- N	pre- mean(SD)	Post- N	Post- mean(SD)
None/Fam	110	4.845(1.823)	44	4.818(1.846)
None/Unfam	120	4.867(1.796)	84	5.060(1.891)
PR/Fam	110	5.455(1.816)	12	6.167(2.250)
PR/Unfam	106	5.208(1.798)	12	5.333(1.557)
RR/Fam	115	5.357(1.812)	24	5.792(1.774)
RR/Unfam	92	5.087(1.838)	33	5.182(1.911)

It can be seen from Figure 2 that the credible intervals are very wide, and there is no discernible pattern in the plotted mean points other than a weak trend toward the same findings as in the pilot study.

As planned, we conducted a between-subjects Bayesian ANOVA choosing all priors according to default settings in JASP⁵ ([24, 25] using the statistical software JASP ([26, 24,

⁵JASP's Bayesian statistics calculations for ANOVA are based on Rouder and Morey's BayesFactor R

Table 3
Model Comparison

Models	P(M)	P(M data)	BF _M	BF ₁₀
Null model	0.200	0.545	4.787	1.000
Preregistration	0.200	0.282	1.570	0.517
Familiarity*Fam	0.200	0.108	0.485	0.199
PR + Fam	0.200	0.048	0.204	0.089
PR + Fam + PR*Fam	0.200	0.017	0.068	0.031

Table 4
Analysis of Effects - Trust

Effects	P(incl)	P(incl data)	BF _{Incl.}
Preregistration	0.600	0.350	0.359
Familiarity	0.600	0.177	0.143
PR*Fam	0.200	0.017	0.069

25]; and see <https://static.jasp-stats.org/about-bayesian-anova.html>). Table 3 shows the standard Bayesian ANOVA comparisons between individual models. Column $P(M|data)$ shows that after observing the data, the most plausible model of those under consideration is the null model.

Perhaps a more informative way to interpret the results is by means of *inclusion* Bayes factors. Inclusion Bayes factors ($BF_{inclusion}$) allow one to compare all models that include any given predictor (e.g., in this context either preregistration status or familiarity) with all those that do not, to determine the relative strength of evidence for each factor on trustworthiness (the dependent variable), based on the data.⁶ The JASP inclusion Bayes factor analysis yielded ambiguous results, as shown in Table 4. We now describe the results in the context of their support of the hypotheses.

Observing the current data did little to change the odds for a model including the preregistration variable. That is, given our prior distributions on the effect size parameter, there is more support for exclusion of the preregistration variable and the familiarity variable in our model, than for inclusion of them. This said, the absence of any real evidence means that we cannot make claims for anything in terms of the two main variables. We may only conclude that we need more data. Regarding their interaction, however, given our prior distributions over effect size, there is strong support for leaving the interaction term out of the models.

Discussion

This study was designed to assess whether researchers put more trust in the findings of others when these findings have been obtained from a preregistered study or a registered report, as opposed to from a study conducted without any plans stated prior to data collection. We had three questions: (1) *does preregistering a research protocol increase the*

package. We used the default setting for the prior distributions.

⁶A brief worked example of this approach using the data obtained in the pilot study is shown in Appendix A.

trust a fellow researcher places in the resulting research findings?, (2) does familiarity with the author of a research study increase the trust a fellow researcher places in the resulting research? and (3) does familiarity combine with preregistration status such that findings which feature both a familiar author and preregistration are maximally trustworthy?

After conducting a promising pilot study, we sought to answer these questions by administering an experiment in which we presented fictional study vignettes that differed from one another in the level of preregistration. We asked participants to rate their trust of the findings presented in those scenarios. Our aim was to have at least 80 participants per cell satisfy the inclusion criteria (i.e. based on recognized successful manipulation). Unfortunately, a manipulation check revealed that 86% of participants were not reliably manipulated, leading to group Ns of less than 50 in all conditions except for one.

Possibly as a result of this, our analysis showed that the evidence for our main hypotheses was ambiguous (“not worth more than a bare mention” according to [27, Appendix B]). We obtained strong evidence in favor of leaving the interaction between the two independent variables out, however, given that the independent variables do not reflect meaningful results, this result too is unlikely to be substantial.

Limitations

Exclusions. A surprisingly high percentage of the sample failed the manipulation checks we set, leading to 86% of the sample being excluded from the analysis. In hindsight, it would have been useful to conduct a small pilot on the materials after they were developed for the full study, as we did for the original materials in the pilot study; an approach Houtkoop and colleagues [28] have recently used. This may have flagged potential problems with the manipulations early enough such that we could have adjusted our materials to prevent losing so much data.

This being said, it is possible that the manipulations did in fact work for more people than our checks showed, but that the manipulation was so subtle that it influenced some people’s responses without them consciously noticing that influence. If this were the case though, it would be difficult if not impossible to separate those participants from others who were genuinely unsuccessfully manipulated.

Materials. In response to receiving overwhelming suggestions by participants in the pilot study to make the scenario texts more elaborate and realistic, we substantially changed our original materials.⁷ The original materials included a few simple paragraphs describing the results of a study (either a preregistered one, a registered report, or a traditional one with neither) and side-by-side box plots; the current study featured a realistic mini-study vignette. With this change we aimed to facilitate participants’ immersion into the content, as if they were reviewing the study ‘for real’.

The switch to a realistic mini-study vignette may have had an adverse effect on the response pattern, however. Specifically, participants might have glanced over the scenario page, decided that it would take them too much time or effort to respond (either thinking there was too much reading to be done, or that the scenario was too complex for the five

⁷In our pilot, we used open questions to probe why participants might have had trouble answering “how much do you trust these results?” which helped us understand how people interpreted what we had asked them. A content analysis of these responses is available on OSF (<https://osf.io/w6acb/>).

minutes we suggested they spend on it) and given up on completing the study. It may be that the more realistic complex materials are best to test our hypotheses, but that the ‘quick and easy’ five-minute online survey our study features is too simple to capture the effect. It should be noted, however, that the scenario texts were presented to participants *after* the manipulation texts. Therefore, it is unlikely that quick and careless reading by participants directly led to them failing the manipulation check. On the other hand, it is possible that because a large block of text was presented after the manipulations and before the primary experimental question, the information in the manipulations was forgotten by the time the manipulation check questions were presented. This does not mean that the manipulation was ineffective necessarily (because participants may have been manipulated unconsciously), only that our manipulation check is likely to be unreliable.

As discussed earlier, participants were not given a ‘Back’ button in each page of the survey. This was designed to safeguard the dependent variable responses against being contaminated by the questions that appeared after the manipulations and key trust question. This meant that participants were unable to review their scenario or manipulation texts after having seen the trustworthiness question. Although we consider this choice justifiable, it is possible we should have started out saying ‘read the following research scenario, in the end you will be asked to indicate to what extent you trust the results’, such that people would be able to prepare themselves for the eventuality of the question, without giving them the option of going back and contaminating their answers.

Finally, it is possible that explicit mention of the well-known online repository OSF in the preregistration conditions may have increased people’s trust in the findings. Perhaps people attach credibility to individuals who use services like the OSF, because when an author chooses to upload study materials, data and code, they are acting with good faith in a very tangible way. In hindsight, it may have been a good idea to add information about materials and data sharing onto the registered reports conditions explicitly, in addition to describing the registered report process, with in principal acceptance. We are not investigating people’s faith in OSF of course, but it is conceivable that people’s faith in OSF (and people who use it) influenced their faith in the results we presented in the PR conditions.

Sample size. In general, a larger N leads to a more compelling Bayes factor [29]. It is likely, therefore, that whatever the true state of the world is in terms of our predictions (that is, are preregistered/registered report findings *really* perceived as more trustworthy by peers?), this effect would be reflected in the inclusion Bayes factor with greater participant numbers. That is, our current findings do not teach us very much about the true state of the world, given that our N is so small.

Changing Opinions

Support for preregistration and registered reports is growing (over 200 journals now accept the registered report format, for instance: <https://cos.io/rr/#journals>) to the extent where it may become the norm [30]. Recently, there seems to be an increase of discussions in the scientific community on the complexities that surround using preregistration and registered reports. This is another factor that may have influenced peoples’ responses in this study (especially relative to our pilot’s compelling pro-effect findings)⁸.

⁸It does not explain the large amount of exclusions, however.

Twitter hosts discussions about these protocols regularly, demonstrating that researchers are thinking critically about conditions in which preregistration and registered reports are less useful or are inappropriate in some fashion. For instance, there's concern that registered reports/preregistration "dampens exploratory work" [31, 32], restricts authors' freedom [33], or takes too much time to do [34]. Other concerns revolve around the results of articles featuring preregistered plans – people may be hesitant to trust the results because they are unsure of the degree to which authors stuck to their plans. This concern is not entirely without merit. Claesen, Gomes, Tuerlinckx and Vanpaemel recently uploaded a preprint [35] in which they report that none of the 27 studies they scrutinized had completely adhered to their preregistered plans.

We asked participants extra questions to probe their opinions about the protocols and reasons for participation. Of the participants that provided answers to our question about their thoughts on preregistration and registered reports, an overwhelming number (263 of the 545 qualitative responses collected) thought them 'great' and 'useful'. However, 105 indicated that a more complex answer to the question was warranted, and chose to type in their own answer. Several of these answers suggested that the utility of preregistration and registered reports was defined by either the kind of research question, type of study design. One individual stated of preregistration and registered reporting: "... it largely depends on the question and its setting. For straightforward experimental work it is certainly a good idea and useful", while another wrote "Useful in practice when there is a sufficiently focused research Q." A further 104 participants indicated they were neutral about the protocols. Eight people were of the opinion that preregistration and registered reports are bad initiatives. Even a quick look at the qualitative results to the probe reveals that over half of participants have some reservations about the value of preregistration and registered reports. This, however, does not explain why it appears that so many people have failed to register the relevant manipulation information.

Concluding Reflections

The results of this registered study are disappointing in that they are uninformative. This is in sharp contrast to our pilot's results, which were both predicted and compelling. On a more constructive note, however, the experience of conducting this study, and obtaining the findings we did is valuable, providing food for thought on several points. Firstly, the findings being different to what we had expected prompts reflection on what the registered report format offers to good scientific practice. It allows researchers the freedom to focus on several important aspects of studies: the research questions and their theoretical relevance, the quality of the research methodology and its relationship to the hypotheses, and, in terms of the results – what is exploratory and what is confirmatory. We may reflect on these aspects of the experiment without interference from the outcomes (regardless of whether the support the hypotheses or not). We are also free to write up the results of our study faithfully and transparently without fear of rejection by the journal, providing we adhered to our plans (or sufficiently rationalized if we did not). In the case of the current study, we can be transparent about the trouble we had with our sample, our findings, and that we cannot conclude anything from the study. This may then serve as a warning to other researchers who may attempt to study a similar phenomenon.

Although our experiment results failed to meet our expectations, we remain of the

opinion that registered reports and studies using preregistration are of higher quality, and therefore more trustworthy. That being said, the picture is perhaps more complex than our design could capture. It is of course true that people can find ways to ‘hack’ preregistration and registered reporting, and do the wrong thing if they so desire, but our cautiously optimistic view is that, on average, researchers will use these protocols to produce higher-quality science in good faith. That being said, the use of preregistration and registered reporting does not replace critical engagement of researchers with the literature they consume.

Appendix A:
An explanation and hypothetical worked example describing inclusion Bayes factors

As discussed in the results section, using inclusion Bayes factors we can compare all models that include any given factor (e.g., preregistration) with all those that do not, to determine the relative strength of evidence for each factor on trustworthiness, based on the data, by means of the inclusion Bayes factor. This allows us to quantify the relative evidence of a factor affecting the outcome, provided by the data, by taking the ratio of the posterior inclusion odds and the prior inclusion odds. Below is a hypothetical worked example of the calculation of inclusion Bayes factors.

Table 5

Possible models, their prior probabilities, posterior probabilities and the Bayes factors

Effects	P(M)	P(M D)	BF _M	BF ₁₀
1. Null model	.20	.00038	0.00	1.000
2. Preregistration	.20	.812	17.33	2142.34
3. Familiarity	.20	.000093	.00037	0.25
4. Familiarity + Preregistration	.20	.175	0.85	462.60
5. Familiarity * Preregistration	.20	.012	0.05	30.62

In the absence of data, we can assume a priori that each of these five models is equally likely: $p = .2$. Using this information, we can calculate the prior inclusion probability of ‘preregistration’ by adding the prior probabilities of all models that contain the preregistration effect (i.e., models 2, 4 and 5) as $.20 + .20 + .20 = .60$.

Table 6

Effects, their prior probabilities, posterior probabilities and the inclusion Bayes factors

Effects	P(incl)	P(incl D)	BF _{incl}
1. Preregistration	.60	1	1,409.97
2. Familiarity	.60	0.19	0.15
3. Interaction	.20	0.01	0.05

After the data come in, we are able to calculate the posterior inclusion probability. In this example, the posterior probabilities of the three models (2, 4 and 5, in the P(M|D)) are .812, .175 and .012. The posterior inclusion probability of preregistration is therefore $.812 + .175 + .012 = .999$, while the posterior probability of models *without* the effect of preregistration is the sum of the posterior probabilities of models 1 and 3.

Once the prior and posterior inclusion probabilities are obtained, we can calculate BF_{incl}: The change from prior inclusion odds to posterior inclusion odds. The inclusion Bayes factor for the preregistration effect is therefore:

$$\begin{aligned} \text{BF}_{incl} &= \frac{.999 / (.0003792 + .00009315)}{.60 / (1 - .60)} \\ &= 1,409.97 \end{aligned}$$

In this context, having observed the data, the odds for a model including preregistration status have increased by a factor of nearly 1,410. An inclusion Bayes factor can be calculated for the interaction effect between preregistration status and familiarity in the same manner.

References

- [1] Baker M. Is there a reproducibility crisis? Nature 2016; 533: 452–454.
- [2] Pashler H and Wagenmakers EJ. Editors' introduction to the special section on replicability in psychological science a crisis of confidence? Perspectives on Psychological Science 2012; 7: 528–530.
- [3] Simmons JP, Nelson LD and Simonsohn U. False-positive psychology: Undisclosed flexibility in data collection and analysis allows presenting anything as significant. Psychological Science 2011; 22: 1359–1366.
- [4] Nosek BA and Lindsay DS. Preregistration becoming the norm in psychological science. APS Observer 2018, March; 31. URL <https://www.psychologicalscience.org/observer/preregistration-becoming-the-norm-in-psychological-science>.
- [5] Spellman BA, Gilbert EA and Corker KS. Open science. In Wixted J and Wagenmakers EJ (eds.) Stevens' Handbook of Experimental Psychology and Cognitive Neuroscience (4th ed.), Volume 5: Methodology. New York: Wiley, 2018. pp. 729–776.
- [6] Chambers C. Psychology's 'registration revolution'. The Guardian 2014, May 20; URL <https://www.theguardian.com/science/head-quarters/2014/may/20/psychology-registration-revolution>.
- [7] Munafò MR, Nosek BA, Bishop DV et al. A manifesto for reproducible science. Nature Human Behaviour 2017; 1: 1–9.
- [8] Pain E. Register your study as a new publication option. Science 2015, December; 15. URL <https://www.sciencemag.org/careers/2015/12/register-your-study-new-publication-option>.
- [9] Chambers C, Munafò M et al. Trust in science would be improved by study pre-registration. The Guardian 2013, June 5; URL <https://www.theguardian.com/science/blog/2013/jun/05/trust-in-science-study-pre-registration>.
- [10] Chambers CD, Feredoes E, Muthukumaraswamy SD et al. Instead of “playing the game” it is time to change the rules: Registered reports at aims neuroscience and beyond. AIMS Neuroscience 2014; 1: 4–17.
- [11] Alvarez RM. The pros and cons of research preregistration. Oxford University Press Blog 2014, September; 28. URL <http://blog.oup.com/2014/09/pro-con-research-preregistration/>.
- [12] van't Veer AE and Giner-Sorolla R. Pre-registration in social psychology—A discussion and suggested template. Journal of Experimental Social Psychology 2016; 67: 2–12.
- [13] Nosek BA and Lakens D. Registered Reports: A method to increase the credibility of published results. Social Psychology 2014; 45: 137–141.
- [14] Kerr NL. Harking: Hypothesizing after the results are known. Personality and Social Psychology Review 1998; 2: 196–217.

- [15] Nuzzo R. Fooling ourselves. Nature 2015; 526: 182–185.
- [16] John LK, Loewenstein G and Prelec D. Measuring the prevalence of questionable research practices with incentives for truth telling. Psychological Science 2012; 24: 524–532.
- [17] Rosenthal R. The file drawer problem and tolerance for null results. Psychological Bulletin 1979; 86: 638–641.
- [18] Fraser H, Parker T, Nakagawa S et al. Questionable research practices in ecology and evolution. PLOS ONE 2018; 13: 1–16.
- [19] Wagenmakers EJ and Dutilh G. Seven selfish reasons for preregistration. APS Observer 2016, November; 29. URL <https://www.psychologicalscience.org/observer/seven-selfish-reasons-for-preregistration>.
- [20] Chambers CD. Registered Reports: A new publishing initiative at Cortex. Cortex 2013; 49: 609–610.
- [21] Leonelli S. Data interpretation in the digital age. Perspectives on Science 2014; 22: 397–417.
- [22] Metzger MJ and Flanagin AJ. Credibility and trust of information in online environments: The use of cognitive heuristics. Journal of Pragmatics 2013; 59: 210–220.
- [23] Gigerenzer G and Goldstein DG. Betting on one good reason: The Take The Best heuristic. In Gigerenzer G, Todd PM and the ABC Research Group (eds.) Simple heuristics that make us smart. New York: Oxford University Press, 1999. pp. 75–96.
- [24] Rouder JN, Morey RD, Speckman PL et al. Default bayes factors for ANOVA designs. Journal of Mathematical Psychology 2012; 56: 356–374.
- [25] Morey RD, Rouder JN and Jamil T. Computation of Bayes Factors for Common Designs, 2015. URL <http://bayesfactorpcl.r-forge.r-project.org/>.
- [26] JASP Team. JASP (Version 0.10.2)[Computer software], 2019. URL <https://jasp-stats.org/>.
- [27] Jeffreys H. Theory of Probability. Oxford, UK: Oxford University Press, 1961.
- [28] Houtkoop CCMMBDVMNTEWEJ B L. Data sharing in psychology: A survey on barriers and preconditions. Advances in Methods and Practices in Psychological Science 2018; 1: 70–85.
- [29] Bayarri MJ, Berger JO, Forte A et al. Criteria for bayesian model choice with application to variable selection. The Annals of statistics 2012; 40(3): 1550–1577.
- [30] Nosek BA and Lindsay SD. Preregistration becoming the norm in psychological science. APS Observer 2018; 31(3). URL <https://www.psychologicalscience.org/observer/preregistration-becoming-the-norm-in-psychological-science>.

- [31] @VictoriaMousley. (2019, june 14). Definitely! Wasn't much time for follow-up discussion, but some voices of "no" discussed concern about de-incentivising exploratory studies (even after clarification about what pre-registration is and isn't. Most agreed pre-registration is good and has a "time and place." [Twitter post]. URL <https://twitter.com/VictoriaMousley/status/1139573337569583104>.
- [32] Scherer L. My opinion is that preregistration is less useful for exploratory research. go ahead, explore the heck out of your data. and then when you develop a directional hypothesis preregister it for the next study., 2019. URL <https://twitter.com/ldscherer/status/1140101117180694529>.
- [33] @alexmuhl_r. (2019, june 14). Maybe pre-reg some specific parts of PhD, but to do the whole thing could impair it as a training exercise. I'm really not sure, it'd be bad to stifle the development of the little bits and pieces that lead to interesting side work (& potentially other publications) [Twitter post]. URL https://twitter.com/alexmuhl_r/status/1139546949932199937.
- [34] @nichoisa. (2019, june 14). Watching a conference panel now in which a PI asserted that preregistered reports are not possible if a PhD student has to finish in 3 years (prereg generally ok). I'm sceptical of her assertion but maybe those with experience differ on this? [Twitter post]. URL <https://twitter.com/nichoisa/status/1139569547810353153>.
- [35] Claesen A, Gomes SLBT, Tuerlinckx F et al. Preregistration: Comparing dream to reality, 2019. DOI:10.31234/osf.io/d8wex. URL psyarxiv.com/d8wex.

Appendix F

THE EFFECT OF PREREGISTRATION ON TRUST

Stage 2 Registered Report review by Peder M. Isager

Review

The stage 2 draft of this manuscript appears to closely follow the preregistered design and analysis plan outline in stage 1. Unfortunately, the information that can be gained from the collected data is severely reduced due to the high amount of participants that failed the manipulation check. I agree with the authors that it would probably have been a good idea to incorporate a second pilot in order to evaluate manipulation success and allow for further changes to be made to the experimental design.

In fact, I am a bit surprised that the author team decided to go ahead with a second mining of the WoS database after analyzing the manipulation check data for the first batch of responses. I am aware that this was the preregistered sampling strategy, but in this particular case I think it would have been wise to learn from the data and deviate from the preregistered data collection plan. After all, even if the response rate would have been much higher for the second batch, the flawed manipulation would have made any interpretation of the results problematic regardless of the strength of the Bayes factors. My intention here is not to place blame on the authors. Rather, I think the RSOS editorial team should take note of this issue and consider whether more flexible RR format could be offered in cases where follow-up pilot studies are warranted after changes to the experimental design at stage 1 (e.g. the authors could be offered to submit a stage 2 [second pilot] and a stage 3 [main study] version of the report).

Be that as it may, I believe the authors have done a good job of discussing the limitations, potential causes, and lessons learned in light of this unsuspected design flaw. Conclusions drawn from the paper are also suitably cautious. In general I believe the manuscript offers important insights to future researchers who intend to collect data using similar designs.

I have **two major concerns that I believe needs to be addressed before the paper can be accepted for publication (see below)**. I have also noted down a number of minor comments that I recommend the author address, but that I do not consider crucial.

Summary of RSOS reviewer points

- *Whether the data are able to test the authors' proposed hypotheses by passing the approved outcome-neutral criteria (such as absence of floor and ceiling effects or success of positive controls)*

Results from the manipulation checks suggest that the data are unlikely to be suitable to test the authors' hypotheses. This is acknowledged in the article. The authors spends the majority of the results and discussion section wrestling with this issue, and the conclusions presented by the authors are reasonable given the limited information in the data.

- *Whether the Introduction, rationale and stated hypotheses are the same as the approved Stage 1 submission*

The introduction (with the exception of some formatting differences), rationale and stated hypotheses are the same as the approved stage 1 submission.

- *Whether the authors adhered precisely to the registered experimental procedures*

The authors closely adhered to the registered experimental procedures and to the preregistered analysis plan.

- *Where applicable, whether any unregistered exploratory statistical analyses are justified, methodologically sound, and informative*

They are.

- *Whether the authors' conclusions are justified given the data*

They are.

Comments – major concerns:

- Sample size information is unclear throughout the *participants and recruitment* section and needs to be reported more coherently. It appears from the manuscript and the open data that out of a certain number of emails sent out in the main study, a certain number of researchers agreed to participate (over 900?), of which a smaller number completed the full experiment.
 - The total number of responses is only reported as “over 900”. I would prefer to see both the total number of responses and the total number of completed responses reported in the results section.
 - The total number of participants that completed the survey is reported as 652 on page 7, line 15-16, but is then reported as 654 on page 12 line 37.
 - Of the 65[2/4] subjects that completed the survey, 209 seems to have passed the manipulation check. This is reported as 86% data loss from the total sample. However, $(654-209)/654=68\%$, not 86%. Is this simply a reporting error? Or does 86% refer to a different total than 65[2/4]? If the latter, it is currently not clear from the manuscript what that total is.
- As far as I can see, the scenarios used in the main study are not stored as part of the open materials for the main study on OSF, which limits replicability by independent researchers. The only scenarios I could find documented in full were the ones in the pilot study directory. However, these are not the exact same scenarios as the ones used in the main study. I would strongly encourage adding the revised scenarios used in the main study as a separate document in the main study directory on OSF. I would also recommend adding a PDF of screen shots from both the pilot and main study Qualtrics survey to help readers understand how the study was presented to participants. The pilot survey does contain a URL that is intended to link to an example version of the pilot Qualtrics survey. However, this link is non-functional, and redirects to a page that says “Sorry, this survey is not currently active.”

Comments – minor concerns/comments:

- Regarding the data exclusion: Assuming that no one understood the manipulation and everyone answered “yes” or “no” to the manipulation check questions at random, I would still expect about 180 participants to get both manipulation checks right by chance. It seems to me that there is a risk that most of the participants included in the preregistered analysis could still be unaware of the task manipulation. I think it would be wise to at least bring up this problem in the section where the exclusion procedure is discussed, even if the authors believe this to be an unlikely interpretation.
- I suppose we must assume that most of the data included in the pilot study analysis would have failed the manipulation check as well, had it been added. Why does there then seem to be a strong effect of the preregistration manipulation in the pilot model? If all subjects failed to understand the manipulation I would have expected the pilot to yield no effect (for how could there be a manipulation effect if no-one noticed the manipulation?). The authors offers one explanation for this (participants were non-consciously manipulated), but I think it ought to be considered whether the manipulation check itself is problematic. I.e. could it have been stated in such a way that researchers misunderstood the manipulation check questions, even if they did understand the manipulation? If so, perhaps it would be useful to run an exploratory version of the ANOVA, not excluding the failed manipulation check data, since it may not have been valid to exclude these data points based on the current manipulation check.
- As an exploratory analysis, would it be informative to rerun the ANOVA using the pilot study results to inform the priors $P(M)$? It seems reasonable to me to let the pilot data inform conclusions from the main study. On the other hand, perhaps it is unreasonable to use priors based on data for which most participants likely failed the manipulation check?
- The tables at the end of the manuscript have no identifiers and it is not clear which relate to data from the main study dataset and which relate to data from the pilot study. What is the function of these tables?
- Data for pilot study is stored inside the “ethics” subdirectory. This seems quite unintuitive.

Appendix G

Dear Associate Editor,

My coauthors and I are pleased to submit the requested revisions on our registered report manuscript: "The Effect of Preregistration on Trust in Empirical Research Findings: Results of a Registered Report". Once again, we thank the editor and reviewers for the time and care taken with reviewing our work. We have done our best to respond thoroughly to their comments; the manuscript is better after their input.

We copied the texts provided by reviewers below. Please find our responses to each comment in bold.

Warm regards,

Sarahanne M. Field,

And on behalf of E.-J. Wagenmakers, Rink Hoekstra, Anja Ernst, Henk Kiers and Don van Ravenzwaaij.

Comments from the Associate Editor:

Three of the four original reviewers who assessed the Stage 1 manuscript returned to review the Stage 2 submission. The reviews overall indicate a number of concerns that will need to be addressed to achieve full acceptance. All reviewers note with the concern the failure of the manipulation check, with Reviewer 1 outlining a helpful point-by-point list of actions to check that the study was in fact administered correctly (see also Reviewer 4 point 1), and with Reviewers 2 and 4 suggesting additional exploratory analyses. The RR policy does not require authors to conduct additional analyses, but especially given the failure of the manipulation checks, these suggestions are sensible and very well advised. The reviewers also note a potential breach of confidentiality in the OSF data that should be corrected, as well as a range of other revisions to improve structure, clarity and comprehensiveness, both of the manuscript and the materials stored on the OSF.

Given the concerns with manipulation success, Reviewer 2 makes an additional point: "I think the RSOS editorial team should take note of this issue and consider whether more flexible RR format could be offered in cases where follow-up pilot studies are warranted after changes to the experimental design at stage 1 (e.g. the authors could be offered to submit a stage 2 [second pilot] and a stage 3 [main study] version of the report)." This point is well made, and should the authors prefer to do so, I am open to them conducting a modified follow-up study as part of an Incremental Registration, and then including both the original study and the follow up study in the current RR. If the authors wish to do so, please contact the journal office and we will take the necessary administrative steps to return the manuscript to Stage 1. However, I am not going to require this, as to do so would undermine the commitment of the journal as part of Stage 1 in-principle acceptance.

We thank the editor for the opportunity to add a second pilot study and main study. After consideration, we have decided not to go down this route. We have cast such a wide net with our original study that we fear we would have a hard time recruiting additional participants that have not already been approached for the first pilot and

main study. In addition, we think it is plausible that people's trust in preregistration has changed compared to a few years ago (when we conducted the pilot study) when it was still novel. We believe it is plausible that either trust and interest has dwindled, or that people's beliefs about preregistration and what it can (and cannot) be used for have changed or at least have become more complex. We have touched on this issue in the discussion and think it a likely (if partial) explanation.

Reviewer: 1

I applaud the authors for assembling their final report when the data are so confusing. The manuscript is an emblem of bravery.

However, I'm quite concerned that so many participants failed the manipulation checks. I think it would be extremely helpful to ensure that

- a) the conditions were correctly assigned to each participant;
- b) the manipulation check questions were correctly assigned to each participant;
- c) Qualtrics randomizer was working as desired;
- d) the data codes were interpreted correctly; and
- e) the manipulation check questions were interpreted correctly.

Therefore, first, I recommend that the authors return to Qualtrics and take precise screenshots of each and every screen of material that was shown to participants in each of the conditions. Examine those screenshots to ensure that the study proceeded as the authors envisioned it would and that the Qualtrics randomizer worked the way the authors envisioned it would (if not, that could easily explain why nearly 90% of the participants in one condition failed the manipulation check -- they received the wrong condition).

Moreover, such screenshots from Qualtrics should be available in the supplementary materials. They are best practice for research transparency.

Before responding to any of the points you make, we would like to highlight that the actual exclusion rate for only the manipulation check is 68%. This is still high, but nowhere near as concerning as the 86% we have been discussing. We apologize for this confusion. We clarify these things now in the manuscript wherever the exclusions were mentioned, and have made a detailed schematic which we include at the end of this letter (it's too big to easily slip into the text here but see Appendix A). We had been discussing the overall exclusion rate (i.e., participants who started but did not finish the survey as well as those excluded for failing the manipulation checks). The response we gave to R2's first major comment might also be interesting for you: "Although only 14% of the 9,019 people contacted provided complete and usable data sets (i.e., we had to scrap 86% of our data), we only had to exclude 68% of the complete datasets. It is good you raise this issue because we now present the issue of exclusions due to manipulation checks as much more serious than it is. We conflate incomplete data sets with those who failed manipulation checks, which is not good. This issue is now completely clear in the manuscript in several places."

As for your points: we agree that it is pertinent to make sure that the study ran the way we had planned for it to. We had previewed it several times (at least one time for each

condition) before collecting data, however we have done this once again as you suggest. The result is that the study did indeed run correctly.

With regard to your suggestion about the Qualtrics screenshots: this is a good idea. We have uploaded a short screen capture video for each condition, which shows exactly what the participants saw. Additionally, we have made a video of the study flow and materials as they have been set up in Qualtrics, so that interested people can look ‘behind the curtain’ and see what was done. Lastly, we have provided a working link to a copy of the Qualtrics survey we used for participants, and a .qsf file for those who have a Qualtrics account.

Second, I recommend that the authors download their data from Qualtrics both as "numeric values" and as "choice text" and compare the two downloads. Although this might seem pedantic, given the high profile recent instances in which data codes were erroneously interpreted (e.g., a JAMA article in which conditions codes were reversed 1 versus 2), this second check is important.

This check too (done prior to submitting our manuscript also, for the same reasons) failed to yield any mistakes in the codes. These files with the numeric and choice texts are also available on OSF for completion’s sake.

Third, I encourage the authors to go through the Qualtrics study numerous times, noting each time what condition they appear to be receiving (from Qualtrics) and then make sure that condition is correctly coded in Qualtrics. Again, this might seem pedantic, but it is important.

We agree this is important, and had done this after exclusions also. We have done it again at your request, and have found nothing that explains the high exclusion rate. We have logged one of these checks in the document Context.docx, that we have also uploaded on OSF.

Fourth, I recommend that the authors ask 10 other native speakers of English to read the manipulation check questions and report their interpretation of them. I found the questions to be worded complexly; therefore, one problem might be that the manipulation check questions were difficult to understand.

I have passed on this request to several people, and have had somewhat consistent responses. Only two of them are native Anglophones. The reason we chose a mixture of people in terms of language to help with this request is that the survey was sent to an international sample, which presumably contains a large proportion of people for whom English is not their native tongue. Although we see why you made this request for English speakers, we feel that a better test of the manipulation questions involves a mixture of languages.

Six of the eight people who responded in time for this response letter felt that the questions were relatively unclear in relation to the content of the survey (note that the two people who felt the questions *were* clear were not the native English-speakers). Several issues were mentioned, but the word ‘familiarity’ in relation to the idea of having collaborated with someone posed some problems. People didn’t automatically link these two things together. Most who had issues with the questions said they had to

read them more than once, and one person said that unless people could click back to the previously displayed text (which was not possible) it would have been hard to answer the questions. Some people seemed to think that the sentences could have been shorter, which would have helped decrease the cognitive load of the questions. It is quite possible that many people did not fully understand the manipulation check questions (and either answered wrongly, or just guessed). We discuss this in the manuscript.

P. 15/16: “A surprisingly high percentage of the sample failed the manipulation checks, leading to exclusion of 68% of the sample. It is possible that people did not understand what we were asking due to complexities in the wording of the questions, or a lack of clarity in the link between the questions and the study materials the questions referred to. These participants may have answered incorrectly as a result (even though they had, in fact, been manipulated successfully) or have just guessed the right answers.”

We also partly rationalize including a non-registered analysis in the results section by mentioning “uncertainty regarding the validity of the manipulation check questions (which we touch on in the discussion section)” (p. 14/15).

The bottom line: I encourage the authors to painstakingly examine every aspect of the study to ensure that there were no undetected (and of course unintentional) mixups. Otherwise, it is hard to explain a near 90% exclusion rate for this type of study -- particularly given that the participants are other researchers.

We agree with the need to scrutinize the study setup and materials, and have made everything possible available online so that others may do the same thing. To complement the relevant files and videos, we provide a lengthy context document which explains everything in detail. Please note that the actual exclusion rate (due to the manipulation check questions) is 68%, not 86%. the 86% includes exclusions due to incomplete surveys (i.e., missing data). We note this in the manuscript clearly.

Also, I noticed that participants' email addresses are included, along with their responses, in one of the files uploaded on OSF. However, according to the ethics consent, "The research results of this study will be treated confidentially and anonymously. Your data will be processed by means of a participant number. This code is disconnected from your personal data." Therefore, participants' email addresses should most likely be removed.

This is a mistake made by the first author. It is certainly a breach of ethics, and the file was removed in mid-December. Thank you for drawing our attention to it. A new file is there now, with only non-sensitive information available.

Reviewer: 2

The stage 2 draft of this manuscript appears to closely follow the preregistered design and analysis plan outline in stage 1. Unfortunately, the information that can be gained from the collected data is severely reduced due to the high amount of participants that failed the manipulation check. I agree with the authors that it would probably have been a good idea to

incorporate a second pilot in order to evaluate manipulation success and allow for further changes to be made to the experimental design.

In fact, I am a bit surprised that the author team decided to go ahead with a second mining of the WoS database after analyzing the manipulation check data for the first batch of responses. I am aware that this was the preregistered sampling strategy, but in this particular case I think it would have been wise to learn from the data and deviate from the preregistered data collection plan. After all, even if the response rate would have been much higher for the second batch, the flawed manipulation would have made any interpretation of the results problematic regardless of the strength of the Bayes factors. My intention here is not to place blame on the authors. Rather, I think the RSOS editorial team should take note of this issue and consider whether more flexible RR format could be offered in cases where follow-up pilot studies are warranted after changes to the experimental design at stage 1 (e.g. the authors could be offered to submit a stage 2 [second pilot] and a stage 3 [main study] version of the report).

Be that as it may, I believe the authors have done a good job of discussing the limitations, potential causes, and lessons learned in light of this unsuspected design flaw. Conclusions drawn from the paper are also suitably cautious. In general I believe the manuscript offers important insights to future researchers who intend to collect data using similar designs.

I have two major concerns that I believe needs to be addressed before the paper can be accepted for publication (see below). I have also noted down a number of minor comments that I recommend the author address, but that I do not consider crucial.

Summary of RSOS reviewer points

- Whether the data are able to test the authors' proposed hypotheses by passing the approved outcome-neutral criteria (such as absence of floor and ceiling effects or success of positive controls)

Results from the manipulation checks suggest that the data are unlikely to be suitable to test the authors' hypotheses. This is acknowledged in the article. The authors spends the majority of the results and discussion section wrestling with this issue, and the conclusions presented by the authors are reasonable given the limited information in the data.

- Whether the Introduction, rationale and stated hypotheses are the same as the approved Stage 1 submission

The introduction (with the exception of some formatting differences), rationale and stated hypotheses are the same as the approved stage 1 submission.

- Whether the authors adhered precisely to the registered experimental procedures

The authors closely adhered to the registered experimental procedures and to the preregistered analysis plan.

- Where applicable, whether any unregistered exploratory statistical analyses are justified, methodologically sound, and informative

They are.

- *Whether the authors' conclusions are justified given the data*

They are.

Comments – major concerns:

- Sample size information is unclear throughout the *participants and recruitment* section and needs to be reported more coherently. It appears from the manuscript and the open data that out of a certain number of emails sent out in the main study, a certain number of researchers agreed to participate (over 900?), of which a smaller number completed the full experiment.

- o The total number of responses is only reported as “over 900”. I would prefer to see both the total number of responses and the total number of completed responses reported in the results section.

- o The total number of participants that completed the survey is reported as 652 on page 7, line 15-16, but is then reported as 654 on page 12 line 37.

- o Of the 65[2/4] subjects that completed the survey, 209 seems to have passed the manipulation check. This is reported as 86% data loss from the total sample. However, $(654-209)/654=68\%$, not 86%. Is this simply a reporting error? Or does 86% refer to a different total than 65[2/4]? If the latter, it is currently not clear from the manuscript what that total is.

We agree that the reporting of the study N (and how N changed from data collection to analysis) is very confusing. With regard to 68 versus 86% - you are both right and wrong about the numbers here. Although only 14% of the 9,019 people contacted provided complete and usable data sets (i.e., we had to scrap 86% of our data), we only had to exclude 68% of the complete datasets. It is good you raise this issue because we now present the issue of exclusions due to manipulation checks as much more serious than it is. We conflate incomplete data sets with those who failed manipulation checks, which is not good. This issue is now completely clear in the manuscript in several places. We also discuss this issue with R1.

Please find the details set out more clearly in the included excerpts, and have made a detailed schematic which is included in the manuscript and at the end of this response letter (Appendix A).

Abstract: “Our findings are presented along with evidence that our manipulations were ineffective for many participants, leading to the exclusion of 68% of complete data sets, and an underpowered design as a consequence.”

P. 5-7: “The recruitment procedure we followed is complex due to recruitment for the pilot study, and our later decision to collect extra data for the full study. For clarity, we have put the recruitment, filtering and exclusion steps into a schematic (Figure 1) which accompanies the following summary.

The first part of the sample was originally obtained as part of recruitment for the pilot study. A total of 9,996 email addresses were initially extracted from the WoS, from article records between January 2013 and May 2017, and the resulting sample was randomly split into two groups. In addition, we extracted 3,267 addresses, updating the N to 8,265. These two steps resulted in the sample of researchers contacted in the first sweep of data collection. Via the Qualtrics browser-based survey software suite, we sent a total of 8,265 emails for the first sweep. Of these, 1,084 emails were bounced due to expired or incorrect email addresses or inbox spam filters. We were left, therefore, with N = 7,181.

As we describe later, we mined the WoS for more records in 2009 and 2010 to help reach our minimum overall N of 480. This yielded 2,449 emails, of which 611 bounced. The total number of mined email addresses over both batches is 10,714. The number of email addresses which received an initial contact email from us is the sum of the two batches of email addresses extracted, meaning that, in total 9,019 people were successfully contacted (i.e., 8.4% of the contact emails sent out were not delivered).

We observed a response rate of just over 6%. The response rate for the first sweep was approximately 6.7%, while the rate for the second was around 4%. This means that the number of people who completed the study *before* exclusions is 654. Further exclusions are made due to participants failing the manipulation checks. We discuss these in the results section.”

P. 12: “At the conclusion of the second data collection phase, we had the complete data of 654 participants. After excluding participants who had failed the manipulation checks, we were left with a total N of 209 meaning that we excluded approximately 68% of the participants. For clarification, please consult Figure 1: a schematic of the changes to the sample from recruitment to the data analysis.”

P. 15: “Our aim was to have at least 80 participants per cell satisfy the inclusion criteria (i.e. based on recognized successful manipulation). Unfortunately, a manipulation check revealed that 68% of participants were not reliably manipulated, leading to group Ns of less than 50 in all conditions except for one.”

“A surprisingly high percentage of the sample failed the manipulation checks, leading to exclusion of 68% of the sample.”

- As far as I can see, the scenarios used in the main study are not stored as part of the open materials for the main study on OSF, which limits replicability by independent researchers. The only scenarios I could find documented in full were the ones in the pilot study directory. However, these are not the exact same scenarios as the ones used in the main study. I would strongly encourage adding the revised scenarios used in the main study as a separate document in the main study directory on OSF. I would also recommend adding a PDF of screen shots from both the pilot and main study Qualtrics survey to help readers understand how the study was presented to participants. The pilot survey does contain a URL that is intended to link to an example version of the pilot Qualtrics survey. However, this link is non-functional, and redirects to a page that says “Sorry, this survey is not currently active.”

In response to your comments and those of the other reviewers, we have provided as much information and material about the experiment as possible on the OSF page. We copy the response we gave to R1 here, for your information:

“We have uploaded a short screen capture video for each condition, which shows exactly what the participants saw. Anyone can now see this. Additionally, we have made a video of the study flow and materials as they have been set up in Qualtrics, so that interested people can look ‘behind the curtain’ and see what was done. Lastly, we have provided a working link to a copy of the Qualtrics survey we used for participants, and a .qsf file for those who have a Qualtrics account.”

Comments – minor concerns/comments:

- Regarding the data exclusion: Assuming that no one understood the manipulation and everyone answered “yes” or “no” to the manipulation check questions at random, I would still expect about 180 participants to get both manipulation checks right by chance. It seems to me that there is a risk that most of the participants included in the preregistered analysis could still be unaware of the task manipulation. I think it would be wise to at least bring up this problem in the section where the exclusion procedure is discussed, even if the authors believe this to be an unlikely interpretation.

We do consider this a possibility, and have included a short discussion of this issue in the manuscript:

P. 14/15: “Given the concerning number of participants who failed the manipulation checks, we consider it possible that people generally did not understand the study even though they answered as ‘expected’ in the checks (perhaps they were guessing in the manipulation check questions and happened to answer correctly). Although the lack of power in the study makes it hard to say anything substantial, it could be that we would not find the anticipated effects even if we had sufficient power.”

- I suppose we must assume that most of the data included in the pilot study analysis would have failed the manipulation check as well, had it been added. Why does there then seem to be a strong effect of the preregistration manipulation in the pilot model? If all subjects failed to understand the manipulation I would have expected the pilot to yield no effect (for how could there be a manipulation effect if no-one noticed the manipulation?). The authors offers one explanation for this (participants were non-consciously manipulated), but I think it ought to be considered whether the manipulation check itself is problematic. I.e. could it have been stated in such a way that researchers misunderstood the manipulation check questions, even if they did understand the manipulation? If so, perhaps it would be useful to run an exploratory version of the ANOVA, not excluding the failed manipulation check data, since it may not have been valid to exclude these data points based on the current manipulation check.

Although we think it’s likely that at least some participants’ views on preregistration have changed since we conducted our pilot (and indeed, since we developed our study hypotheses), which we touch on in our discussion section, we agree with your line of thinking. We have done an exploratory ANOVA on the pre-exclusion data set in JASP. The JASP file is available on the OSF page: <https://osf.io/tpgf6/> along with its ‘excluded-

data' counterpart. The effects for such an exploratory ANOVA are similarly underwhelming.

Here are copies of the JASP effects analysis tables, for your convenience:

Table 1. Confirmatory analysis: Inclusion Bayes factors for the data POST exclusions

Analysis of Effects - Trust

Effects	P(incl)	P(incl data)	BF _{incl}
Preregistration	0.600	0.347	0.354
Familiarity	0.600	0.173	0.140
Preregistration * Familiarity	0.200	0.017	0.068

Table 2. Exploratory analysis: Inclusion Bayes factors for the data PRE exclusions

Analysis of Effects - Trust

Effects	P(incl)	P(incl data)	BF _{incl}
Preregistration	0.600	0.516	0.710
Familiarity	0.600	0.150	0.117
Preregistration * Familiarity	0.200	0.004	0.016

- As an exploratory analysis, would it be informative to rerun the ANOVA using the pilot study results to inform the priors P(M)? It seems reasonable to me to let the pilot data inform conclusions from the main study. On the other hand, perhaps it is unreasonable to use priors based on data for which most participants likely failed the manipulation check?

Although this is an interesting idea, we decline to follow the suggestion. We do not think it will be informative because the pilot and full study are incomparable in many ways (which we did not see earlier on, unfortunately). We also feel, as you do, that it is possibly fruitless to use priors based on data which is also likely unreliable.

- The tables at the end of the manuscript have no identifiers and it is not clear which relate to data from the main study dataset and which relate to data from the pilot study. What is the function of these tables?

These seem to have been rendered as part of the PDF made by the submission portal. They are copies of tables included in the manuscript as part of the text and were required to be uploaded separately to the manuscript LaTeX file. They do not appear at the end of the actual manuscript.

- Data for pilot study is stored inside the “ethics” subdirectory. This seems quite unintuitive.

We agree – the pilot’s now in the appropriate folder.

Reviewer: 4

Comments to the Author(s)

The current paper reports the results of an in principle accepted manuscript concerning the effects of pre-registration and familiarity on a single item measuring trust in a paper’s findings. The research team had a low response rate to their invitation to participate in the study (6%), and they consequently had to modify their sampling strategy. Additionally, many participants did not pass the manipulation checks that the authors planned to use. The final sample was much smaller than the goal (N = 209, planned N = 480). To complete this review, I read the current version of the manuscript alongside my comments on the proposal draft. I also reviewed the files posted on OSF.

1. As a first note, I am unable to locate a copy of the Qualtrics questionnaire for the main study (.qsf or .docx). I can find it for the pilot study but not for the main study. I wanted to check the coding for the manipulation check questions to make sure that they were executed as planned. The high failure rate for the manipulation check questions is really striking.

We apologize that the necessary files were not available for you to review. You will now find a number of useful files on the OSF project directory. We have also described this to other reviewers, and copy our response to them here:

“We have uploaded a short screen capture video for each condition, which shows exactly what the participants saw. Anyone can now see this. Additionally, we have made a video of the study flow and materials as they have been set up in Qualtrics, so that interested people can look ‘behind the curtain’ and see what was done. Lastly, we have provided a working link to a copy of the Qualtrics survey we used for participants, and a .qsf file for those who have a Qualtrics account.”

2. The data file that is posted has participant IP addresses and email addresses. I imagine these should be removed. There is also no codebook (that I could find) to interpret the data file labels.

The first issue you point out was also mentioned by R1. We copy our response to them below:

“This is a mistake made by the first author. It is certainly a breach of ethics, and the file was removed in mid-December. Thank you for drawing our attention to it. A new file is there now, with only non-sensitive information available.” Any data file labels to be interpreted are explained in this document.

3. I think the manuscript should analyze and present results from all available cases (i.e., before exclusions). These should of course be labeled as exploratory follow up, but given how much data is being discarded in the main analysis, I think it would be useful for readers to be able to consider the results without these exclusions. These results for the full sample

are of course hard to interpret, but they are an important part of what we have.

We agree. We have now included a section called ‘Unregistered (exploratory) Analysis in the manuscript which contains the following text and table:

p. 14/15: “A reviewer suggested that, in light of so much data being discarded for the confirmatory analysis, readers may be interested in seeing the results for the full sample. Given this, and uncertainty regarding the validity of the manipulation check questions (which we touch on in the discussion section), we now present the results of the study for all cases (i.e., the data without exclusions) in addition to our preregistered confirmatory analysis. The results of the Bayesian ANOVA on the full data are shown in Table 5. As with the confirmatory analysis, the null model (i.e., without the two independent variables included) is the most likely model given the data observed. Comparison of Tables 4 and 5 show that results for the exploratory and confirmatory analysis are qualitatively similar.”

Table 5

Exploratory analysis: Effects, their prior probabilities, posterior probabilities and the inclusion Bayes factors

Effects	P(incl)	P(incl data)	BF _{incl}
Preregistration	0.600	0.516	0.710
Familiarity	0.600	0.150	0.117
Preregistration * Familiarity	0.200	0.004	0.016

4. I previously worried about the validity/reliability of a single item ad hoc measure of trust. Now that the results are in, I find myself wishing we had more data to consider about this outcome. There are many open questions that the current results leave unsettled, but even if the full intended sample had been collected, there would still be unresolved issues related to what exactly is being measured as the DV here.

In order to acknowledge this potential concern, we added the following paragraph to the discussion:

P. 16: “Finally, we note our use of a single dependent measure of trust. A reviewer has pointed out that, in light of such unclear results, it would have been useful to have more data on the dependent variable. Although one of the aims for this study was to make the survey as simple, quick and easy as possible for participants to respond, we recognize the issues with validity and reliability that a single-indicator measure might present.”

5. I would be supportive of a more detailed analysis of the qualitative/open-ended data to help better understand the failed manipulation.

The participants gave reasons for why they had participated in the first place, how they felt about preregistration/registered reports and what they thought our study hypothesis was. Although we recognize the need to better understand the high rate of failed manipulations, we are not sure what type of analysis on the open-ended data would inform this issue.

6. The current study varied in a number of ways from the pilot study (many at our direction). Comparisons to the pilot study should be very cautious and consider these differences. For instance, the pilot study manipulation of RR was confounded with a reference to journal prestige/impact. That confound was (appropriately) removed here. Hence, the results are not directly comparable. (Note: Another way in which the current study differed from the pilot study was that explanatory text to clarify the meaning of trust was added. Perhaps this text altered the way in which participants responded to the trust question.)

We agree, and have tempered our discussion of the pilot:

P. 5: “Though the experimental design for the pilot study are similar to those described in the full study, they are not the same and comparisons should be cautious. The analysis strategies are identical.”

7. The authors appear to have followed the pre-registered research plan and disclosed deviations from that plan.

Appendix A. Schematic referenced in the response letter and featured in the study manuscript.